# In situ chemical composition measurement of individual cloud residue particles at a mountain site, South China

Qinhao Lin[1,2], Guohua Zhang[1], Long Peng[1,2], Xinhui Bi[1,*], Xinming Wang[1], Fred J. Brechtel[3], Mei Li[4], Duohong Chen[5], Ping'an Peng[1], Guoying Sheng[1], Zhen Zhou[4]

[1] State Key Laboratory of Organic Geochemistry and Guangdong Key Laboratory of Environmental Protection and Resources Utilization, Guangzhou Institute of Geochemistry, Chinese Academy of Sciences, Guangzhou, 510640, PR China

[2] University of Chinese Academy of Sciences, Beijing, 100049, PR China

[3] Brechtel Manufacturing Inc., Hayward, 94544, California, USA

[4] Atmospheric Environment Institute of Safety and Pollution Control, Jinan University, Guangzhou 510632, PR China

[5] State Environmental Protection Key Laboratory of Regional Air Quality Monitoring, Guangdong Environmental Monitoring Center, Guangzhou 510308, PR China

* Correspondence to: Xinhui Bi (bixh@gig.ac.cn)

Tel.: +86-20-85290195

**Highlights**

1. EC-containing particles were the largest fraction of the total cloud residues (49.3% by number), dominating in the range of 0.2-1.0 μm.

2. The Nf of the cloud residue types was influenced by air mass chemistry.

3. Amine particles represented from 0.2% to 15.1% by number of the total cloud residues when air masses changed from northerly to southwesterly.

4. Compared with non-activated particles, nitrate intensity decreased in cloud residues except dust type.

**Abstract**

To investigate how atmospheric aerosol particles interact with chemical composition of cloud droplets, a ground-based counterflow virtual impactor (GCVI) coupled with a real-time single-particle aerosol mass spectrometer (SPAMS) was used to assess the chemical composition and mixing state of individual cloud residue particles in the Nanling Mountain Range (1,690 m a.s.l.), South China, in Jan 2016. The cloud residues were classified into nine particle types: Aged elemental carbon (EC), Potassium-rich (K-rich), Amine, Dust, Pb, Fe, Organic carbon (OC), Sodium-rich (Na-rich) and Other. The largest fraction of the total cloud residues was the Aged EC type (49.3% by number), followed by the K-rich type (33.9% by number). Abundant Aged EC cloud residues that mixed internally with inorganic salts were found in air masses from northerly polluted areas. The number fraction (Nf) of the K-rich cloud residues increased within southwesterly air masses from fire activities in Southeast Asia. When air masses changed from northerly polluted areas to southwesterly ocean and livestock areas, the Amine particles increased from 0.2% to 15.1% of the total cloud residues by number. The Dust, Fe, Pb, Na-rich and OC particle types had a low contribution (0.5-4.1% by number) to the total cloud residues. Higher fraction of nitrate (88-89% by number) was found in the Dust and Na-rich cloud residues relative to sulfate (41-42%) and ammonium (15-23%). Higher intensity of nitrate was found in the cloud residues relative to the ambient particles. Compared with non-activated particles, nitrate intensity decreased in cloud residues except dust type. To our knowledge, this study is the first report on in situ observation of the chemical composition and mixing state of individual cloud residue particles in China.

Keywords: GCVI, SPAMS, cloud residues, mixing state, South China

**1 Introduction**

Aerosol-cloud interactions influence the thermodynamic and radiation balance of the atmosphere (IPCC, Boucher et al., 2013). Anthropogenic particles can increase number concentration of small cloud droplets, and, in turn, affect reflectivity and life time of clouds (Stier et al., 2005; Lohmann et al., 2007; Rosenfeld et al., 2008). In situ cloud chemical measurements have shown varied chemical composition of cloud water or residues at various regions (Sorooshian et al., 2007a; Roth et al., 2016; Li et al., 2017). Despite  a large number of aerosol/cloud studies over the past 20 years, the uncertainty for evaluating radiative forcing due to aerosol-cloud interactions has not been reduced (Seinfeld. et al., 2016). Therefore, it is crucial to assess how atmospheric aerosol particles contribute and interact with cloud droplets.

The ability of aerosol particles to act as cloud condensation nuclei (CCN) is dependent on the size and chemical composition of particles at a given supersaturation (McFiggans et al., 2006). Wiedensohler et al. (2009) found that the enhancement of particles CCN ability was related to an increase in the average sulfate mass concentration. Dusek et al. (2006) demonstrated that CCN behavior was more effected by aerosol size than chemical composition. Meanwhile, aerosol mixing state also play an important role in the ability of aerosol to act as CCN. It has been reported that freshly emitted elemental carbon (EC) particles generally exhibit low CCN activity, whereas aged EC particles show high CCN activity after experienced atmospheric processes (Zhang et al., 2008). Pratt et al. (2011) found that number fractions of ammonium or oxalate internally mixed with biomass

burning particles increased with an aged time of 81-88 min, which promote CCN
behavior. Laboratory studies have shown that low-solubility organic particles internally
mixed with ammonium sulfate would suppress water uptake of mixed particle and thus
might affect CCN activity (Wise et al., 2003; Svenningsson et al., 2006; Sjogren et al.,
2007). An over prediction of CCN concentration by up to 35% was estimated based on
particle internal mixing state assumption (Medina et al., 2007; Collins et al., 2013). The
influence of mixing state on aerosol CCN activity varies depending on the proximity to
the pollution plume source and/or photochemical ageing activity (Ervens and Volkamer,
2010). More detailed measurements to characterize the mixing state of CCN particles
would improve our understanding of aerosol-cloud interactions.
The combined technique of a counterflow virtual impactor (CVI) and an Aerosol Mass
Spectrometer (AMS) or other online/offline single particle instruments  is widely used to
characterize the chemical composition and/or mixing state of cloud/fog droplet residue
particles. These studies were mainly conducted in North America including Wyoming
(Pratt et al., 2010a), Ohio (Hayden et al., 2008), Oklahoma (Berg et al., 2009), Florida
(Cziczo et al., 2004; Twohy et al., 2005), California (Coggon et al., 2014), Europe
including Schmücke (Roth et al., 2016; Schneider et al., 2017), Jungfraujoch (Kamphus
et al., 2010), Åreskutan (Drewnick et al., 2007), Scandinavia (Targino et al., 2006),
Arctic (Zelenyuk et al., 2010), Central America (Cziczo et al., 2013), West Africa
(Matsuki et al., 2010) and marine areas  (Twohy and Anderson 2008; Twohy et al., 2009;
Shingler et al., 2012).
Over the past three decades, China has undergone rapid economic growth accompanied
by increased aerosol emissions. Although scientists have worked to increase our
understanding of an emissions inventory and the temporal and spatial variation of
atmospheric aerosols in China (Zhang et al., 2012b), only few studies have employed
direct observation of the chemical composition and mixing state of cloud/fog droplets. Li
et al. (2011b) utilized transmission electron microscopy to obtain the mixing state of
individual ambient particles during cloud events at Mt. Tai in northern China. This result
showed that sulfate-related salts dominated in larger particles. Bi et al. (2016) used a
ground-counterflow virtual impactor (GCVI) coupled with a real-time single particle
aerosol mass spectrometer (SPAMS) to explore the chemical composition and mixing
state of individual fog residual particles at ground level in an urban area of South China.
They found an abundance of EC-containing particles in fog residues.
Here, we present a study on the chemical composition and mixing state of individual
cloud residue particles at a mountain site in South China. The same experimental
methods of Bi et al. (2016) were used in this study. The size distribution, chemical
composition and mixing state of cloud residues during cloud events are discussed.
Moreover, the chemical compositions of ambient and non-activated particles were also
compared with the cloud residues. The aim of this study is to assess the potential effects
of anthropogenic aerosols from regional transportation on cloud formation and to
investigate the dominant particle types in cloud droplets at a mountain site in South China.

**2 Experimental**
**2.1 Measurement site**
Our measurements were carried out during 15-26 Jan, 2016. The sampling site was
located in the Nanling Background Station (112 °53' 56" E, 24° 41' 56" N, 1,690 m a.s.l.)
at the National Air Pollution Monitoring System in South China (Figure S1). This station
is located at 200 km north of the metropolitan city Guangzhou and 350 km north of the
South China Sea (Figure S1). This site is also surrounded by a national park forest (273
$km^2$), where there are scarcely any emissions from anthropogenic activities. During the
winter monsoon period, air pollution from northern China moves to southern China and
crosses the study region (Lee et al., 2005).

**2.2 Instrumentation**
In this study, a GCVI inlet system (GCVI Model 1205, Brechtel Mfg. Inc.) was used to
sample cloud droplets with a diameter greater than 8 μm. The ambient temperature on
average was 6.9 ℃ (ranging from -7.2 to 11.4 ℃) during cloud events. Only 20 cloud
residues that accounted for 0.08% of the total cloud residues were detected when the
ambient temperature was below -7 ℃ observed from 06:00 to 08:00 on 23 Jan. Thus,
cloud droplets were dominated by liquid water droplets. The measurements of the droplet
size spectra in this region performed during the winter of 1999-2001 showed that size of
cloud droplets ranged from 4 to 25 μm with average size of 10 μm and a corresponding
liquid water content of 0.11-0.15 g $m^{-3}$ (Deng et al., 2007). Previous study in other
mountain site also showed an average size at ~10 μm (Borys et al., 2000). Hence,
assuming that size distribution of cloud droplets mostly was above 8 μm in this region.
The sampled cloud droplets were passed through an evaporation chamber (air flow
temperature at 40 ℃), where the associated water was removed and the dry residue
particles (with the air flow RH lower than 30%) remained. A stream of filtered and
heated ambient air (counterflow) was provided by a compressor. The particle
transmission efficiency of the cut size (8 μm) was 50%. The enrichment factor of the
particles collected by the GCVI inlet was estimated to be 5.25 based on theoretical
calculation (Shingler et al., 2012). Ambient particles were collected through an ambient
inlet with a cut-off aerodynamic diameter ($d_a$) of 2.5 μm when cloud-free periods were
present. Non-activated (interstitial) particles were sampled through the ambient inlet
during the cloud events in this study. The ambient or non-activated particles inlet was
dried using a silica gel diffusion dryer. During cloud-free periods, a ratio of particle
concentration measured behind the CVI (below 1 cm$^{-3}$) to ambient aerosol concentration
(2,000 cm$^{-3}$) was 0.0005, indicating that instances of particle breakthrough and small
particle contamination were absent. The cloud droplet residues, ambient or non-activated
particles were subsequently analyzed by a suite of aerosol measurement devices,
including a SPAMS (Hexin Analytical Instrument Co., Ltd., Guangzhou, China), a
scanning mobility particle sizer (SMPS) (MSP Cooperation) and an aethalometer (AE-33,
Magee Scientific Inc.).
A detailed operational principle of the SPAMS has been described elsewhere (Li et al.,
2011a). Briefly, aerosol particles are drawn into SPAMS through a critical orifice. The
particles are focused and aerodynamically sized by two continuous diode Nd:YAG laser
beams (532 nm). The particles are subsequently desorbed/ionized by a pulsed laser (266
nm) triggered exactly based on the velocity of the specific particle. The positive and
negative ions generated are recorded with the corresponding size of individual particles.
Polystyrene latex spheres (Nanosphere Size Standards, Duke Scientific Corp., Palo Alto)
of 0.2-2.0 μm in diameter were used to calibrate the sizes of the detected particles. The
ambient pressure was 830 hPa (826-842 hPa) during the measurements and the
calibration. Particles measured by SPAMS mostly fell within the size range of $d_{va}$ 0.2-2.0
μm (Li et al., 2011a).

**2.3 Definition of cloud events**
To reliably identify the presence of cloud events, an upper-limit visibility threshold of 5
km and a lower-limit relative humidity (RH) threshold of 95% were set in the GCVI
software (Bi et al., 2016). Three long-lasting cloud events occurred during the periods of
16:00 (local time) 15 Jan - 07:00 17 Jan (cloud I), 20:00 18 Jan - 12:00 19 Jan (cloud II)
and 17:00 19 Jan - 13:00 23 Jan (cloud III), as marked in Figure 1. In addition, a cloud
event occurred at 14:40 - 15:00 on 17 Jan, but we did not do an analysis due to the short
duration. The measured cloud residual concentration was integrated by the SMPS and
was then corrected by the enrichment factor and transmission efficiency of the GCVI.
The corrected cloud residual concentrations on average were 436 cm$^{-3}$, 568 cm$^{-3}$ and 544
cm$^{-3}$ for cloud I, cloud II and cloud III, respectively (Figure S2). From 10:00 21 Jan to
13:00 23 Jan, cloud residues and non-activated particles were alternately sampled with an
interval of one hour. During this period, a ratio of number residues to total number
particles (sum of cloud residues and non-activated particle) on average was 0.43±0.20.
Low levels of PM$_{2.5}$ (~ 12.7 μg m$^{-3}$) exclude the influence of hazy days. A rainfall
detector of the GCVI system was also used to exclude rain droplet contamination. When
cloud events occurred without precipitation, sampling was automatically triggered by the
GCVI control software (Bi et al., 2016).


**2.4 Particle classification**

During this study period, a total of 73,996 particles including 49,322 ambient, 23,611 cloud residual and 1,063 non-activated particles with bipolar mass spectra were chemically analyzed in the size range of $d_{va}$ 0.2-1.9 μm. The sampled particles were firstly classified into 101 clusters using an Adaptive Resonance Theory neural network (ART-2a) with a vigilance factor of 0.75, a learning rate of 0.05, and 20 iterations (Song et al., 1999). Then by manually combining similar clusters, eight major particle types Aged EC, Potassium-rich (K-rich), Amine, Dust, Fe, Pb, Organic carbon (OC), and Sodium-rich (Na-rich) with distinct chemical patterns were obtained, which represented ~99.9% of the population of the detected particles. The remaining particles were grouped together as "Other". Assuming that the number of individual particles followed Poisson distribution, standard errors for number fraction of particle type were estimated (Pratt et al., 2010a).

**3 Results and discussion**

**3.1 Back trajectories and meteorological conditions**

Back trajectories in this study were calculated using the Hybrid Single Particle Lagrangian Integrated Trajectory (HYSPLIT Model). A height of the HYSPLIT model in the study region (a spatial resolution of $0.5° \times 0.5°$) is averaged 500 m a.s.l., lower than height of the observed site (1,690 m a.s.l.). Thus, a height of 1,800 m a.s.l. (approximately 100 m above the observed site) was chosen as an endpoint in the model. The station was mainly affected by southwesterly or northerly air masses in this study (Figure 2). In addition, the beginning altitude of the southwesterly air masses traversed at

lower heights relative to the northerly air masses (Figure 2). The southwesterly air masses,
accompanied by warm moist airflows, occurred during 15-17 and 20-21 Jan, which
promoted high RH condition (Figure 1). Conversely, the northerly air masses, associated
with cool dry airstreams, occurred during 18 and 23-24 Jan and led to a decrease in
temperature and relative humidity. Note that, on 18-19 and 22-23 Jan, the air mass
encountered initial mixing of northerly cloud-free air and southwesterly cloudy air.
Entrainment of nuclei particles originated from northern air masses might be activated to
cloud droplets (Sect. 3.4).
Meteorological conditions were unstable, with high southwesterly flow ($\sim 6.5$ m s$^{-1}$)
during 15-17 and 20-22 Jan (Figure 1). The level of PM$_{2.5}$ remained a low value of
approximately 3 μg m$^{-3}$ for this time period. A high level of PM$_{2.5}$ ($\sim$20 μg m$^{-3}$) was
observed during 18 Jan when the northerly flow dominated. Similarly, the average PM$_{2.5}$
value reached 24 μg m$^{-3}$ during 24 Jan. Although the local northerly and southwesterly
flows occurred alternately, the particles were still originated from the northerly air mass
for this period (Figure 2). During 23-24 Jan, a sharp decrease in temperature (Figure 1)
was observed due to a cold wave associated with a violent northerly flow. The wind
speed during the cold wave exceeded the upper-limit speed ($\sim$12 m/s) of a wind speed
sensor.

**3.2 The chemical characterization of cloud droplet residues**
Figure 3 shows the average positive and negative mass spectra of the main six particle
types. The Aged EC particles were characterized by EC cluster ions (e.g., m/z $\pm12C^{+/-}$,
$\pm36C_3^{+/-}$, $\pm48C_4^{+/-}$, $\pm60C_5^{+/-}$, …) and a strong K$^+$ ion signal (m/z 39K$^+$) as well as a
sulfate ion signal (m/z $-97HSO_4^-$), and some minor organic markers (m/z $27C_2H_3^+$,
$43C_2H_3O^+$) (Moffet and Prather, 2009). EC particles mainly originated from combustion
processes (Bond et al., 2013). The strong $K^+$ ion signal in the Aged EC particles implies
partially originated from biomass burning sources (Bi et al. 2011). The Aged EC particle
type was the largest fraction (49.3% by number) of the total cloud residues (Figure S3).
In addition, number fraction (Nf) of the Aged EC residues significantly decreased from
54.1% in the size range of 0.2-1.0 μm to 19.2% in the size range of 1.1-1.9 μm (Figure 4).
Note that the chemical composition of cloud residues is dependent on the particle size
(Roth et al., 2016), and the number reported for each particle type might suffer from the
bias related to size-dependent transmission efficiency (Qin et al., 2006). The relative
fraction of cloud residues in 0.1 μm size interval is presented to minimize the influence of
size-dependent transmission efficiency of single particle mass spectrometry (Roth et al.,

2016).

The K-rich particles exhibited the highest peak at m/z $39K^+$, mainly combined with

sulfate and nitrate (m/z $-46NO_2^-$, $-62NO_3^-$) and presumably derived from biomass/biofuel
burning source (Moffet et al., 2008; Pratt et al. 2011; Zhang et al., 2013).  An aged time
of 81-88 min biomass burning particles were found to show an increase in the mass
fractions of ammonium, sulfate, and nitrate (Pratt et al. 2011). In this study, the K-rich
particles would be expected to experience aged process due to strong sulfate and nitrate
signals (Hudson et al. 2004; Pratt et al. 2011). Aged biomass burning particles can
participate in cloud droplets formation and show an effective CCN activity (Pratt et al.
2010a). The K-rich particle type, the second largest contributor, accounted for 33.9% by
number of the total cloud residues (Figure S3).

The abundant aged soot/EC and biomass burning particles were often detected in cloud residues (Pratt et al., 2010a; Roth et al., 2016). The contribution of local anthropogenic origins to aged soot and/or biomass burning particles in cloud/fog residues has been reported in Schmücke (Roth et al., 2016) and Guangzhou city (Bi et al., 2016). At the North Slope of Alaska, the measurement of biomass burning particles in cloud residues mainly resulted from local vicinity or as far away as Siberia and Asian sources (Zelenyuk et al., 2010; Hiranuma et al., 2013). Similarly, the majority of Aged EC and K-rich cloud residues observed here are expected to originate from long-range transportation due to insignificant sources of local anthropogenic emissions and the fire dots (Figure 2). At the Jungfraujoch station (3,580 m a.s.l.) in Europe, the K-rich (biomass burning) particles only contributed 3% of the cloud droplets, and the Aged EC residuals were insignificant (<1% by number) (Kamphus et al., 2010). The Jungraujoch station is predominantly within the free tropospheric condition, such that the biomass burning contribution can be expected to be lower than at other sites.

The Amine particles were characterized by related amine ion signals at m/z $58C_2H_5NHCH_2^+$ (diethylamine, DEA), $59N(CH_3)_3^+$ (trimethylamine, TMA) and $86C_5H_{12}N^+$ (triethylamine, TEA) (Angelino et al., 2001; Moffet et al., 2008). This particle type also contained sulfuric acid ion signals at m/z $-195H(HSO_4)_2^-$, indicative of acidic particles (Rehbein et al., 2011). The Amine particles represented 3.8% by number of the total cloud residues (Figure S3). Higher Nf of the Amine residues was detected in the size range from 0.7 to 1.9 μm relative to the size range from 0.2 to 0.6 μm (16.7% versus 0.4%), as shown in Figure 4. Aqueous reactions improving the participation of amine have been observed in Guangzhou (Zhang et al., 2012a) and Southern Ontario

(Rehbein et al., 2011). A recent study also showed a clear enhancement of amine-
containing particles in cloud residues compared to the ambient particles (9% versus 2%
by number) (Roth et al., 2016). It indicates a preferential formation of amine within the
cloud, which is in contrast to the observations of Bi et al. (2016). It might suggest that
enhancement of particle amine is not only depend on high RH or fog/cloud process, but
also sensitive to other parameters, such as presence of gas phase amine source (Rehbein
et al., 2011).

The Dust particles presented significant ions at m/z $40Ca^+$, $56CaO^+/Fe^+$, $96Ca_2O^+$

and $-76SiO_3^-$ and sulfate as well as nitrate markers. Previous studies showed that dust
particles that are internally mixed with sulfate and nitrate are expected to act as CCN
(Twohy and Anderson 2008; Twohy et al., 2009), despite sulfate and nitrate partially
forming from in-cloud production. This type contributed 2.9% by number of the total
cloud residues (Figure S3). A slight increase in Nf of the Dust residues was observed in
size range above 0.5 μm relative to that below 0.5 μm (3.0% versus 1.0% by number). At
Mt. Tai in northern China, a high concentration of $Ca^{2+}$ in cloud/fog water was mainly
attributed to a sandstorm event during the spring season (Wang et al., 2011). At Mt. Heng
in southern China, the abundant crust-related elements (e.g., Al) observed in cloud water
is due to Asian dust storms that occurred in March-May (Li et al., 2017). Based on the
backward trajectory, the site was unlikely affected by sandstorm source in northwestern
China during the cloud events. Local dust emissions by anthropogenic-disturbing soils or
removing vegetation cover can be excluded as a result of forest protection. Additionally,
dust residues may have occupied larger CCN (Tang et al., 2016), which cannot be
detected by the SPAMS. Hence, a low fraction (2.9% by number) of dust cloud residue
might be due to the limitation of the SPAMS.
The Fe particles had its typical ions at m/z $56Fe^+$ and internally mixed with sulfate
and nitrate, made up 4.1% by number of the total cloud residues. Approximately 16% of
the Fe cloud residues contained $Ca^+$ peak (m/z 40). Predominant Fe ion peaks possibly
indicates the contribution from anthropogenic sources (Zhang et al., 2014), especially the
northern air masses across iron/steel industrial activities in Yangtze River Mid-Reaches
city clusters (Figure 2). The contribution of anthropogenic and natural Fe-containing
particles sources (Moteki et al., 2017) to observed Fe-containing residues is expected.
The presence of Fe in the cloud droplets play an important role in aqueous-phase $SO_2$
catalytic oxidation in cloud processing (Harris et al., 2013), thus accelerating the sulfate
content of Fe-containing particles in cloud processing.
The Na-rich particles were mainly composed of ion peaks at m/z $23Na^+$ and $39K^+$ in
the positive mass spectra, and nitrate and sulfate species in the negative mass spectra,
made up 3.0% by number of the total cloud residues. Na-rich particles are formed from
varied sources including industrial emissions, sea salt or dry lake beds (Moffet et al.
2008). The Nf of Na-rich cloud residues did not increase from continental (Northerly) air
mass on 19 Jan to maritime (southwesterly) air mass on 21 Jan (3.3% versus 2.4% by
number). However, the related sea salt ion peak area (m/z, 81/83 $Na_2{}^{35}Cl^+/Na_2{}^{37}Cl^+$) were
enhanced for Na-rich particles origination from maritime air mass relative to continental
air mass (3.8 $\pm$ 2.4 times). The continental air masses crossed industrial areas where the
Yangtze River Mid-Reaches city cluster is located (Figure 2). Industrial emissions were a
possible contributor to Na-rich particles under the influence of continental air masses
(Wang et al. 2016). This might suggest that the Na-rich particles were originated from
both industrial emissions and sea salts.

The OC, Pb and Other particle types contributed 0.1%-2.3% by number to the total

cloud residues (Figure S3). Their average mass spectra can be found in Figure S4. The
OC particles presented dominant intense OC signals (e.g., m/z $27C_2H_3^+$, $37C_3H^+$,
$43C_2H_3O^+$ and $51C_4H_3^+$) and abundant sulfate. Presence of $K^+$ signal was found in the
OC particles, suggesting possible biomass burning sources (Bi et al. 2011). OC particles
might exist in smaller cloud residues (Sellegri et al., 2003a), which cannot be detected by
the SPAMS. The Pb particles showed its typical ions at m/z $208Pb^+$ and internally mixed
with $K^+$ and $Cl^-$. Previous studies have found that K and Cl internally mixed with Pb
particles have a possible origination of waste incineration (Zhang et al., 2009) or iron and
steel products manufacturing facilities (Tsai et al., 2007). Only three particles were found
containing calcium, organic carbon, organic nitrogen and phosphate ion signals, implying
a possible existence of biological particles (Pratt et al., 2009a). Such particles were
classified as the Other type due to low number.

Previous measurements have found that dust, playa salts, sea salt or metal particles

were often enriched in larger cloud droplets (~20 μm) (Bator and Collett, 1997; Moore et
al., 2004; Pratt et al., 2010b). Organic carbon tended to be enriched in small cloud/fog
droplets, extending to 4 μm (Herckes et al., 2013). The size of cloud droplets were above
8 μm in the present study. Additionally, the particle transmission efficiency increased
with increasing cloud droplet size (Shingler et al., 2012). Thus, it partially leads to
relatively larger fractions of the observed Dust, Na-rich and metal cloud residues, and the
less fraction of the OC cloud residues in this study.

**3.3 Mixing state of secondary species in cloud residues**

The high Nfs of sulfate-containing particles were found in the K-rich (91%), OC (100%), Aged EC (98%), Pb (74%), Fe (93%) and Amine (99%) cloud residues, as shown in Figure 5. Lower Nfs of sulfate-containing particles were observed in the Na-rich (41%) and Dust (42%) cloud residues. In contrast, nitrate-containing particles contributed 89% and 88% by number to the Na-rich and Dust cloud residues, respectively. The acid displacement reaction of sea salt chloride (Na-rich particles) by $HNO_3$ might lead to a depletion of chloride and enhancement of nitrate (Laskin et al., 2012). Similarly, the heterogeneous chemistry of $HNO_3$ in the dust particles also contributes the preferential enrichment of nitrate (Tang et al., 2016). Moreover, after activation, uptake of gas-phase $HNO_3$ would increase nitrate level in the cloud residues (Schneider et al., 2017). The nitrate in the cloud residues was thought to be in the form of ammonium nitrate by estimating the ratio of m/z 30 to m/z 46 in AMS data (Drewnick et al., 2007; Hayden et al., 2008). Relative to nitrate, low portions of ammonium (m/z, $18NH_4^+$) in the Na-rich (23% by number) and Dust (15% by number) cloud residues suggest that in this region, ammonium nitrate was not a predominant form of nitrate in the two cloud residual types. The Na-rich and Dust types were mainly composed of alkaline ion peaks (m/z, $23Na^+$, $39K^+$ and $40Ca^+$) in position mass spectra (Figure 3), accompanied with larger fraction (88-89%) of nitrate. Thus, our data suggests that nitrate might exist in the form of $Ca(NO_3)_2$, $NaNO_3$ or $KNO_3$ in the Dust and Na-rich cloud residues. It should be noted that the evaporation chamber of the GCVI may lead to a reduction of ammonium nitrate in the cloud residues (Hayden et al., 2008). The nitrate-containing particles accounted for only 46% by number of the Aged EC cloud residues, which is significantly less than the

sulfate-containing particles. Previous field studies have found that Aged EC (soot) fog/cloud residues are mainly internally mixed with sulfate (Pratt et al., 2010a; Harris et al., 2014; Bi et al., 2016). Aged EC particles mixed with sulfate are good CCN, rather than formed by in-cloud processing (Bi et al., 2016; Roth et al., 2016). Laboratory measurements have also demonstrated that EC particles internally mixed with sulfate showed a high hygroscopic behavior and thus affect CCN ability (McMeeking et al., 2011). High portions (75-86% by number) of ammonium-containing particles were observed for the OC and Aged EC cloud residues, suggesting that ammonium will mostly be in the form of ammonium sulfate or ammonium nitrate for two cloud residual types (Zhang et al., 2017). This result also implies that ammonium-containing particles are preferentially activated or enhanced uptake of gaseous $NH_3$ to neutralize acidic cloud droplets for the OC and EC types.

Water soluble organics (e.g., amine and oxalate) have previously been measured in cloud water/residues (Sellegri et al., 2003b; Sorooshian et al., 2007a; Pratt et al., 2010a). The presence of TMA (93% by number) in the Amine cloud residues is expected to promote water uptake activity (Sorooshian et al., 2007b). A total of 3,410 oxalate-containing particles (m/z, $-89HC_2O_4^-$) represented 14.4% by number of the total cloud residues, which was mainly associated with the K-rich cloud residues (~70% by number). Oxalate-containing particles (~30% by number) in the metal (Pb, Fe) cloud residues might be in the form of metal oxalate complexes from reactions of in-cloud formation oxalate with metals (Furukawa and Takahashi, 2011). Oxalate can readily partition into the particle phase to form amine salts (Pratt et al., 2009b). It may facilitate the entrainment of oxalate (33% by number) in the Amine residues. A low fraction (4%) of

oxalate-containing particles in the OC type is a result of restrictive classification. Classification of the OC particles is mainly based on intense organic carbon ion signals (e.g., m/z $27C_2H_3^+$, $37C_3H^+$, $43C_2H_3O^+$ and $51C_4H_3^+$).The majority of oxalate-containing particles were internally mixed with the K-rich type. Therefore, oxalate was classified to the K-rich type and probably contributed from biomass burning. The K-rich particles could contain an abundance of organics (Pratt et al. 2011), however, the signals of organics were covered by the potassium due to its high sensitivity to the laser.

**3.4 Comparison of cloud residues in different air mass sources**

Figure 6 displays the hourly detected particle counts and Nf values of the nine types of cloud residues and ambient particles. The Nf of the Aged EC type showed a very abrupt increase from cloud residues to ambient particles on Jan 17. The ambient RH showed an abrupt decrease from nearly 100% at 10:00 to 85% at 11:00 on 17 Jan (Figure 1). The ambient temperature also decreased from 10 ℃ at 11:00 to 4 ℃ at 18:00 on 17 Jan (Figure 1). These changes imply that the air mass shifted from southwesterly cloudy air to northerly cloud-free air around noon on 17 Jan (Figure 2). The entrained particles originated from northern air mass might have insufficient supersaturation to be activated as cloud droplets. It resulted in the remarkable increase of the Aged EC particles in ambient particles on Jan 17 (Figure 6).

The ambient RH increased from 60% at 19:00 to nearly 100% at 21:00 on 18 Jan (Figure 1). The ambient temperature also increased from 1.3 ℃ at 22:00 on 18 Jan to 3.2 ℃ at 06:00 on 19 Jan (Figure 1). These changes imply that the air mass changed from northerly cloud-free air to southwesterly cloudy air at night on 18 Jan (Figure 2). During

18-19 Jan, the cloud residues and ambient particles showed similar chemical
characteristics and were dominated by Aged EC particles (Figure 6). A lack of significant
variation in the particle types for this period suggests that nuclei particles originated from
northerly cloud-free air could be activated to become cloud droplets. When a cloud-free
event occurred at 11:00-17:00 on 19 Jan, ambient particles remained a high level of $PM_{2.5}$
($\sim$ 22.7 $\mu$g m$^{-3}$). The southwesterly wind flow on 19-20 Jan was too weak ($\sim$ 2.75 m s$^{-1}$)
to dilute particles originated from the northerly air masses (Figure 1). Additionally, a
high RH (90%) air mass at height 1,500 m (a.s.l.) gradually moved to northern China
from 19 to 20 Jan (Figure S5). These changes might have led to similar residual particle
types observed from 19 Jan to 20 Jan, although the site encountered southwesterly cloudy
air on 19-20 Jan (Figure 2).

As mentioned above, the Nf of the cloud residue types significantly changed as the

air mass origin varied from northerly to southwesterly. To further investigate the
influence of air mass history, we selected to analyze cloud residues that had arrived from
a northerly air mass on 18-19 Jan as compared to cloud residues that originating from a
southwesterly air mass during the periods of 16-17 and 21-22 Jan. The detected number
of cloud residues for both the northerly and southwesterly air masses are given in Table
S1. The southwesterly air masses accompanied by high relative humidity (90%) (Figure
S5) may have triggered particles activated to CCN prior to their arrival to the sampling
site.

The K-rich type was found to contribute 23.9% to the cloud residues in the northerly

air mass, which was significantly lower than its contribution to the southwesterly air mass
(51.5%), as shown in Figure 7. A similarity in averaged mass spectrum of the K-rich
residues was found for the southwesterly and northerly air masses (Figure S6). The
considerable increase of K-rich cloud residues suggests a major influence of regional
biomass-burning activities. Biomass-burning emissions from Southeast Asia, including
Myanmar, Vietnam, Laos and Thailand, where abundant fire dots are observed (Figure 2),
could have been transported to the sampling site under a southwesterly air mass (Duncan
et al., 2003). In contrast, the Aged EC type represented only 23.7% of the cloud residues
under the influence of the southwesterly air mass, which was significantly lower than
observations for the northerly air mass (59.9%), as shown in Figure 7. This result
suggests that the northern air mass has a greater influence on the presence of Aged EC
cloud residues.
An obvious increase in Nf of the Amine type was observed in the southwesterly air
mass (15.1%) compared to the northerly air mass (0.2%), as shown in Figure 7. This data
implies that the sources or formation mechanisms of amine in cloud residues varied in
different air masses. The southwesterly air masses arrived from as far as the Bay of
Bengal and then travelled through Southeast Asia region before reaching South China
(Figure 2). The potential gas amine emissions from ocean (Facchini et al., 2008) and
livestock areas (90 million animals, data was available at the website
http://faostat3.fao.org) in Southeast Asia region might promote the enrichment of amine
particles. Furthermore, after activation, the partitioning of the gas amine on cloud
droplets may further contribute to the enhanced Amine cloud residues (Rehbein et al.,
2011), especially for air masses delivered via routes with high relative humidity, as
mentioned above (Figure S5). In contrast, northerly air mass accompanied with dry
airstreams may unfavorably induce the partitioning of gas amines into the particle phase
(Rehbein et al., 2011).

**3.5 Comparison of cloud residues with ambient and non-activated particles**
A direct comparison between cloud residues and ambient particles was limited because of
their differences in air mass origins. During the sampling period, the cloud events
occurred once the southwesterly air masses were dominant. Hence a comparison between
cloud residues and ambient particles cannot be addressed under the influence of
southwesterly air masses. Here, we chose five hours before and after the beginning of the
cloud II period in order to compare cloud residues and ambient particles with similar
northerly air mass origins, as discussed in Sect. 3.4.
The cloud residues and non-activated particles were alternately sampled with an
interval of one hour from 21 Jan to 23 Jan. The ambient temperature decreased from 6 ℃
at 11:00 to 0 ℃ at 23:00 on 22 Jan (Figure 1). Ambient particles level (sum of residual
and non-activated particles) showed a clearly increase from 156 $cm^{-3}$ to 1460 $cm^{-3}$ during
this period (Figure S2). Thus, the data suggests that the initial mixing of northerly cloud-
free air and southwesterly cloudy air occurred around noon on 22 Jan. A reduction of
supersaturation due to entrainment of the dry northern air mass might have insufficient
moisture to activate small particles, leading to unactivated particles above 0.2 μm
( Figure S7) (Mertes et al., 2005; Kleinman et al., 2012; Hammer et al., 2014), which can
be detected by the SPAMS.
The contribution of K-rich particles in cloud residues (23.9%) slightly decreased
relative to ambient particles (30.7%), as shown in Figure 7. Previous studies have found
that there were no significant changes in Nf of biomass-burning particles for cloud
residues relative to ambient particles (Pratt et al., 2010a; Roth et al., 2016). The biomass-
burning particles internally mixed with soluble species (e.g., sulfate, nitrate and oxalate)
enhanced their ability to act as CCN, as discussed in Sect. 3.3. Kamphus et al. (2010)
reported that biomass-burning particles account for only 3% of cloud residues compared
with 43% of ambient particles, and they suspected that biomass-burning particles might
exist in the form of tar balls (hydrophobic materials). A slight increase in Nf of the Aged
EC cloud residues (59.9%) was observed relative to ambient particles (53.8%), as shown
in Figure 7. In general, freshly emitted EC particles are less hydrophilic and are not
active as CCN (Bond et al., 2013). The Aged EC particles show a high degree of internal
mixing with secondary inorganic compounds in this study (Figure 5), improving their
ability to act as CCN. The remaining particle types showed no clear differences in Nf
between cloud residues and ambient particles.
When comparing the cloud residues with non-activated particles, a significant change
in Nf was found for the Aged EC and K-rich type. A higher Nf of K-rich particles and a
lower Nf of EC particles were found for the cloud residues relative to the non-activated
particles (Figure 7). Entrainment of northerly cloud-free air might lower supersaturation
during this period. Aged EC particles may require very high supersaturation to grow into
cloud droplets and thus, only form hydrated non-activated aerosol (Hallberg et al., 1994).
Figure 8 and 9 show the differences in average mass spectra for cloud residues
versus ambient particles, as well as cloud residues versus non-activated particles,
respectively. Nitrate intensity (average ion peak area) enhanced in the cloud residues
when compared to ambient particles. In addition, nitrate-containing particles accounted
for 70% of the cloud residues compared to 38% of the ambient particles. Drewnick et al.
(2007) suggested that rather than sulfate, high nitrate content in pre-existing particles
preferentially acted as cloud droplets. Compared with nitrate-containing ambient particles,
larger size of containing-nitrate residues (Figure S8) possibly reflect the uptake of
gaseous $HNO_3$ during cloud process (Hayden et al. 2008; Roth et al., 2016). A recent
study also confirmed that the update of gaseous $HNO_3$ is an important contributor for the
increased nitrate level in the cloud residuals (Schneider et al., 2017). Interestingly, we
observed a decrease in nitrate intensity in cloud residues except dust type (Figure 9), and
a large size distribution of nitrate-containing cloud residues (Figure S7) when compared
with non-activated particles. This result suggests that particle size, rather than nitrate
content, plays a more important role in the activation of particles into cloud droplets.
Sulfate intensity enhancement was only observed in the OC cloud residues relative to
both ambient and non-activated particles. Although the in-cloud addition of sulfate can be
produced from aqueous Fe-catalyzed or oxidation by $H_2O_2/O_3$ reactions (Harris et al.,
2014), sulfate abundance was found in the Fe cloud residues relative to non-activated
particles, but no enhancement relative to ambient particles. Previous studies also showed
that the mass or number fraction of sulfate-containing particles in the cloud residues
changed between ambient and non-activated particles (Drewnick et al., 2007; Twohy and
Anderson, 2008; Schneider et al., 2017). However, the reason for these changes remains
unclear.
The in-cloud process has been reported to be an important pathway for the production
of amine particles (Rehbein et al., 2011; Zhang et al., 2012a). In this study, no
remarkable change in Nf of the Amine cloud residues was obtained relative to the
ambient particles (0.2% versus 0.2% by number), as shown in Figure 7. The absence of
amine species in cloud residues may be partially affected by droplet evaporation in the
GCVI (Bi et al., 2016). However, there was a high fraction of the amine cloud residuals
when the southwesterly air mass prevailed, as discussed in Sect. 3.4. A lack of gas-phase
amines may be the cause of few amine particles detected in the ambient particles and
cloud residues (Rehbein et al., 2011). An increase in Nf of cloud residues was observed
compared with non-activated particles (5.2% versus 0.1% by number), as shown in
Figure 7. An increase of particle water content facilitates partitioning of gas-phase amine
species into the aqueous phase when gas-phase amines present (Rehbein et al., 2011).

**4 Conclusions**

This study presented an in situ observation of individual cloud residues, non-

activated and ambient particles at a mountain site in South China. The finding shows that
the Aged EC (49.3%) and K-rich types (33.9%) dominate the cloud residues in a remote
area of China, followed by the Fe (4.1%), Amine (3.8%), Na-rich (3.0%) and Dust (2.9%)
types. The OC, Pb and Other types contributed 0.1%-2.3% by number to the total cloud
residues. The observed change in Nf of the cloud residue types, influenced by various air
masses, highlights the important role of regional transportation in the observed cloud
residual chemistry. Amine particles represented from 0.2% to 15.1% by number of the
total cloud residues dependent on the air mass history. Sulfate was found to be highly
mixed with the K-rich, OC, Aged EC, Pb, Fe and Amine cloud residues, while nitrate was
highly mixed with the Na-rich and Dust cloud residues. Compared with non-activated
particles, nitrate intensity decreased in cloud residues except dust type, and sulfate
intensity enhancement was only observed in the OC and Fe cloud residues.

**Acknowledgments**
This work was supported by the National Key Research and Development Program of
China (2017YFC0210100), the National Nature Science Foundation of China (No.
91544101 and 41405131) and the Foundation for Leading Talents of the Guangdong
Province Government. The authors thank Ji Ou from Shaoguan city Environmental
Monitoring Center for the help during the study. We also acknowledge the NOAA Air
Resources Laboratory (ARL) for the provision of the HYSPLIT transport and dispersion
model and/or READY website (http://ready.arl.noaa.gov) used in this publication. All the
data can be obtained by contacting the corresponding author.

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

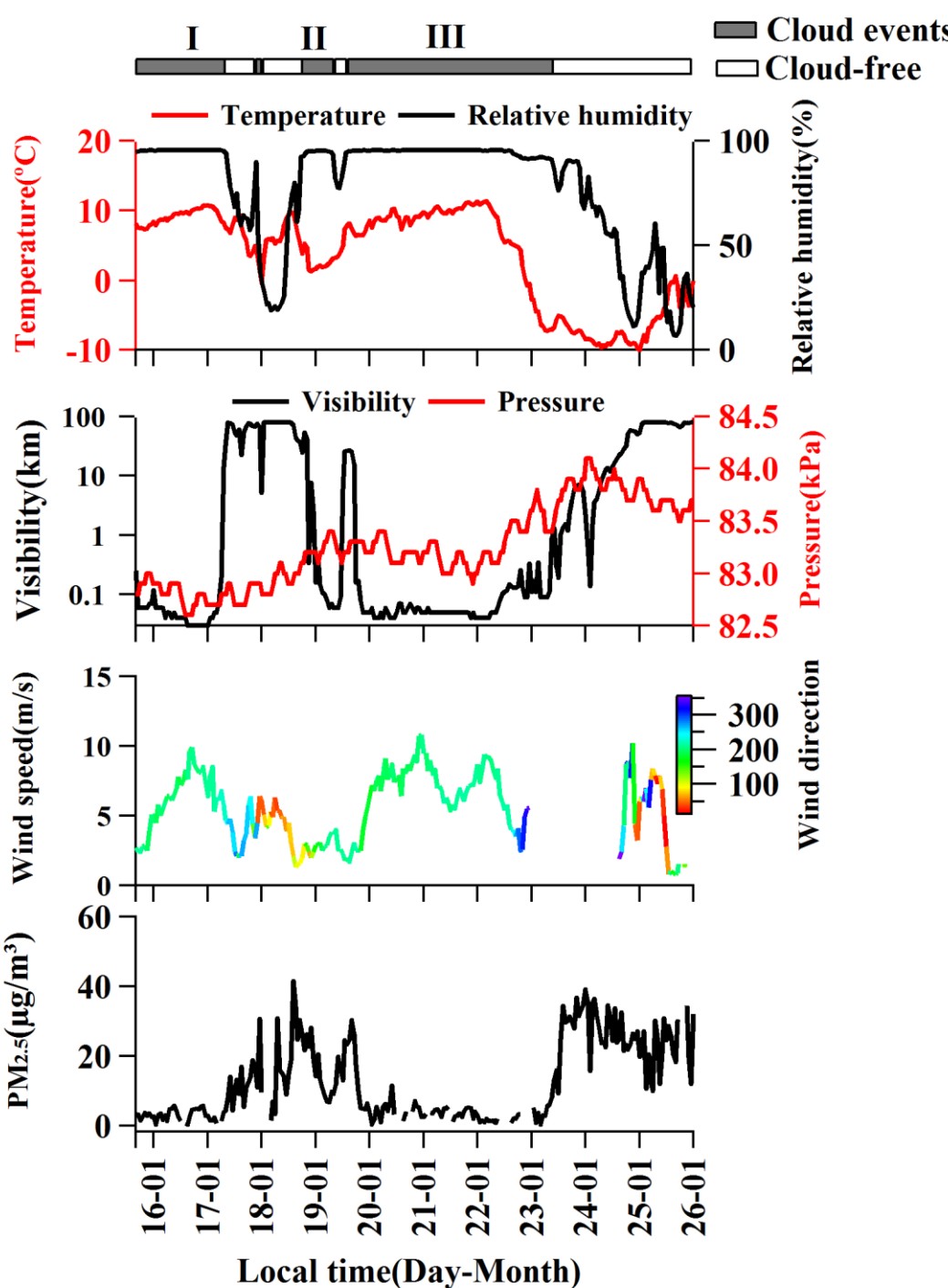


Figure 1: The hourly average variations in meteorological conditions (temperature,
relative humidity, visibility, pressure, wind speed and direction) and $PM_{2.5}$.

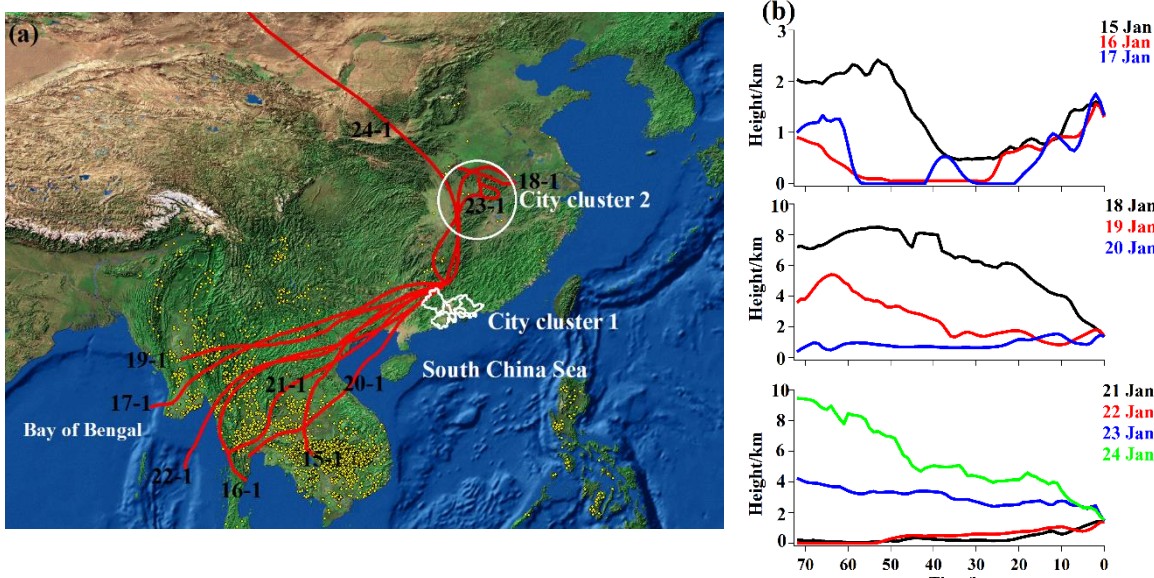

Figure 2: (a) HYSPLIT back trajectories (72 h) for air masses at 1,800 m during the whole sampling period. The white borders and circle refer to the Pearl River Delta (city cluster 1) and Yangtze River Mid-Reaches city clusters (city cluster 2), respectively. The fire date (yellow dots) are available at https://earthdata.nasa.gov/; (b) Heights (above model ground) of the air masses as a function of time.

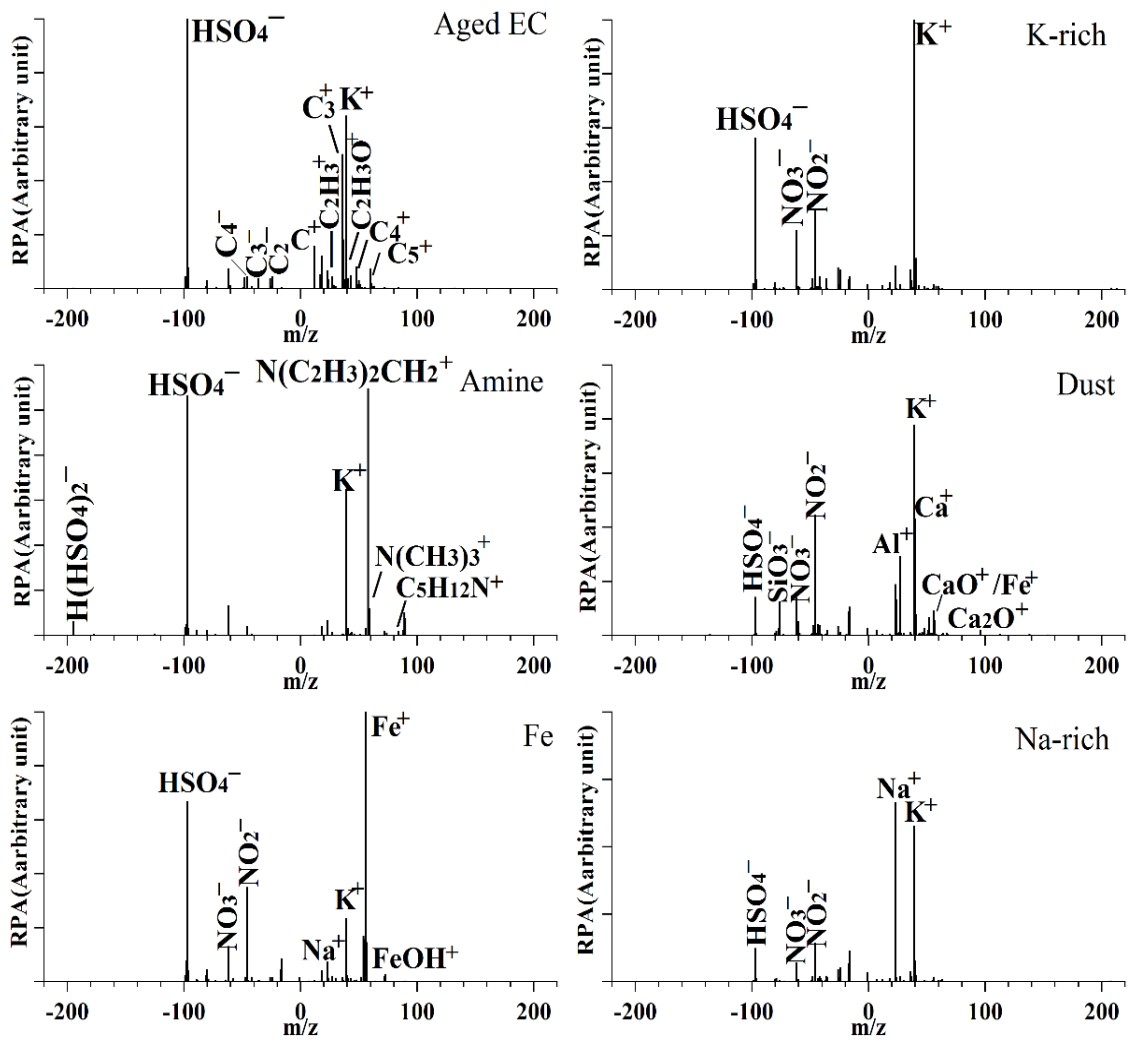

Figure 3: Averaged positive and negative mass spectra for the main 6 particle types
(Aged EC, K-rich, Amine, Dust, Fe, Na-rich) of the sampled particles. RPA in the
vertical axis refers to relative peak area. m/z in the horizontal axis represents mass-to-
charge ratio.

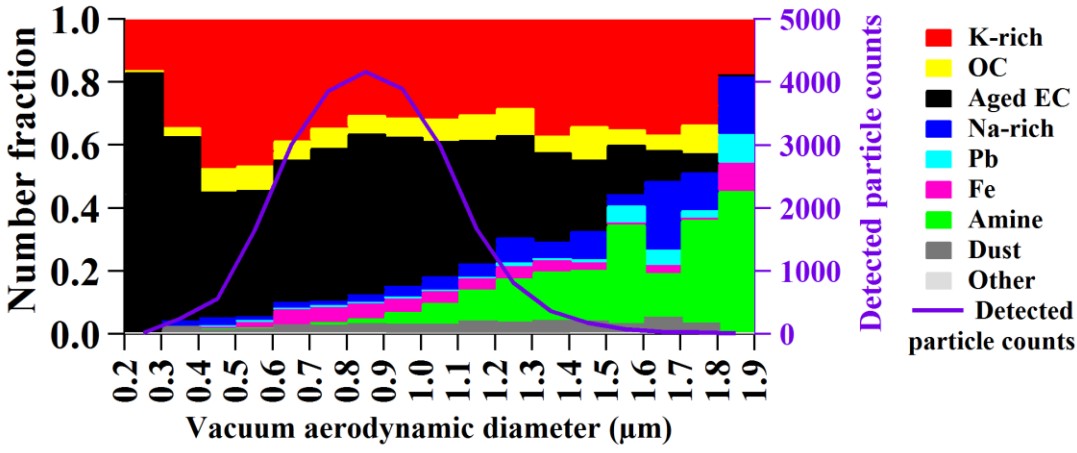


Figure 4: Number fraction for size distribution of the cloud residual types in 100 nm size
intervals.

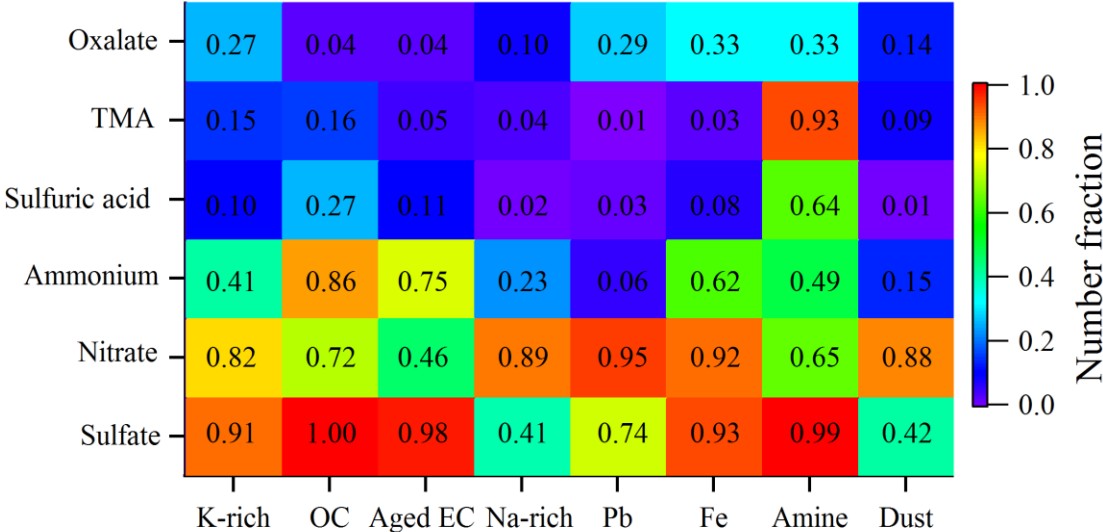


Figure 5: Number fraction of secondary markers associated with the total cloud residues
types; Sulfate (m/z, $-97HSO_4^-$), Nitrate (m/z, $-46NO_2^-$ or $-62NO_3^-$), Ammonium (m/z,
$18NH_4^+$), Sulfuric acid (m/z, $-195H(HSO_4)_2^-$), TMA (m/z, $59N(CH_3)_3^+$), Oxalate (m/z, -
$89HC_2O_4^-$).

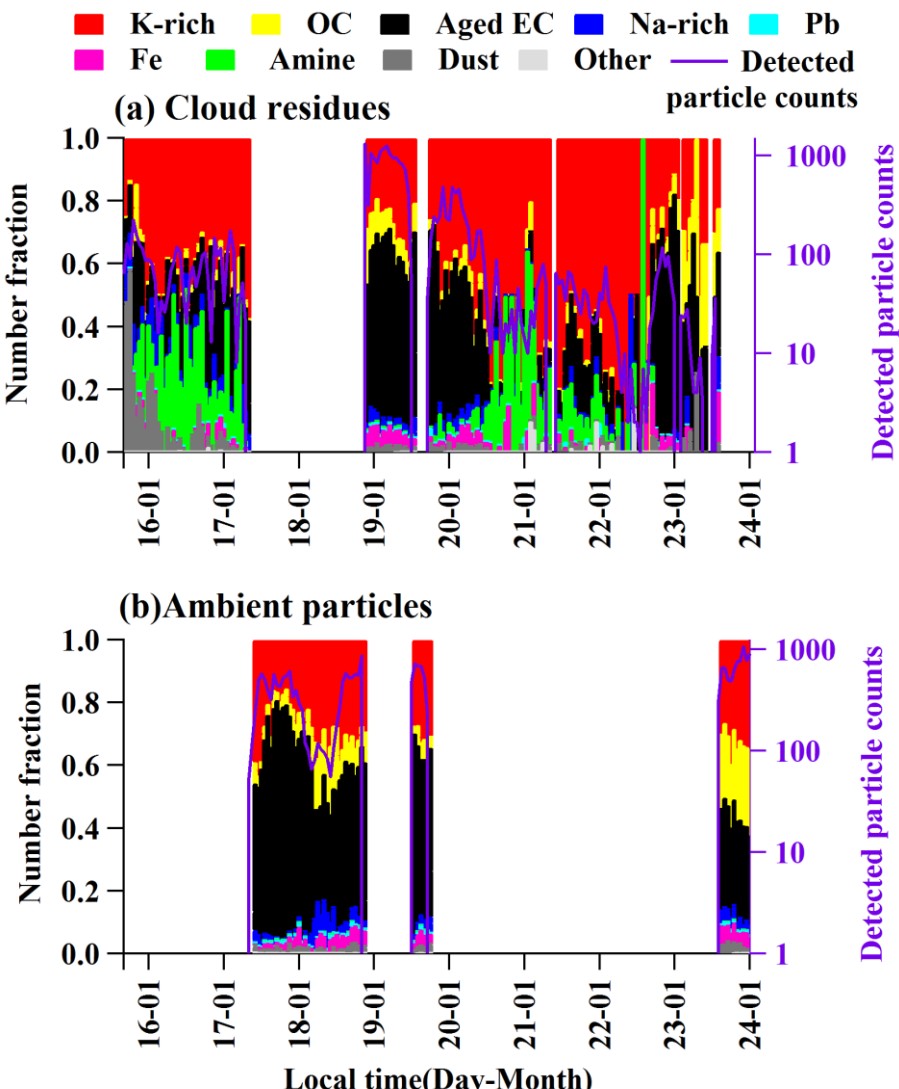

Figure 6: The hourly average variations in the cloud residual and ambient particles during
the whole sampling period.






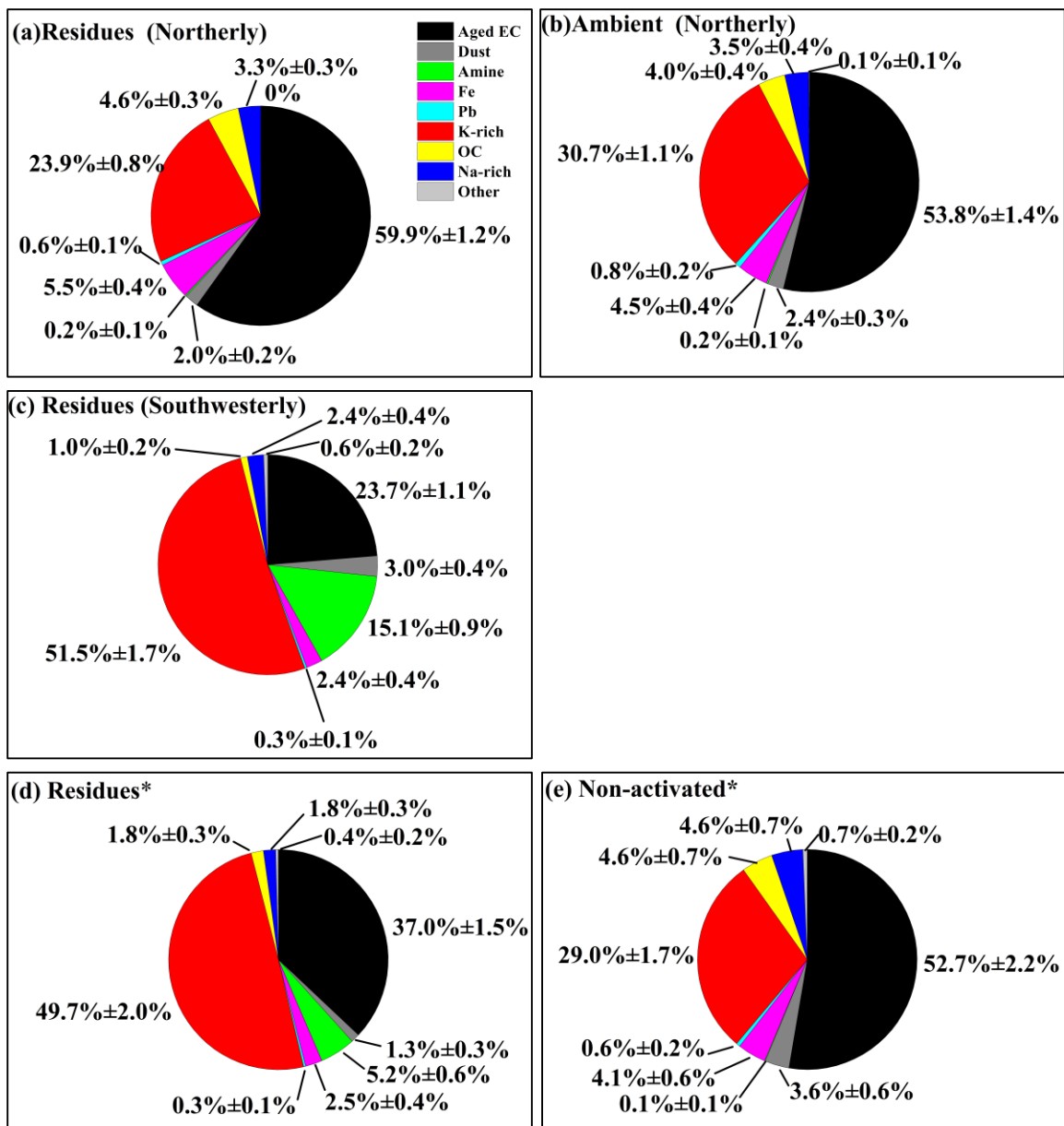


Figure 7: Number fraction of the cloud residues, ambient and non-activated particles. (a) cloud residues during northerly air mass; (b) ambient particle during northerly air mass; (c) cloud residues during southwesterly air mass; (d) cloud residues and (e) non-activated particles were alternately sampled with an interval of one hour during cloud III event. Uncertainties were calculated assuming Poisson statistics for analyzed particles.

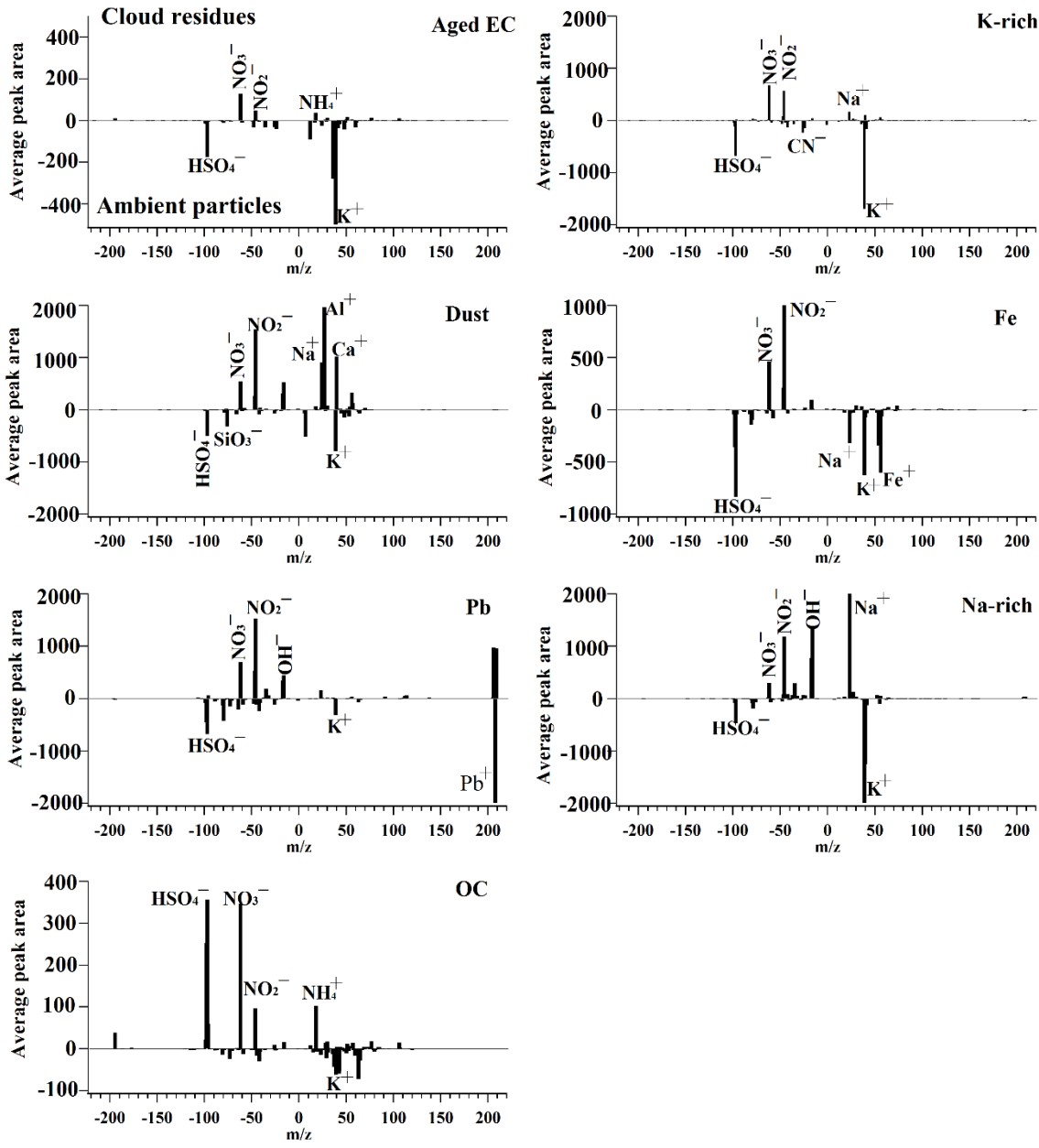



Figure 8: Mass spectral subtraction plot of the average mass spectrum corresponding to
cloud residues minus ambient particles. Positive area peaks correspond to higher
abundance in cloud residues, whereas negative area peaks show higher intensity in
ambient particles.

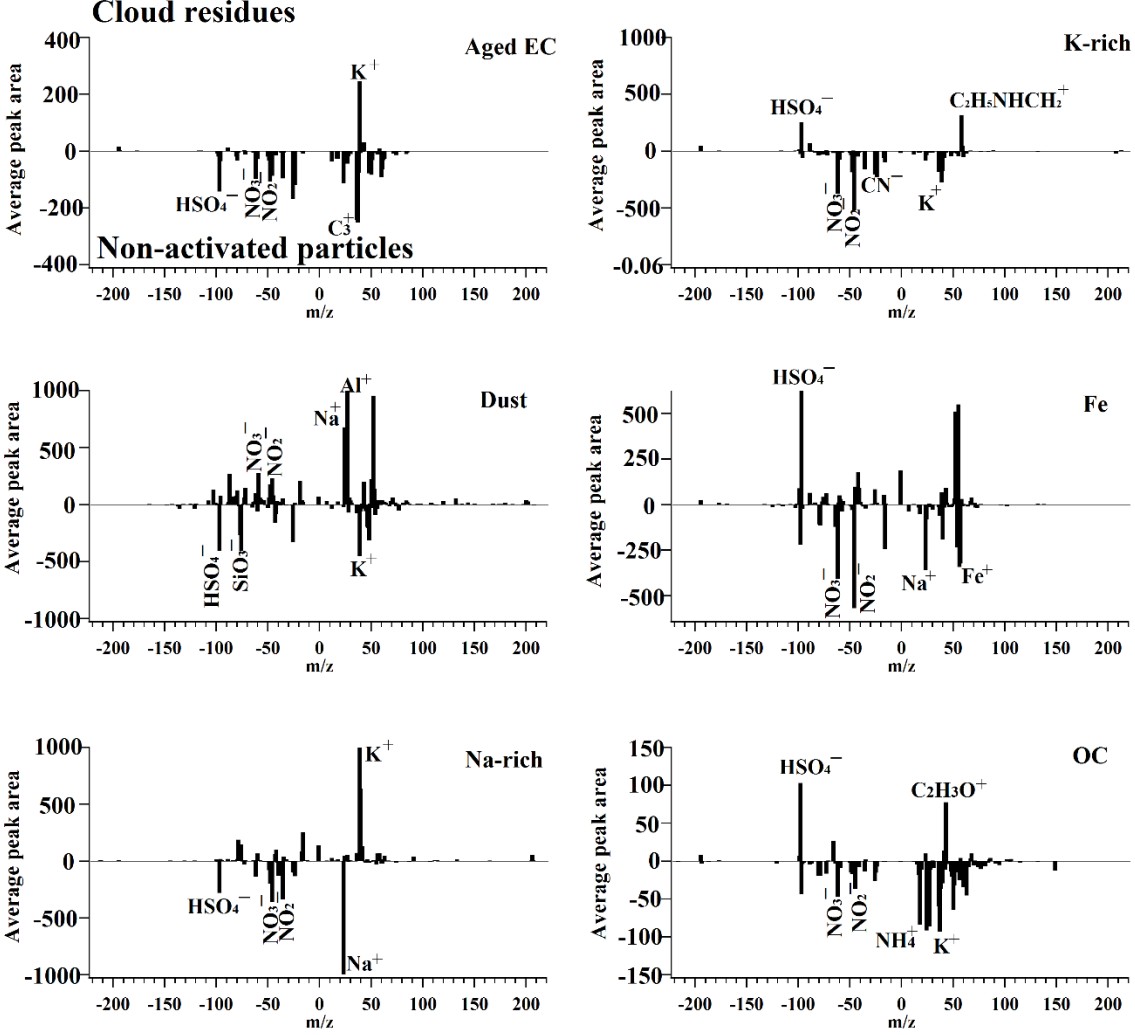

Figure 9: Mass spectral subtraction plot of the average mass spectrum corresponding to
cloud residues minus non-activated particles. Positive area peaks correspond to higher
abundance in cloud residues, whereas negative area peaks show higher intensity in non-
activated particles.