# Peer review of "In situ chemical composition measurement of individual cloud residue"

_Atmospheric Chemistry and Physics, 2017_

## Referee Comment (RC1) · Anonymous Referee #1 · 20 Mar 2017

Lin et al describe measurements of the chemistry of cloud droplet residues at a mountain-top site in South China in January 2016. Few measurements of cloud droplet residual chemistry exist, so these are important measurements to help improve our knowledge of cloud formation and properties, which are important for predicting weather and climate. Detailed comments follow.

It would be helpful for the authors to provide additional information about the cloud events. Please provide, at minimum, ambient temperature during the cloud events to justify the presence of cloud droplets only and no influence from ice crystals would be useful to discuss (related to Line 123); this is shown in Fig 1 and would be useful to refer to earlier to justify the presence of cloud droplets only. Is it correct that measurements

of cloud droplet size were not completed during this study? It would be helpful to know what fraction of the cloud droplet population was sampled, given the cut size of 8 um. It is stated on Line 135 that previous studies found an average cloud droplet size of ~10 um at this site, but the distribution is not discussed, nor is the time of year of the previous measurements. Since this work is published in Chinese, these information are not easily obtained by the reader. So, additional discussion would be helpful. For interpretation of the comparison between the cloud droplet residues and ambient particles, it is important to understand what fraction of the cloud droplets were measured. Previous studies (e.g. Bator & Collett 1997, J. Geophys. Res.) have found that cloud chemistry varies with droplet size. Sampling only the larger cloud droplets may also bias the cloud droplet residue size to larger particles, which is one of the observations. Since the cloud droplet activation process is also size-dependent, it is not possible for the reader to evaluate measurement vs droplet activation size dependencies currently.

Major Comments:

Lines 20-23, lines 220-221, Figure 4, & numerous other locations: Do these number fractions take into account the size bias in the instrument inlet transmission efficiency, which is clear in Figure S3? It is clear that there are particle size dependencies to the cloud residual chemical composition, particularly for the amine and aged EC particle types, that should be considered when reporting fractions. For example, even on lines 220-221, it is not clear if the authors are reporting 3.8% of the total cloud residues, or 3.8% of the particles measured from 0.7-1.9 um, or 3.8% across each of the size bins from 0.7-1.9 um.

Lines 64-68: Only two other cloud studies are mentioned, or referenced, here, which incorrectly suggests to the reader that the measurement of anthropogenic particles in clouds has only been measured twice. While not all papers need to necessarily be referenced here, it is important to provide a comprehensive view to the reader.

Lines 87-89: This list is not comprehensive and is missing many papers. The authors are strongly suggested to conduct a detailed literature search, as comparison of their results with these papers is important.

Line 205 & Figure 3: Provide possible ion marker formulae here. Provide a reference for the aged EC type based on comparison to other single-particle mass spectrometry studies (e.g., reference to Moffet and Prather 2009, PNAS). Given the strong K+ signal here, it is likely that both the "Aged EC" and "K-rich" are from biomass burning. Similarly, the OC particles are likely aged biomass burning particles as well, given the strong K+ signal.

Lines 209-210: The prevalence of wildfires, shown in Figure 2, suggests that these particles are primarily from biomass burning (see Pratt et al. 2011, ACP for a single-particle mass spectrum of fresh biomass burning particles). Pratt et al. (2010, J. Atmos. Sci.) and Hudson et al. (2004, J. Geophys. Res.) discuss the identification of aged biomass burning particles by single-particle mass spectrometry. It seems that the authors can say with greater certainty the source of these particles, and discussion of this would elevate the paper by providing another evaluation of the influence of biomass burning aerosols on cloud activation, which is important and interesting.

Lines 221-227: It is not clear if Roth et al. observed a decreased fraction of amines in the clouds compared to the ambient, or if amines just do not influence the site. The presence and behavior of amines would be location and season dependent, so this discussion is not clear and seems to be comparing studies without considering commonalities.

Lines 244-247: To further understand the source of the Na-rich particles, were they present for both coastal and non-coastal wind directions? Is there a difference in the average mass spectra for these wind directions considering the minor peaks? Are industrial sources located in both directions?

Lines 263-267: While ammonium nitrate does not contribute to the dust nitrate observed here, it is not appropriate to generalize this statement to suggest that previous studies in other very different locations did not measure ammonium nitrate.

Lines 319-329: Was there a mass spectral difference in the K-rich particle type between the N and SW wind directions that would aid in source identification for the two air masses?

Section 3.6: This authors should consider incorporating this discussion into the other sections of the manuscript so that comparisons are made when results are discussed. In addition, the authors should consider differences in atmospheric composition at the various sites when discussing specific comparisons (i.e. are the contributing sources and magnitudes the same, or may this be a reason for differences? Or, are the seasons the same?). Currently only a general statement on lines 431-433 states that differences are specific to geographic location. As noted above, the literature cited is also not comprehensive, and this section would benefit from additional literature searching.

Figure 4: It would be useful to add comparisons to the ambient and interstitial particles here. In addition, since the cloud residue types changed significantly based on air mass origin (N vs SW), as stated in Section 3.4, it would be useful to separate out these wind directions and show the fractions of cloud residue, interstitial, and ambient, separated for the two wind directions.

Technical Comments:

Lines 24-26, 45, & other locations (search document): Please clarify what is meant by "intensity" in these statements.

Lines 29-30: The phrase "To estimate how atmospheric aerosol particles respond to chemical properties of cloud droplets" is not clear since aerosols determine cloud droplet chemistry, outside of aqueous processing from dissolved trace gases.

Lines 41-42: For readers not familiar with the region, it would be helpful to know the suggested source of the amine particles, which the authors are presumably referring

to with respect to the wind direction change.

Line 44: What does "highly associated" mean in this context? This phrase occurs in other locations in the manuscript as well.

Lines 76-80: This sentence incorrectly cites Pratt et al 2010a for cloud droplet residues. The authors should also consider Drewnick et al 2007, J. Atmos. Chem., who did observe lower sulfate mass fractions for droplet residues compared to ambient aerosol. Pratt et al 2010 (J. Geophs. Res.) shows increased mixing with sulfate/nitrate for liquid clouds, compared to ice clouds, as another example.

Line 161: It should be clarified that "cloud droplet residue concentrations", not "cloud droplet concentrations" were measured by the SMPS. This is an important distinction.

Lines 163-165: Wouldn't hazy days with low visibility be characterized by high, rather than low, PM2.5 concentrations? This is confusing.

Line 165: Change "rainy" to "rain".

Line 216: It is important to consider the relative enhancement in amine peaks when using a 266 nm laser and that the amines themselves may not comprise the majority of the particle mass. See Pratt et al. 2009, Environ. Sci. Technol.

Lines 230-232: It should be clarified when discussing previous vs the current study, and if previous studies are being discussed, the season should be noted if there are seasonal variations.

Lines 231-237: Some grammar fixes are needed here.

Line 279: Please be more specific with the statement "plays a key role in cloud processes". Do you mean that these particles were preferentially activated? Is there evidence of this?

Lines 358-396: The phrasing in these paragraphs should be improved for greater clarity and correct grammar.

Line 390: This is an important finding, yet the phrasing "sulfate was observed to diminish" is not clear, particularly when considering the following sentence. Please clarify.

Lines 395-396: What discrepancy? This is not clear.

Line 397: Quantitatively, what is "no remarkable change"? The phrase "remarkable change" is used elsewhere in the manuscript as well, but it isn't defined.

Table 2: Not sure what the authors mean by "way" here.

Figure 2: The lines and numbers on this map are difficult to read. It would be useful to make the text bold perhaps and increase the width of the lines.

Figure 3: This plot is difficult to read. The authors should consider showing only the most abundant (and discussed) particle types in the main text figure and moving the others (including "Other", which is somewhat meaningless as an average mass spectrum if it is made up of a diverse population of particles) to the supplemental information.

Figures 7 and 8: Please indicate whether positive peak areas indicate preferentially in the cloud residues and negative indicate preferentially in the ambient/interstitial particles. This is currently not clear in the figure captions.

Figure S4: It is not clear in the maps where RH < and > 90% are located.

---

## Referee Comment (RC2) · J. Schneider (Referee) · 21 Mar 2017

In their manuscript "In situ chemical measurement of individual cloud residue particles at a mountain site, South China", Qinhao Lin and co-workers report on the analysis of single particles from cloud residues using a single particle aerosol mass spectrometer. They observed a high fraction of EC-containing particle in the residuals and detected amines with a high variability. Nitrate was found to be increased in residuals compared to ambient particles, while sulfate showed a dependency on the chemical composition of the residues. The topic of the paper is well suited for ACP, and the data itself are interesting, because single particle measurements of cloud residuals are still sparse. However, the manuscript suffers from many unclear statements and some severe uncertainties regarding the analysis of interstitial particles. I have many points where more information is needed or where I disagree. None of these points alone would be a "major" comment, but the multitude of my remarks and questions suggest to require a major revision and to reconsider the manuscript after my comments listed below have been addressed.

Comments and remarks:

Title: I suggest to change the title to "In situ chemical composition measurement of individual cloud residue particles at a mountain site, South China"

Page 4, lines 64 – 68:

More references are needed here to discuss the anthropogenic influence on cloud particles, not just two papers on single particle analysis.

Page 5, line 79:

Replace "Nf of sulfate" by "NF of sulfate-containing particles"

Page 5, line 80:

Replace "other study" by "other studies"

Page 7, line 122:

Was the humidity measured in the evaporation chamber? How do you make sure that all water evaporates?

Page 8, lines 147-149:

Did you do the size calibration on the mountain top station? What was the ambient pressure during the measurements and during the calibration? Did you check the inlet flow or the pressure inside the aerodynamic lens?

Page 8, line 161:

The SMPS does not measure the cloud droplet concentration but the cloud residue concentration. Cloud droplets would have to be measured outside in the cloud (by FSSP or similar instrumentation).

Page 9, line 171:

As I will outline in more detail below, I doubt the existence of interstitial particles in this size range.

Page 9 lines 184-185 and Figure 2:

More info on the trajectories is needed: How did the vertical evolution look like? Ho well is the mountain represented in the model? Is 1800 m the best altitude that represents the mountain site? Please add also the most important megacities to the map to help estimating the influence of anthropogenic emissions.

Page 10, line 207:

I suggest moving Figure S3 to the main paper.

Page 11, lines 221-222:

But Roth et al. found a clear enhancement of amines in residues compared to the background aerosol (9% to about 2%).

Page 12, lines 235-237:

I agree that dust is found more frequently in the coarse particle size range, but then I would expect to see an increase of the dust fraction in the residues with increasing diameter. This is not seen in Figure S3. Is the identification of dust reliable? What about the Fe-containing particles? They might be dust as well.

Page 12, lines 238-243:

What could be the source of these Pb- and Fe-containing particles? See also comment above. Can the Fe-containing particles belong to the dust-type?

Page 12, lines 244-252:

If these particles were from sea salt, they should contain chloride ions. That is hard to see in Figure 3. Are these Na-rich particles correlated with air masses coming from the ocean?

Page 13, line 259 and Figure 5:

How do you distinguish between sulfuric acid and sulfate? Besides, spelling (sulfate, sulphuric acid) should be consistent ("f" or "ph").

Page 13, lines 265-267:

What other forms of nitrate do you suggest to be present on the Na-rich and dust residues? What about uptake of nitric acid from the gas-phase by the cloud droplets?

How certain is the identification of ammonium? Which peak was used? Page 13, lines 268-272:

The stability of ammonium nitrate depends also on the humidity. In the book by Seinfeld and Pandis (2nd edition, Wiley and Sons, 2006, Chapter 10.4.3) it is shown that at 30% RH ammonium nitrate does not exist above 30°C. I would assume that the dry carrier gas in the evaporation section is below 30%RH. Thus, it may well be that NH4NO3 evaporates in your system.

Page 13, lines 275-276:

The sentences "The presence of abundant sulfate in aged EC cloud residues was considered to be a good CCN species before activation. . ." needs rephrasing. It is not clear to me what you want to say. Do you mean "aged EC particles mixed with sulfate are good CCN"?

Page 13, line 279:

Ammonium will most likely play a key role in the form of ammonium sulfate or ammonium nitrate. In organic particles, amines may play that role (methylamines). Again: how do you identify ammonium and how do you distinguish between amines and ammonium?

Page 14, line 281 (and Figure 5):

Why does oxalate nor correlate with OC?

Page 14, lines 284-285:

What do you mean by "enrichment of TMA in amine cloud residuals"? You observe that in 93% of those cloud residuals that are assigned to the "amine" type contain TMA. That is not surprising, more surprising is that it's not 100%. But that's inside the measurement uncertainties, to my opinion.

Page 14, lines 294-295:

"This may result in 33% by number to the Amine residues containing oxalate." Please rephrase, not clear what you want to say.

Page 14, line 298:

What does "unscaled" mean? These are absolute particle numbers.

Page 14, lines 302-303:

You say that the air masses change from northerly on 18 Jan to southwesterly on 19 Jan, but the particles remain similar from 17 Jan (around noon) to 20 Jan (noon). On the other hand, the change in particle types is very abrupt from cloud residuals to ambient on Jan 17.

Page 15, lines 322-325:

Do verify the possible transport of biomass burning particles to the site, the vertical history of the trajectories is required.

Page 16, lines 337-229:

"Note that after the activation of amine particles, the partitioning of the gas amine on cloud droplets may further contribute to the enhanced Amine cloud residues". That is true, but holds also for other species, as nitrate (HNO3) or water-soluble OC.

Section 3.5

Here I have a major concern: You report interstitial particles containing sulfate and nitrate in the size range between 200 and 1300 nm. It is very hard to believe (not to say impossible) that such large particles are not activated in a cloud.

Later (page 17, lines 366-368) you write " However, few studies have focused on this issue, in part because interstitial particles show a smaller size than that detected by single-particle mass spectrometry (Roth et al., 2016)." Since the SPAMS has a very similar lower detection size range as the ALABAMA used by my group in Roth et al., 2016), you can not expect that you detect non-activated interstitial particles which should be in the size range below 200 nm.

My suspicion is: The clouds became thinner, and entrainment of cloud-free air has mixed "normal" aerosol particles into the cloud. But such particles cannot be referred to as "interstitial". As long as you don't have cloud microphysics (number and size of cloud droplets) or alt leas liquid water content (Particle Volume Monitor) available, I would suggest to remove this chapter on interstitial particles.

Page 17, line 358 / Table 1:

I would prefer a graph with bars or pie charts. I also strongly recommend showing the SMPS size distributions from residues, ambient and interstitial. That might help to identify the issues with the large interstitial particles.

Page 18, lines 374-375 / Figure 7 & 8:

How are the difference mass spectra of Figure 7 and 8 calculated? Is it ambient - residues and interstitial – residues? Or vice versa? How were the spectra normalized? Please explain.

Page 18, lines 376-382:

Why not? I drew the same conclusion as Hayden et al. (2008) in my 2017 paper (Schneider et al., 2017, please note the update from ACPD 2016 to ACP 2017). HNO3 uptake may not be the source of the particles but explains the high amount of nitrate found on many particles, also on the Na-rich and dust particles discussed above.

Page 18, lines 384-386:

I agree with that, but wouldn't that support the idea of uptake of HNO3 from the gas phase? If the nitrate content does not play the major role in the activation, but more nitrate is found in the residues, that's an argument for HNO3 uptake.

Page 18, lines 387-388:

Can intensity simply be compared like this? What about size effects and matrix effects? But again, an explanation how Figures 7 and 8 were calculated might help here.

Page 18, lines 391-392:

"Compared with interstitial particles, sulfate enhanced in the Fe cloud residues." I think an "is" is missing here.

Page 19, lines 398-399:

Better: "The in-cloud process has been reported to be an important pathway..."

Page 20, lines 421-422:

The Jungfraujoch is a station located mostly in the free troposphere and in a remote region, so the biomass burning contribution can be expected to be lower than at other sites.

Figures

Figure 3: Please improve resolution. Labels can't be read upon zooming in.

Figure 6: The ambient particle time series (b) are broader than the corresponding gaps in (a). Please make the Figure broader. You can move the legend with the particle types to above or below the graphs, plus the legend is only needed once.

References:

Roth, A., Schneider, J., Klimach, T., Mertes, S., van Pinxteren, D., Herrmann, H., and Borrmann, S.: Aerosol properties, source identification, and cloud processing in orographic clouds measured by single particle mass spectrometry on a central European mountain site during HCCT-2010, Atmos. Chem. Phys., 16, 505-524, doi:10.5194/acp-16-505-2016, 2016.

Schneider, J., Mertes, S., van Pinxteren, D., Herrmann, H., and Borrmann, S.: Uptake of nitric acid, ammonia, and organics in orographic clouds: mass spectrometric analyses of droplet residual and interstitial aerosol particles, Atmos. Chem. Phys., 17, 1571-1593, doi:10.5194/acp-17-1571-2017, 2017.

Seinfeld, J. H., and Pandis, S. N.: Atmospheric chemistry and physics: from air pollution to climate change John Wiley and Sons, Hoboken, NJ, 2006.
* * *

---

## Author Comment (AC2) · 15 May 2017

| 1  | Response to comments                                                                   |
|----|-----------------------------------------------------------------------------------------------|
| 2  |                                                                                               |
| 3  | Manuscript Number: acp-2017-23                                                                |
| 4  | Title: In situ chemical measurement of individual cloud residue particles at a mountain site, |
| 5  | South China. Qinhao Lin et al.                                                                |
| 6  | Received and published: 20 March 2017                                                         |
| 7  |                                                                                               |
| 8  | Referee #2: J. Schneider                                                                      |
| 9  |                                                                                               |
| 10 | In their manuscript "In situ chemical measurement of individual cloud residue particles at    |
| 11 | a mountain site, South China", Qinhao Lin and co-workers report on the analysis of single     |
| 12 | particles from cloud residues using a single particle aerosol mass spectrometer. They         |
| 13 | observed a high fraction of EC-containing particle in the residuals and detected amines       |
| 14 | with a high variability. Nitrate was found to be increased in residuals compared to ambient   |
| 15 | particles, while sulfate showed a dependency on the chemical composition of the residues.     |
| 16 | The topic of the paper is well suited for ACP, and the data itself are interesting, because   |
| 17 | single particle measurements of cloud residuals are still sparse. However, the manuscript     |
| 18 | suffers from many unclear statements and some severe uncertainties regarding the analysis     |
| 19 | of interstitial particles. I have many points where more information is needed or where I     |
| 20 | disagree. None of these points alone would be a "major" comment, but the multitude of my      |
| 21 | remarks and questions suggest to require a major revision and to reconsider the manuscript    |
| 22 | after my comments listed below have been addressed.                                           |
| 23 |                                                                                               |
| 24 | We would like to thank Prof. J. Schneider for his useful comments and recommendations         |
| 25 | to improve the manuscript. We agree with the comments, and careful revision has been          |
| 26 | made accordingly, please refer to the following responses for details.                        |
| 27 |                                                                                               |
| 28 | Comments and remarks:                                                                         |
| 29 | Title: I suggest to change the title to "In situ chemical composition measurement of          |
| 30 | individual cloud residue particles at a mountain site, South China"                           |
| 31 |                                                                                               |
| 32 | We have changed accordingly. Please refer to Lines 1-2 of the revised manuscript.             |
| 33 |                                                                                               |
| 34 | Page 4, lines 64 – 68:                                                                        |
| 35 | More references are needed here to discuss the anthropogenic influence on cloud particles,    |
| 36 | not just two papers on single particle analysis.                                              |
|    |                                                                                               |

|  <li>As also suggested by Reviewer 1, We have added references (Stier et al., 2005; Sorooshia et al., 2007b; Lohmann et al., 2007; Rosenfeld et al., 2008; Roth et al., 2016; Seinfeld, al., 2016; Li et al., 2017) to discuss the anthropogenic influence on cloud particles</li> <li>Anthropogenic particles can increase number concentration of small cloud droplets, in turn affect reflectivity and life time of clouds (Rosenfeld et al., 2008; Stier et al., 2009; Lohmann et al., 2007). In-situ cloud chemical measurements show varied chemicat composition of cloud droplets at various regions (Sorooshian et al., 2007a; Roth et al., 201</li> <li>Li et al., 2017). Although a large number of aerosol/cloud studies over the past 20 year the uncertainty for evaluating radiative forcing due to aerosol-cloud interactions has not been reduced (Seinfeld, et al., 2016). Please refer to Lines 55-62 of the revised manuscript</li> <li>Page 5, line 79: Replace "Nf of sulfate" by "NF of sulfate-containing particles"</li> <li>We have changed it accordingly. Please refer to Line 74 of the revised manuscript.</li> <li>Page 5, line 80: Replace "other study" by "other studies"</li>  | 37 |                                                                                                |
|---------------------------------------------------------------------------------------------------------------------------------------------------------------------------------------------------------------------------------------------------------------------------------------------------------------------------------------------------------------------------------------------------------------------------------------------------------------------------------------------------------------------------------------------------------------------------------------------------------------------------------------------------------------------------------------------------------------------------------------------------------------------------------------------------------------------------------------------------------------------------------------------------------------------------------------------------------------------------------------------------------------------------------------------------------------------------------------------------------------------------------------------------------------------------------------------------------------------------|----|------------------------------------------------------------------------------------------------|
|  <li>et al., 2007b; Lohmann et al., 2007; Rosenfeld et al., 2008; Roth et al., 2016; Seinfeld. et al., 2016; Li et al., 2017) to discuss the anthropogenic influence on cloud particles</li> <li>Anthropogenic particles can increase number concentration of small cloud droplets, in turn affect reflectivity and life time of clouds (Rosenfeld et al., 2008; Stier et al., 2007)</li> <li>Lohmann et al., 2007). In-situ cloud chemical measurements show varied chemicat composition of cloud droplets at various regions (Sorooshian et al., 2007a; Roth et al., 201</li> <li>Li et al., 2017). Although a large number of aerosol/cloud studies over the past 20 year the uncertainty for evaluating radiative forcing due to aerosol-cloud interactions has not been reduced (Seinfeld. et al., 2016). Please refer to Lines 55-62 of the revised manuscript</li> <li>Page 5, line 79: Replace "Nf of sulfate" by "NF of sulfate-containing particles"</li> <li>We have changed it accordingly. Please refer to Line 74 of the revised manuscript.</li> <li>Page 5, line 80: Replace "other study" by "other studies"</li>                                                                               | 38 | As also suggested by Reviewer 1, We have added references (Stier et al., 2005; Sorooshian      |
|  <li>al., 2016; Li et al., 2017) to discuss the anthropogenic influence on cloud particle.</li> <li>Anthropogenic particles can increase number concentration of small cloud droplets, in turn</li> <li>affect reflectivity and life time of clouds (Rosenfeld et al., 2008; Stier et al., 2009)</li> <li>Lohmann et al., 2007). In-situ cloud chemical measurements show varied chemical</li> <li>composition of cloud droplets at various regions (Sorooshian et al., 2007a; Roth et al., 2011)</li> <li>Li et al., 2017). Although a large number of aerosol/cloud studies over the past 20 year</li> <li>the uncertainty for evaluating radiative forcing due to aerosol-cloud interactions has not</li> <li>been reduced (Seinfeld. et al., 2016). Please refer to Lines 55-62 of the revised manuscript</li> <li>Page 5, line 79: Replace "Nf of sulfate" by "NF of sulfate-containing particles"</li> <li>We have changed it accordingly. Please refer to Line 74 of the revised manuscript.</li> <li>Page 5, line 80: Replace "other study" by "other studies"</li>                                                                                                                                      | 39 | et al., 2007b; Lohmann et al., 2007; Rosenfeld et al., 2008; Roth et al., 2016; Seinfeld. et   |
|  <li>Anthropogenic particles can increase number concentration of small cloud droplets, in turn
affect reflectivity and life time of clouds (Rosenfeld et al., 2008; Stier et al., 2009)</li> <li>Lohmann et al., 2007). In-situ cloud chemical measurements show varied chemical
composition of cloud droplets at various regions (Sorooshian et al., 2007a; Roth et al., 2011)</li> <li>Li et al., 2017). Although a large number of aerosol/cloud studies over the past 20 year
the uncertainty for evaluating radiative forcing due to aerosol-cloud interactions has not
been reduced (Seinfeld. et al., 2016). Please refer to Lines 55-62 of the revised manuscrip</li> <li>Page 5, line 79: Replace "Nf of sulfate" by "NF of sulfate-containing particles"</li> <li>We have changed it accordingly. Please refer to Line 74 of the revised manuscript.</li> <li>Page 5, line 80: Replace "other study" by "other studies"</li>                                                                                                                                                                                                                                            | 40 | al., 2016; Li et al., 2017) to discuss the anthropogenic influence on cloud particles.         |
|  <li>affect reflectivity and life time of clouds (Rosenfeld et al., 2008; Stier et al., 2005)</li> <li>Lohmann et al., 2007). In-situ cloud chemical measurements show varied chemical</li> <li>composition of cloud droplets at various regions (Sorooshian et al., 2007a; Roth et al., 2014)</li> <li>Li et al., 2017). Although a large number of aerosol/cloud studies over the past 20 year</li> <li>the uncertainty for evaluating radiative forcing due to aerosol-cloud interactions has not</li> <li>been reduced (Seinfeld. et al., 2016). Please refer to Lines 55-62 of the revised manuscrip</li> <li>Page 5, line 79: Replace "Nf of sulfate" by "NF of sulfate-containing particles"</li> <li>We have changed it accordingly. Please refer to Line 74 of the revised manuscript.</li> <li>Page 5, line 80: Replace "other study" by "other studies"</li>                                                                                                                                                                                                                                                                                                                            | 41 | Anthropogenic particles can increase number concentration of small cloud droplets, in turn,    |
|  <li>Lohmann et al., 2007). In-situ cloud chemical measurements show varied chemical composition of cloud droplets at various regions (Sorooshian et al., 2007a; Roth et al., 2014). Li et al., 2017). Although a large number of aerosol/cloud studies over the past 20 years the uncertainty for evaluating radiative forcing due to aerosol-cloud interactions has not been reduced (Seinfeld. et al., 2016). Please refer to Lines 55-62 of the revised manuscript</li> <li>Page 5, line 79: Replace "Nf of sulfate" by "NF of sulfate-containing particles"</li> <li>We have changed it accordingly. Please refer to Line 74 of the revised manuscript.</li> <li>Page 5, line 80: Replace "other study" by "other studies"</li>                                                                                                                                                                                                                                                                                                                                                                                                                                                               | 42 | affect reflectivity and life time of clouds (Rosenfeld et al., 2008; Stier et al., 2005;       |
|  <li>composition of cloud droplets at various regions (Sorooshian et al., 2007a; Roth et al., 201</li> <li>Li et al., 2017). Although a large number of aerosol/cloud studies over the past 20 year</li> <li>the uncertainty for evaluating radiative forcing due to aerosol-cloud interactions has no</li> <li>been reduced (Seinfeld. et al., 2016). Please refer to Lines 55-62 of the revised manuscrip</li> <li>Page 5, line 79: Replace "Nf of sulfate" by "NF of sulfate-containing particles"</li> <li>We have changed it accordingly. Please refer to Line 74 of the revised manuscript.</li> <li>Page 5, line 80: Replace "other study" by "other studies"</li>                                                                                                                                                                                                                                                                                                                                                                                                                                                                                                                          | 43 | Lohmann et al., 2007). In-situ cloud chemical measurements show varied chemical                |
|  <li>Li et al., 2017). Although a large number of aerosol/cloud studies over the past 20 year</li> <li>the uncertainty for evaluating radiative forcing due to aerosol-cloud interactions has no</li> <li>been reduced (Seinfeld. et al., 2016). Please refer to Lines 55-62 of the revised manuscrip</li> <li>Page 5, line 79: Replace "Nf of sulfate" by "NF of sulfate-containing particles"</li> <li>We have changed it accordingly. Please refer to Line 74 of the revised manuscript.</li> <li>Page 5, line 80: Replace "other study" by "other studies"</li>                                                                                                                                                                                                                                                                                                                                                                                                                                                                                                                                                                                                                                | 44 | composition of cloud droplets at various regions (Sorooshian et al., 2007a; Roth et al., 2016; |
|  <li>the uncertainty for evaluating radiative forcing due to aerosol-cloud interactions has no been reduced (Seinfeld. et al., 2016). Please refer to Lines 55-62 of the revised manuscrip</li> <li>Page 5, line 79: Replace "Nf of sulfate" by "NF of sulfate-containing particles"</li> <li>We have changed it accordingly. Please refer to Line 74 of the revised manuscript.</li> <li>Page 5, line 80: Replace "other study" by "other studies"</li>                                                                                                                                                                                                                                                                                                                                                                                                                                                                                                                                                                                                                                                                                                                                           | 45 | Li et al., 2017). Although a large number of aerosol/cloud studies over the past 20 years,     |
|  <li>been reduced (Seinfeld. et al., 2016). Please refer to Lines 55-62 of the revised manuscrip</li> <li>Page 5, line 79: Replace "Nf of sulfate" by "NF of sulfate-containing particles"</li> <li>We have changed it accordingly. Please refer to Line 74 of the revised manuscript.</li> <li>Page 5, line 80: Replace "other study" by "other studies"</li>                                                                                                                                                                                                                                                                                                                                                                                                                                                                                                                                                                                                                                                                                                                                                                                                                                     | 46 | the uncertainty for evaluating radiative forcing due to aerosol-cloud interactions has not     |
|  <li>48</li> <li>49</li> <li>50 Page 5, line 79: Replace "Nf of sulfate" by "NF of sulfate-containing particles"</li> <li>51</li> <li>52 We have changed it accordingly. Please refer to Line 74 of the revised manuscript.</li> <li>53</li> <li>54 Page 5, line 80: Replace "other study" by "other studies"</li>                                                                                                                                                                                                                                                                                                                                                                                                                                                                                                                                                                                                                                                                                                                                                                                                                                                                                               | 47 | been reduced (Seinfeld. et al., 2016). Please refer to Lines 55-62 of the revised manuscript.  |
|  <li>49</li> <li>50 Page 5, line 79: Replace "Nf of sulfate" by "NF of sulfate-containing particles"</li> <li>51</li> <li>52 We have changed it accordingly. Please refer to Line 74 of the revised manuscript.</li> <li>53</li> <li>54 Page 5, line 80: Replace "other study" by "other studies"</li>                                                                                                                                                                                                                                                                                                                                                                                                                                                                                                                                                                                                                                                                                                                                                                                                                                                                                                           | 48 |                                                                                                |
|  <li>Page 5, line 79: Replace "Nf of sulfate" by "NF of sulfate-containing particles"</li> <li>We have changed it accordingly. Please refer to Line 74 of the revised manuscript.</li> <li>Page 5, line 80: Replace "other study" by "other studies"</li>                                                                                                                                                                                                                                                                                                                                                                                                                                                                                                                                                                                                                                                                                                                                                                                                                                                                                                                                                        | 49 |                                                                                                |
|  <li>51</li> <li>52 We have changed it accordingly. Please refer to Line 74 of the revised manuscript.</li> <li>53</li> <li>54 Page 5, line 80: Replace "other study" by "other studies"</li>                                                                                                                                                                                                                                                                                                                                                                                                                                                                                                                                                                                                                                                                                                                                                                                                                                                                                                                                                                                                                    | 50 | Page 5, line 79: Replace "Nf of sulfate" by "NF of sulfate-containing particles"               |
|  <li>We have changed it accordingly. Please refer to Line 74 of the revised manuscript.</li> <li>Page 5, line 80: Replace "other study" by "other studies"</li>                                                                                                                                                                                                                                                                                                                                                                                                                                                                                                                                                                                                                                                                                                                                                                                                                                                                                                                                                                                                                                                  | 51 |                                                                                                |
|  <li>53</li> <li>54 Page 5, line 80: Replace "other study" by "other studies"</li>                                                                                                                                                                                                                                                                                                                                                                                                                                                                                                                                                                                                                                                                                                                                                                                                                                                                                                                                                                                                                                                                                                                               | 52 | We have changed it accordingly. Please refer to Line 74 of the revised manuscript.             |
| 54 Page 5, line 80: Replace "other study" by "other studies"                                                                                                                                                                                                                                                                                                                                                                                                                                                                                                                                                                                                                                                                                                                                                                                                                                                                                                                                                                                                                                                                                                                                                              | 53 |                                                                                                |
|                                                                                                                                                                                                                                                                                                                                                                                                                                                                                                                                                                                                                                                                                                                                                                                                                                                                                                                                                                                                                                                                                                                                                                                                                           | 54 | Page 5, line 80: Replace "other study" by "other studies"                                      |
| 55                                                                                                                                                                                                                                                                                                                                                                                                                                                                                                                                                                                                                                                                                                                                                                                                                                                                                                                                                                                                                                                                                                                                                                                                                        | 55 |                                                                                                |
| 56 We have changed it accordingly. Please refer to Line 76 of the revised manuscript.                                                                                                                                                                                                                                                                                                                                                                                                                                                                                                                                                                                                                                                                                                                                                                                                                                                                                                                                                                                                                                                                                                                                     | 56 | We have changed it accordingly. Please refer to Line 76 of the revised manuscript.             |
| 57                                                                                                                                                                                                                                                                                                                                                                                                                                                                                                                                                                                                                                                                                                                                                                                                                                                                                                                                                                                                                                                                                                                                                                                                                        | 57 |                                                                                                |
| 58 Page 7, line 122: Was the humidity measured in the evaporation chamber? How do yo                                                                                                                                                                                                                                                                                                                                                                                                                                                                                                                                                                                                                                                                                                                                                                                                                                                                                                                                                                                                                                                                                                                                      | 58 | Page 7, line 122: Was the humidity measured in the evaporation chamber? How do you             |
| 59 make sure that all water evaporates?                                                                                                                                                                                                                                                                                                                                                                                                                                                                                                                                                                                                                                                                                                                                                                                                                                                                                                                                                                                                                                                                                                                                                                                   | 59 | make sure that all water evaporates?                                                           |
| 60                                                                                                                                                                                                                                                                                                                                                                                                                                                                                                                                                                                                                                                                                                                                                                                                                                                                                                                                                                                                                                                                                                                                                                                                                        | 60 | Deleting brandities (DID) many arrest d 200% in the arrest in a branches there it are be       |
| 61 Relative humidity (RH) was around 30% in the evaporation chamber, thus it can b                                                                                                                                                                                                                                                                                                                                                                                                                                                                                                                                                                                                                                                                                                                                                                                                                                                                                                                                                                                                                                                                                                                                        | 61 | Relative numidity (RH) was around 30% in the evaporation chamber, thus it can be               |
| assumed that the majority of water was evaporated. Please refer to Lines 151-154 of the                                                                                                                                                                                                                                                                                                                                                                                                                                                                                                                                                                                                                                                                                                                                                                                                                                                                                                                                                                                                                                                                                                                                   | 62 | assumed that the majority of water was evaporated. Please refer to Lines 151-154 of the        |
| 63 revised manuscript.                                                                                                                                                                                                                                                                                                                                                                                                                                                                                                                                                                                                                                                                                                                                                                                                                                                                                                                                                                                                                                                                                                                                                                                                    | 63 | Tevised manuscript.                                                                            |
| 5 Page 8 lines 147-140. Did you do the size calibration on the mountain top station? Whe                                                                                                                                                                                                                                                                                                                                                                                                                                                                                                                                                                                                                                                                                                                                                                                                                                                                                                                                                                                                                                                                                                                                  | 65 | Page 8 lines 147-140. Did you do the size calibration on the mountain top station? What        |
| 55 Tage 6, thes 147-147. Dia you do the size cultoration on the mountain top station: whe
56 was the ambient pressure during the measurements and during the calibration? Did you                                                                                                                                                                                                                                                                                                                                                                                                                                                                                                                                                                                                                                                                                                                                                                                                                                                                                                                                                                                                                                      | 66 | was the ambient pressure during the measurements and during the calibration? Did you           |
| 67 check the inlet flow or the pressure inside the aerodynamic lens?                                                                                                                                                                                                                                                                                                                                                                                                                                                                                                                                                                                                                                                                                                                                                                                                                                                                                                                                                                                                                                                                                                                                                      | 67 | check the inlet flow or the pressure inside the aerodynamic lens?                              |
| 68                                                                                                                                                                                                                                                                                                                                                                                                                                                                                                                                                                                                                                                                                                                                                                                                                                                                                                                                                                                                                                                                                                                                                                                                                        | 68 | encer the interfiew of the pressure instac the acroaynamic tens.                               |
| 69 We did the size calibration on the mountain top station. Polystyrene latex sphere                                                                                                                                                                                                                                                                                                                                                                                                                                                                                                                                                                                                                                                                                                                                                                                                                                                                                                                                                                                                                                                                                                                                      | 69 | We did the size calibration on the mountain top station. Polystyrene latex spheres             |
| 70 (Nanosphere Size Standards, Duke Scientific Corp., Palo Alto) of 0.2-2.0 um in diameter                                                                                                                                                                                                                                                                                                                                                                                                                                                                                                                                                                                                                                                                                                                                                                                                                                                                                                                                                                                                                                                                                                                                | 70 | (Nanosphere Size Standards, Duke Scientific Corp., Palo Alto) of 0.2-2.0 um in diameter        |
| 71 were used to calibrate the sizes of the detected particles on the mountain top station. Th                                                                                                                                                                                                                                                                                                                                                                                                                                                                                                                                                                                                                                                                                                                                                                                                                                                                                                                                                                                                                                                                                                                             | 71 | were used to calibrate the sizes of the detected particles on the mountain top station. The    |
| ambient pressure was 830 hPa (826-842 hPa) during the measurements and during th                                                                                                                                                                                                                                                                                                                                                                                                                                                                                                                                                                                                                                                                                                                                                                                                                                                                                                                                                                                                                                                                                                                                          | 72 | ambient pressure was 830 hPa (826-842 hPa) during the measurements and during the              |

calibration. The pressure inside the aerodynamic lens maintains about 3 hPa during the
measurements and during the calibration. Please refer to Lines 152-155 of the revised
manuscript.

76

Page 8, line 161: The SMPS does not measure the cloud droplet concentration but the
cloud residue concentration. Cloud droplets would have to be measured outside in the
cloud (by FSSP or similar instrumentation).

80

We have corrected the mistake. "cloud droplet concentration" was replaced with "cloud
residual concentration". Please refer to Lines 165-168 of the revised manuscript.

83

Page 9, line 171: As I will outline in more detail below, I doubt the existence of interstitial
particles in this size range.

86

The period of collecting interstitial particles on 22-23 Jan encountered initial mixing of
northerly cloud-free air (dry and cold airstreams) and southwesterly cloudy air (moist
airflows). The dry northern air mass might lower supersaturation, only larger particles
could be activated. This might result in non-activated particles observed to be above 200
nm here (Mertes et al., 2005; Kleinman et al., 2012; Hammer et al., 2014). To make it more
accurate, we prefer to name "non-activated particles", rather than "interstitial particles".
We have clarified them. Please refer to Lines 452-461 of the revised manuscript.

94

Page 9 lines 184-185 and Figure 2: More info on the trajectories is needed: How did the
vertical evolution look like? How well is the mountain represented in the model? Is 1800
m the best altitude that represents the mountain site? Please add also the most important
megacities to the map to help estimating the influence of anthropogenic emissions.

99

100 We have added the vertical evolution of the trajectories. The beginning of southwesterly 101 air masses traversed at lower heights relative to northerly air masses. Please refer to the Figure 2 (b). Heights of the HYSPLIT model in the study region (a spatial resolution of 102  $0.5^{\circ} \times 0.5^{\circ}$ ) was averaged 500 m a.s.l, which was lower than height of the observed site 103 104 (1,690 m a.s.l). Therefore, a height of 1,800 m a.s.l. (approximately 100 m above the 105 observed site) was used as an endpoint in the model. Continental air masses crossed industrial areas where located in the Yangtze River Mid-Reaches city cluster (Figure 2a). 106 107 The site was possibly affected by industrial emissions under the influence of continental air masses. Please refer to Lines 189-196 of the revised manuscript. 108

109 (a)

---

## Author Response (AR1)

**Response to comments**

*Manuscript Number: acp-2017-23*
*Title: In situ chemical measurement of individual cloud residue particles at a mountain site, South China. Qinhao Lin et al.*

**Anonymous Referee #1**

*Lin et al describe measurements of the chemistry of cloud droplet residues at a mountain-top site in South China in January 2016. Few measurements of cloud droplet residual chemistry exist, so these are important measurements to help improve our knowledge of cloud formation and properties, which are important for predicting weather and climate. Detailed comments follow.*

We would like to thank the reviewer for his/her useful comments and recommendations to improve the manuscript. We agree with the comments, and careful revision has been made according to the suggestions.

*It would be helpful for the authors to provide additional information about the cloud events. Please provide, at minimum, ambient temperature during the cloud events to justify the presence of cloud droplets only and no influence from ice crystals would be useful to discuss (related to Line 123); this is shown in Fig 1 and would be useful to refer to earlier to justify the presence of cloud droplets only.*

We agree with the comments, and additional information has been added in the revised manuscript as suggested: Ambient temperature on average was 6.9 ℃ (ranging from -7.2 to 11.4 ℃) during the cloud events in this study. Therefore, the clouds here consisted of liquid droplets only. Please refer to Lines 124-126 of the revised manuscript.

*It is stated on Line 135 that previous studies found an average cloud droplet size of 10 um at this site, but the distribution is not discussed, nor is the time of year of the previous measurements. Since this work is published in Chinese, these information are not easily obtained by the reader. So, additional discussion would be helpful.*

Measurement of drop size spectrum in this region performed during winter of 1999-2001 shows that size of cloud droplets ranged from 4 to 25 μm, with average size of 10 μm and a corresponding liquid water content of 0.11-0.15 g m$^{-3}$ (Deng et al., 2007). Some studies in other locations also showed an average size at ~10 μm (Freud et al.,

2008; Shingler et al., 2012). Therefore, it is reasonable to select a cut size at 8 μm for cloud droplets in the present study. The discussion has been added in section 2 as suggested, Please refer to Lines 126-131 of the revised manuscript.

*Is it correct that measurements of cloud droplet size were not completed during this study? It would be helpful to know what fraction of the cloud droplet population was sampled, given the cut size of 8 um. For interpretation of the comparison between the cloud droplet residues and ambient particles, it is important to understand what fraction of the cloud droplets were measured. Previous studies (e.g. Bator & Collett 1997, J. Geophys. Res.) have found that cloud chemistry varies with droplet size. Sampling only the larger cloud droplets may also bias the cloud droplet residue size to larger particles, which is one of the observations. Since the cloud droplet activation process is also size-dependent, it is not possible for the reader to evaluate measurement vs droplet activation size dependencies currently.*

It is true that measurements of cloud droplet size were not completed during this study. We agree with the comment. Sampling only the larger cloud droplets may also bias the cloud droplet residue size to larger particles. Previous measurements found that dust, playa salts or sea salt particles are often enriched in larger cloud droplets (~20 μm) (Bator and Collett, 1997; Pratt et al., 2010b). Organic carbon tends to be enriched in small cloud/fog droplets, extending to 4 μm (Herckes et al., 2013). It has been clarified that cloud droplets above 8 μm were sampled by the GCVI. Thus, it partially leads to relatively larger fractions of the Dust and Na-rich cloud residues observed, while the fraction of the OC cloud residues might be underestimated. Please refer to Lines 320-325 of the revised manuscript.

Major Comments:
*Lines 20-23, lines 220-221, Figure 4, & numerous other locations: Do these number fractions take into account the size bias in the instrument inlet transmission efficiency, which is clear in Figure S3? It is clear that there are particle size dependencies to the cloud residual chemical composition, particularly for the amine and aged EC particle types, that should be considered when reporting fractions. For example, even on lines 220-221, it is not clear if the authors are reporting 3.8% of the total cloud residues, or 3.8% of the particles measured from 0.7-1.9 um, or 3.8% across each of the size bins from 0.7-1.9 um.*

We agree with the comments. The chemical composition of cloud residues is dependent on the particle size (Roth et al., 2016), and the number reported for each particle type might suffer the bias from size-dependent transmission efficiency (Qin et al., 2006). The relative fraction of cloud residues in 100 nm size interval is presented to minimize the size-dependent transmission efficiency of single particle mass spectrometry (Roth et al., 2016). Similarly, we have provided information on the number fractions of amine and aged EC particle types in cloud residues with size. Nf of the aged EC residues significantly decreased from 54.1% in the size range of 0.2-1.0 μm to 19.2% in the size range of 1.1-1.9 μm. The Amine particles contributed to 3.8% by number of the total cloud residues. Moreover, higher Nf of the Amine residues was detected in size range from 0.7 to 1.9 μm relative to size range from 0.2 to 0.6 μm (16.7% versus 0.4%). Please refer to Lines 20-23, 223-230 and 260-263 of the revised manuscript.

*Lines 64-68: Only two other cloud studies are mentioned, or referenced, here, which incorrectly suggests to the reader that the measurement of anthropogenic particles in clouds has only been measured twice. While not all papers need to necessarily be referenced here, it is important to provide a comprehensive view to the reader.*

We have added references (Stier et al., 2005; Sorooshian et al., 2007b; Lohmann et al., 2007; Rosenfeld et al., 2008; Roth et al., 2016; Seinfeld. et al., 2016; Li et al., 2017) to discuss the anthropogenic influence on cloud. Anthropogenic particles can increase number concentration of small cloud droplets, in turn, affect reflectivity and life time of clouds (Stier et al., 2005; Lohmann et al., 2007; Rosenfeld et al., 2008). In-situ cloud chemical measurements show varied chemical composition of cloud droplets at various regions (Sorooshian et al., 2007a; Roth et al., 2016; Li et al., 2017). Although a large number of aerosol/cloud studies over the past 20 years, the uncertainty for evaluating radiative forcing due to aerosol-cloud interactions has not been reduced (Seinfeld. et al., 2016). Please refer to Lines 55-62 of the revised manuscript.

*Lines 87-89: This list is not comprehensive and is missing many papers. The authors are strongly suggested to conduct a detailed literature search, as comparison of their results with these papers is important.*

We agree with the comment, and we have added related references about combined technique of a CVI and AMS or single particle measurement. These studies were mainly conducted in North America including Wyoming (Pratt et al., 2010a), Ohio (Hayden et al., 2008), Oklahoma (Berg et al., 2009), Florida (Cziczo et al., 2004; Twohy et al., 2005), California (Coggon et al., 2014), Europe including Schmücke (Roth et al., 2016; Schneider et al., 2017), Jungfraujoch (Kamphus et al., 2010), Åreskutan (Drewnick et al., 2007), Scandinavia (Targino et al., 2006), Arctic (Zelenyuk et al., 2010), Central America (Cziczo et al., 2013), West Africa (Matsuki et al., 2010) and Oceans (Twohy et al., 2009; Twohy et al., 2008; Shingler et al., 2012). Please refer to Lines 81-88 of the revised manuscript.

*Line 205 & Figure 3: Provide possible ion marker formulae here. Provide a reference for the aged EC type based on comparison to other single-particle mass spectrometry studies (e.g., reference to Moffet and Prather 2009, PNAS). Given the strong K+ signal here, it is likely that both the "Aged EC" and "K-rich" are from biomass burning. Similarly, the OC particles are likely aged biomass burning particles as well, given the strong K+ signal.*

Possible ion marker formulae (m/z 27 $C_2H_3^+$, 43 $C_2H_3O^+$) were provided as suggested. We also noted that the aged EC type is similarly observed by other single-particle mass spectrometry studies (e.g., Moffet and Prather 2009, PNAS). We agree with the comment that the "Aged EC", "OC" and "K-rich" might be from biomass burning,as also discussed later. We have discussed the possibility, please refer to Lines 219-222, 232-233 and 314-315 of the revised manuscript.

*Lines 209-210: The prevalence of wildfires, shown in Figure 2, suggests that these particles are primarily from biomass burning (see Pratt et al. 2011, ACP for a single particle mass spectrum of fresh biomass burning particles). Pratt et al. (2010, J. Atmos. Sci.) and Hudson et al. (2004, J. Geophys. Res.) discuss the identification of aged biomass burning particles by single-particle mass spectrometry. It seems that the authors can say with greater certainty the source of these particles, and discussion of this would elevate the paper by providing another evaluation of the influence of biomass burning aerosols on cloud activation, which is important and interesting.*

We greatly appreciate the comment. Pratt et al (2011) has been cited here to identify the K-rich particles as biomass burning origin. Related references (Hudson et al., 2004; Pratt et al., 2010) were also cited to discuss the identification of aged biomass burning particles and to evaluate the influence of biomass burning aerosols on cloud activation. Majority of the K-rich cloud residues observed here are expected to originate from long-range transportation. Aging process during long-range transportation can increase soluble species (e.g., sulfate, nitrate and oxalate) in the K-rich particles, in turn, improve CCN activity. Please refer to Lines 232-238 and 464-468 of the revised manuscript.

*Lines 221-227: It is not clear if Roth et al. observed a decreased fraction of amines in*

*the clouds compared to the ambient, or if amines just do not influence the site. The presence and behavior of amines would be location and season dependent, so this discussion is not clear and seems to be comparing studies without considering commonalities.*

We agree with the comment that the presence and behavior of amines would be location and season dependent. We have made it clear by comparing an Nf of amine-containing aerosol between cloud residues and background aerosol (9% versus 2% by number) in Roth et al. (2016). It indicates a preferential formation of amine in cloud. Aqueous reaction improving the participation of amine has been observed in Guangzhou (Zhang et al., 2012a) and Southern Ontario (Rehbein et al., 2011). Please refer to Lines 263-267 of the revised manuscript.

*Lines 244-247: To further understand the source of the Na-rich particles, were they present for both coastal and non-coastal wind directions? Is there a difference in the average mass spectra for these wind directions considering the minor peaks? Are industrial sources located in both directions?*

Na-rich particles were resulted from varied sources, including industrial emissions, sea salt or dry lake beds (Moffet et al. 2008). The Nf of the Na-rich cloud residues did not increase from continental (Northerly) air mass to maritime (southwesterly) air mass on 21 Jan (3.3% versus 2.4% by number). However, related sea salt ion peak area (m/z, $81/83Na_2{}^{35}Cl/Na_2{}^{37}Cl$) were enhanced for Na-rich particles origination from maritime air mass relative to continental air mass (3.8 ± 2.4 times). Continental air masses crossed industrial areas where located in the Yangtze River Mid-Reaches city cluster (Figure 2). Industrial emissions was a possible contributor to Na-rich particles under the influence of continental air masses (Wang et al. 2016). This might suggests that the Na-rich particles were contributed by both the industrial emissions and sea salt. Therefore, under the influence of maritime air mass, the signals for sea salt contribution became stronger. Please refer to Lines 299-310 of the revised manuscript.

*Lines 263-267: While ammonium nitrate does not contribute to the dust nitrate observed here, it is not appropriate to generalize this statement to suggest that previous studies in other very different locations did not measure ammonium nitrate.*

We agree with the comment. We have revised this sentence to "in this region, ammonium nitrate was not a predominant form of nitrate in the Dust cloud residues", to make it clear. Please refer to Lines 339-340 of the revised manuscript.

*Lines 319-329: Was there a mass spectral difference in the K-rich particle type between the N and SW wind directions that would aid in source identification for the two air masses?*

A similarity in averaged mass spectrum of the K-rich residues was found for the southwesterly and northerly air masses (Figure S6). Please refer to Lines 419-420 of the revised manuscript.

[Figure]

Figure S6: Average mass spectra of K-rich residues for southwesterly (a) and northerly (b) air masses.

*Section 3.6: This authors should consider incorporating this discussion into the other sections of the manuscript so that comparisons are made when results are discussed. In addition, the authors should consider differences in atmospheric composition at the various sites. when discussing specific comparisons (i.e. are the contributing sources and magnitudes the same, or may this be a reason for differences? Or, are the seasons the same?). Currently only a general statement on lines 431-433 states that differences are specific to geographic location. As noted above, the literature cited is also not comprehensive, and this section would benefit from additional literature searching.*

We agree with the comment. We have incorporated this discussion into the other sections of the manuscript. We have also discussed same or different reason for sources and magnitudes of cloud residues at various sites. We have also cited related literatures (Drewnick et al., 2007; Twohy et al., 2008; Twohy et al., 2009; Matsuki et al., 2010; Kamphus et al., 2010; Pratt et al., 2010b; Zelenyuk et al., 2010; Roth et al., 2016; Bi et al., 2016). Please refer to Lines 241-253, 273-275, 464-471, 489-494 and 505-509 of the revised manuscript.

*Figure 4: It would be useful to add comparisons to the ambient and interstitial particles here. In addition, since the cloud residue types changed significantly based on air mass*

*origin (N vs SW), as stated in Section 3.4, it would be useful to separate out these wind directions and show the fractions of cloud residue, interstitial, and ambient, separated for the two wind directions.*

We agree with the the comment. The fractions of the ambient and non-activated particles were provided in Figure 7. The fractions of cloud residue in comparison to ambient particles was performed based on northerly air mass. Please refer to section 3.4 of the revised manuscript.

During the sampling period, the cloud events occurred once the southwesterly air masses were dominant. Therefore, a comparison between cloud residues and ambient particles cannot be addressed under the influence of southwesterly air masses. Please refer to Lines 445-448 of the revised manuscript.

A comparison of cloud residues and non-activated particles has been performed. However, from 22 to 23 Jan during cloud III events, the air mass encountered initial mixing of cloud-free air originated from north and cloudy air originated from southwest. Therefore, a comparison of cloud residues and non-activated particles was not performed for a special wind direction during cloud III events. Please refer to section 3.5 of the revised manuscript.

[Figure]

Figure 7: Number fraction of the cloud residues, ambient and non-activated particles. (a) cloud residues during northerly air mass; (b) ambient particle during northerly air mass; (c) cloud residues during southwesterly air mass; (d) cloud residues and (e) non-activated particles were alternately sampled with interval of one hour during the cloud III event; Uncertainties were calculated assuming Poisson statistics for analyzed particles.

Technical Comments:

*Lines 24-26, 45, & other locations (search document): Please clarify what is meant by "intensity" in these statements.*

Intensity refers to average ion peak area. We have clarified them. Please refer to Lines 24, 43-44 and 486 of the revised manuscript.

*Lines 29-30: The phrase "To estimate how atmospheric aerosol particles respond to chemical properties of cloud droplets" is not clear since aerosols determine cloud droplet chemistry, outside of aqueous processing from dissolved trace gases.*

The phrase has been changed to " To estimate how atmospheric aerosol particles interact with chemical composition of cloud". Please refer to Line 27 of the revised manuscript.

*Lines 41-42: For readers not familiar with the region, it would be helpful to know the suggested source of the amine particles, which the authors are presumably referring to with respect to the wind direction change.*

Sources of the amine particles (e.g., ocean and livestock areas) were provided under the influence of southwesterly air masses. Please refer to Lines 40 of the revised manuscript.

*Line 44: What does "highly associated" mean in this context? This phrase occurs in other locations in the manuscript as well.*

We have modified them. Higher Nfs of nitrate (88-89%) were found in the Dust and Na-rich cloud residues relative to sulfate (41-42%) and ammonium (15-23%). Please refer to Lines 41-43 and 530-532 of the revised manuscript.

*Lines 76-80: This sentence incorrectly cites Pratt et al 2010a for cloud droplet residues. The authors should also consider Drewnick et al 2007, J. Atmos. Chem., who did observe lower sulfate mass fractions for droplet residues compared to ambient aerosol. Pratt et al 2010 (J. Geophs. Res.) shows increased mixing with sulfate/nitrate for liquid clouds, compared to ice clouds, as another example.*

We have deleted the citation of Pratt et al 2010a and added a citation of Drewnick et al 2007 here. Please refer to Lines 74-75 of the revised manuscript.

A comparison of liquid clouds and ice clouds is beyond the scope of this work, Pratt et al 2010 (J. Geophs. Res.) was not cited here.

*Line 161: It should be clarified that "cloud droplet residue concentrations", not "cloud droplet concentrations" were measured by the SMPS. This is an important distinction.*

We have corrected "cloud droplet concentration" to "cloud residual concentration" throughout the manuscript. Please refer to Lines 165-168 of the revised manuscript.

*Lines 163-165: Wouldn't hazy days with low visibility be characterized by high, rather than low, PM2.5 concentrations? This is confusing.*

Low level of $PM_{2.5}$ ($\sim$ 12.7 μg m$^{-3}$) excludes the influence of hazy days. Please refer to Lines 169-170 of the revised manuscript.

*Line 165: Change "rainy" to "rain".*

We have changed "rainy" to "rain" accordingly. Please refer to Line 171 of the revised manuscript.

*Line 216: It is important to consider the relative enhancement in amine peaks when using a 266 nm laser and that the amines themselves may not comprise the majority of the particle mass. See Pratt et al. 2009, Environ. Sci. Technol.*

We have emphasized the effect of 266 nm ionization laser on amine peaks (Pratt et al., 2009). Please refer to Lines 257-259 of the revised manuscript.

*Lines 230-232: It should be clarified when discussing previous vs the current study, and if previous studies are being discussed, the season should be noted if there are seasonal variations.*

We agree with the comments. We have clarified the specific season in the discussion. At Mt. Tai in northern China, a high concentration of $Ca^{2+}$ in cloud/fog water was mainly attributed to a sandstorm event during spring season (Wang et al., 2011). At Mt. Heng in southern China, abundant crust-related elements (e.g., Al) observed in cloud water is due to Asian dust storms occurring on March−May (Li et al., 2017). Based on backward trajectory, the site in this study was less affected by sandstorm source in northwestern China during cloud events. Local dust emission by anthropogenic-disturbing soils or removing vegetation cover can be excluded as a result of forest protection. Therefore, a low fraction (2.9% by number) of dust cloud residue is acceptable in the present study. Please refer to Lines 278-287 of the revised manuscript.

*Lines 231-237: Some grammar fixes are needed here.*

The language has been edited by a native speaker.

*Line 279: Please be more specific with the statement "plays a key role in cloud processes". you mean that these particles were preferentially activated? Is there evidence of this?*

We have reworded this sentence: "This result also implies that ammonium-containing particles are preferentially activated or enhanced by uptake of gaseous $NH_3$ to neutralize acidic cloud droplets for the OC and EC types. Please refer to Lines 354-356 of the revised manuscript.

*Lines 358-396: The phrasing in these paragraphs should be improved for greater clarity and correct grammar.*

These paragraphs have been reworded. Please refer to Lines 463-509 of the revised manuscript.

*Line 390: This is an important finding, yet the phrasing "sulfate was observed to diminish" not clear, particularly when considering the following sentence. Please clarify.*

We have rephrased and changed to "sulfate intensity was observed to diminish". Please refer to Lines 502-503 of the revised manuscript.

*Lines 395-396: What discrepancy? This is not clear.*

The discrepancy refers to "the mass or number fraction of sulfate-containing particles in the cloud residues changed between ambient and interstitial (non-activated) particles (Drewnick et al., 2007; Twohy and Anderson, 2008; Schneider et al., 2017)". We have changed "this discrepancy" to "these changes" to make it clear. Please refer to Lines 505-509 of the revised manuscript.

*Line 397: Quantitatively, what is "no remarkable change"? The phrase "remarkable*

*change" is used elsewhere in the manuscript as well, but it isn't defined.*

"no remarkable change": We added the data on number factions when the residual particles were compared with ambient or non-activated particles. "remarkable change" has been modified to "remarkable decrease/increase" and added the data on number factions when the residual particles were compared with ambient or non-activated particles. Please refer to Lines 511-513 and 518-519 of the revised manuscript.

*Table 2: Not sure what the authors mean by "way" here.*

"way" : Cloud residues and non-activated particles were alternately sampled with interval of one hour during the cloud III event. Table 2 has been replaced by pie charts in Figure 7. Please refer to Lines 452-453 of the revised manuscript and the caption of Figure 7.

*Figure 2: The lines and numbers on this map are difficult to read. It would be useful to make the text bold perhaps and increase the width of the lines.*

We have changed accordingly. Please refer to the modified Figure 2a.

[Figure]

*Figure 3: This plot is difficult to read. The authors should consider showing only the most abundant (and discussed) particle types in the main text figure and moving the others (including "Other", which is somewhat meaningless as an average mass spectrum if it is made up of a diverse population of particles) to the supplemental*

*information.*

[Figure]

Figure 3: Averaged positive and negative mass spectra for the main 6 particle types (Aged EC, K-rich, Amine, Dust, Fe, Na-rich) of the sampled particles during the whole sampling period. RPA in the vertical axis refers to relative peak area. m/z in the horizontal axis represents mass-to-charge ratio.

[Figure]

Figure S4: Averaged positive and negative mass spectra for Pb, OC and Other types of the sampled particles during the whole sampling period. RPA in the vertical axis refers to relative peak area. m/z in the horizontal axis represents mass-to-charge ratio.

*Figures 7 and 8: Please indicate whether positive peak areas indicate preferentially in the cloud residues and negative indicate preferentially in the ambient/interstitial particles. This is currently not clear in the figure captions.*

We have clarified them in the captions of Figures 8 and 9 of the revised manuscript.
Figure 8: Mass spectral subtraction plot of the average mass spectrum corresponding to cloud residues minus ambient particles. Positive peak area corresponds to higher abundance in cloud residues, whereas negative peak area show higher intensity in ambient particles.
Figure 9: Mass spectral subtraction plot of the average mass spectrum corresponding to cloud residues minus non-activated particles. Positive peak area correspond to higher abundance in cloud residues, whereas peak area show higher intensity in non-activated particles.

*Figure S4: It is not clear in the maps where RH < and > 90% are located.*

We have added contour lines of two relative humidity (50%, 70%) in the Figure S5. Please refer to the modified Figure S5.

[Figure]

Figure S5: At 850 hPa (about 1,500 m a.s.l), Contour lines (red lines) of relative humidity 90%; Contour lines (green lines) of relative humidity 70%; Contour lines (yellow lines) of relative humidity 50%. Black mark represents the observed site. Data is available at ftp://arlftp.arlhq.noaa.gov/pub/archives/gdas1/.

References:

Bator, A. and Collett, J.L.: Cloud chemistry varies with drop size, J. Geophys. Res. Atmos., 102, 28071-28078. 1997.

Berg, L.K., Berkowitz, C.M., Hubbe, J.M., Ogren, J.A., Hostetler, C.A., Ferrare, R.A., Hair, J.W., Dubey, M.K., Mazzoleni, C. and Andrews, E.: Overview of the cumulus humilis aerosol processing study, B. Am. Meteorol. Soc. 90, 1653-1667, 2009.

Bi, X., Lin, Q., Peng, L., Zhang, G., Wang, X., Brechtel, F. J., Chen, D., Li, M., Peng, P. a., Sheng, G. and Zhou, Z.: In situ detection of the chemistry of individual fog droplet residues in the Pearl River Delta region, China, J. Geophys. Res. Atmos., 121, 9105-9116, 2016.

Coggon, M.M., Sorooshian, A., Wang, Z., Craven, J.S., Metcalf, A.R., Lin, J.J., Nenes, A., Jonsson, H.H., Flagan, R.C., Seinfeld, J.H.: Observations of continental biogenic impacts on marine aerosol and clouds off the coast of California, J. Geophys. Res. Atmos., 119, 6724-6748, 2014.

Cziczo, D.J., Froyd, K.D., Hoose, C., Jensen, E.J., Diao, M., Zondlo, M.A., Smith, J.B., Twohy, C.H., Murphy, D.M.: Clarifying the dominant sources and mechanisms of cirrus cloud formation,

Science, 340, 1320-1324, 2013.

Cziczo, D.J., Murphy, D.M., Hudson, P.K., Thomson, D.S.: Single particle measurements of the chemical composition of cirrus ice residue during CRYSTAL-FACE, J. Geophys. Res. Atmos., 109, D04201, doi:10.1029/2003JD004032, 2004.

Deng, X., Wu, D., Shi, Y., Tang, H., Fan, S., Huang, H., Mao, W. and Ye, Y.: Comprehensive analysis of the macro-and micro-physical characteristics of dense fog in the area south of the Nanling Mountains (in Chinese), J.Trop. Meteorol., 23, 424-434. 2007,

Drewnick, F., Schneider, J., Hings, S. S., Hock, N., Noone, K., Targino, A., Weimer, S., and Borrmann, S.: Measurement of ambient, interstitial, and residual aerosol particles on a mountaintop site in central Sweden using an aerosol mass spectrometer and a CVI, J. Atmos. Chem., 56, 1-20, 2007.

Freud, E., Rosenfeld, D., Andreae, M.O., Costa, A.A. and Artaxo, P.: Robust relations between CCN and the vertical evolution of cloud drop size distribution in deep convective clouds, Atmos. Chem. Phys., 8, 1661-1675, 2008.

[revised manuscript text omitted]

**Referee #2: J. Schneider**

*In their manuscript "In situ chemical measurement of individual cloud residue particles at a mountain site, South China", Qinhao Lin and co-workers report on the analysis of single particles from cloud residues using a single particle aerosol mass spectrometer. They observed a high fraction of EC-containing particle in the residuals and detected amines with a high variability. Nitrate was found to be increased in residuals compared to ambient particles, while sulfate showed a dependency on the chemical composition of the residues. The topic of the paper is well suited for ACP, and the data itself are interesting, because single particle measurements of cloud residuals are still sparse. However, the manuscript suffers from many unclear statements and some severe uncertainties regarding the analysis of interstitial particles. I have many points where more information is needed or where I disagree. None of these points alone would be a "major" comment, but the multitude of my remarks and questions suggest to require a major revision and to reconsider the manuscript after my comments listed below have been addressed.*

We would like to thank Prof. J. Schneider for his useful comments and recommendations to improve the manuscript. We agree with the comments, and careful revision has been made accordingly, please refer to the following responses for details.

Comments and remarks:
*Title: I suggest to change the title to "In situ chemical composition measurement of individual cloud residue particles at a mountain site, South China"*

We have changed accordingly. Please refer to Lines 1-2 of the revised manuscript.

*Page 4, lines 64 – 68:*
*More references are needed here to discuss the anthropogenic influence on cloud particles, not just two papers on single particle analysis.*

As also suggested by Reviewer 1, We have added references (Stier et al., 2005; Sorooshian et al., 2007b; Lohmann et al., 2007; Rosenfeld et al., 2008; Roth et al., 2016; Seinfeld. et al., 2016; Li et al., 2017) to discuss the anthropogenic influence on cloud particles. Anthropogenic particles can increase number concentration of small cloud droplets, in turn, affect reflectivity and life time of clouds (Rosenfeld et al., 2008; Stier et al., 2005; Lohmann et al., 2007). In-situ cloud chemical measurements show varied chemical composition of cloud droplets at various regions (Sorooshian et al., 2007a; Roth et al., 2016; Li et al., 2017). Although a large number of aerosol/cloud studies over the past 20 years, the uncertainty for evaluating radiative forcing due to aerosol-cloud interactions has not been reduced (Seinfeld. et al., 2016). Please refer to Lines 55-62 of the revised manuscript.

*Page 5, line 79: Replace "Nf of sulfate" by "NF of sulfate-containing particles"*

We have changed it accordingly. Please refer to Line 74 of the revised manuscript.

*Page 5, line 80: Replace "other study" by "other studies"*

We have changed it accordingly. Please refer to Line 76 of the revised manuscript.

*Page 7, line 122: Was the humidity measured in the evaporation chamber? How do you make sure that all water evaporates?*

Relative humidity (RH) was around 30% in the evaporation chamber, thus it can be assumed that the majority of water was evaporated. Please refer to Lines 131-134 of the revised manuscript.

*Page 8, lines 147-149: Did you do the size calibration on the mountain top station? What was the ambient pressure during the measurements and during the calibration? Did you check the inlet flow or the pressure inside the aerodynamic lens?*

We did the size calibration on the mountain top station. Polystyrene latex spheres (Nanosphere Size Standards, Duke Scientific Corp., Palo Alto) of 0.2-2.0 μm in diameter were used to calibrate the sizes of the detected particles on the mountain top station. The ambient pressure was 830 hPa (826-842 hPa) during the measurements and during the calibration. The pressure inside the aerodynamic lens maintains about 3 hPa during the measurements and during the calibration. Please refer to Lines 152-155 of the revised manuscript.

*Page 8, line 161: The SMPS does not measure the cloud droplet concentration but the cloud residue concentration. Cloud droplets would have to be measured outside in the cloud (by FSSP or similar instrumentation).*

We have corrected the mistake. "cloud droplet concentration" was replaced with "cloud residual concentration". Please refer to Lines 165-168 of the revised manuscript.

*Page 9, line 171: As I will outline in more detail below, I doubt the existence of interstitial particles in this size range.*

The period of collecting interstitial particles on 22-23 Jan encountered initial mixing of northerly cloud-free air (dry and cold airstreams) and southwesterly cloudy air (moist airflows). The dry northern air mass might lower supersaturation, only larger particles could be activated. This might result in non-activated particles observed to be above 200 nm here (Mertes et al., 2005; Kleinman et al., 2012; Hammer et al., 2014). To make it more accurate, we prefer to name "non-activated particles", rather than "interstitial particles". We have clarified them. Please refer to Lines 452-461 of the revised manuscript.

*Page 9 lines 184-185 and Figure 2: More info on the trajectories is needed: How did the vertical evolution look like? How well is the mountain represented in the model? Is 1800 m the best altitude that represents the mountain site? Please add also the most important megacities to the map to help estimating the influence of anthropogenic emissions.*

We have added the vertical evolution of the trajectories. The beginning of southwesterly air masses traversed at lower heights relative to northerly air masses. Please refer to the Figure 2 (b). Heights of the HYSPLIT model in the study region (a spatial resolution of $0.5° \times 0.5°$) was averaged 500 m a.s.l, which was lower than height of the observed site (1,690 m a.s.l). Therefore, a height of 1,800 m a.s.l. (approximately 100 m above the observed site) was used as an endpoint in the model. Continental air masses crossed industrial areas where located in the Yangtze River Mid-Reaches city cluster (Figure 2a). The site was possibly affected by industrial emissions under the influence of continental air masses. Please refer to Lines 189-196 of the revised manuscript.

(a)

[Figure]

(b)

[Figure]

Figure 2: (a) HYSPLIT back trajectories (72 h) for air masses at 1,800 m during the whole
sampling period. The white borders and circle refers to the Pearl River Delta (city cluster
1) and Yangtze River Mid-Reaches city cluster (city cluster 2), respectively. The yellow
rots represent fire dots during the study periods. The fire dots are available at
https://earthdata.nasa.gov/; (b) Heights (above model ground) of the air masses as a
function of time.

*Page 10, line 207: I suggest moving Figure S3 to the main paper.*

Figure S3 has been moved to the main paper, please refer to Figure 4 in the revised manuscript.

*Page 11, lines 221-222: But Roth et al. found a clear enhancement of amines in residues compared to the background aerosol (9% to about 2%).*

We have added a comparison Nf of amine-containing particles between cloud residues and background aerosol reported by Roth et al., (2016). Please refer to Lines 265-267 of the revised manuscript.

*Page 12, lines 235-237: I agree that dust is found more frequently in the coarse particle size range, but then I would expect to see an increase of the dust fraction in the residues with increasing diameter. This is not seen in Figure S3. Is the identification of dust reliable? What about the Fe-containing particles? They might be dust as well.*

We agree with the comment. As a matter of fact, Nf of dust cloud residues generally increased with increasing diameter. Please refer to Lines 276-278 of the revised manuscript.

Approximately 16% of the Fe cloud residues contained Ca peak (m/z 40). Relatively weak Na and K peaks in the Fe particles possibly contributes to anthropogenic sources (Zhang et al., 2014), especially northern air mass across iron/steel industrial activities in Yangtze River Mid-Reaches city clusters (Figure 2). These might suggest that Fe cloud residues was likely to have come from mixed sources. Please refer to Lines 290-294 of the revised manuscript.

[Figure]

*Page 12, lines 238-243: What could be the source of these Pb- and Fe-containing particles?*

*See also comment above. Can the Fe-containing particles belong to the dust-type?*

As mentioned above, the Fe cloud residues contain Ca peak (m/z 40, 16% by number) and relatively weak Na and K peaks, which possibly contributes to anthropogenic sources (Zhang et al., 2014), especially northern air mass across iron/steel industrial activities in

Yangtze River Mid-Reaches city clusters (Figure 2). These might suggest that this particle type likely originated from mixed sources. Please refer to Lines 290-294 of the revised manuscript.

The Pb particles showed its typical ions at m/z $208Pb^+$ and internally mixed with K and Cl.

Previous studies found that K and Cl internally mixed with Pb particles have a possible origination of waste incineration (Zhang et al., 2009) or iron and steel facility (Tsai et al.,

2007). Please refer to Lines 315-318 of the revised manuscript.

*Page 12, lines 244-252: If these particles were from sea salt, they should contain chloride*

*ions. That is hard to see in Figure 3. Are these Na-rich particles correlated with air masses*

*coming from the ocean?*

Na-rich particles result from varied sources of industrial emissions or sea salt particles and dry lake beds (Moffet et al. 2008). The Nf of the Na-rich cloud residues did not increase from continental (Northerly) air mass to maritime (southwesterly) air mass on 21 Jan (3.3%

versus 2.4% by number). However, sea salt ion peak areas (m/z, $81/83Na_2{}^{35}Cl/Na_2{}^{37}Cl$)

were enhanced for Na-rich particles origination from maritime air mass relative to continental air mass (3.8 $\pm$l2.4 times). Continental air masses crossed industrial areas where located in the Yangtze River Mid-Reaches city cluster (Figure 2). Industrial emissions was a possible contributor to Na-rich particles under the influence of continental air masses (Wang et al. 2016). This might suggests that the Na-rich particles were contributed by both the industrial emissions and sea salt. Therefore, under the influence of maritime air mass, the signals for sea salt contribution became stronger. Please refer to

Lines 299-310 of the revised manuscript.

*Page 13, line 259 and Figure 5: How do you distinguish between sulfuric acid and sulfate?*

*Besides, spelling (sulfate, sulphuric acid) should be consistent ("f" or "ph").*

Sulfate ion peak at m/z -97 $HSO_4^-$ and sulfuric acid cluster ion peak at m/z -195 $[H(HSO_4)_2^-]$

were given in previous single particle studies (Pratt et al., 2009; Rehbein et al., 2011). "Sulphuric acid" has been replaced by "Sulfuric acid". Please refer to the caption of Figure 5.

*Page 13, lines 265-267: What other forms of nitrate do you suggest to be present on the Na-rich and dust residues? What about uptake of nitric acid from the gas-phase by the cloud droplets? How certain is the identification of ammonium? Which peak was used?*

The Na-rich and Dust types were mainly composed of alkaline ion peaks (m/z, $23Na^+$, $39K^+$ and $40Ca^+$) in the position mass spectra (Figure 3). This suggests that rather than $NH_4NO_3$, nitrate might exist in the form of $Ca(NO_3)_2$, $NaNO_3$ or $KNO_3$ in the dust and Na-rich cloud residues. Please refer to Lines 340-344 of the revised manuscript.

We agree with the comment. We have discussed the contribution of uptake of gas-phase $HNO_3$ to enhanced nitrate in the cloud residues and cited Schneider et al. (2017). Please refer to Lines 334-335 of the revised manuscript.

Generally, a $NH_4^+$ ion signal (m/z 18) was used for identification of ammonium in the analysis of single particle mass spectrometry (Pratt et al., 2009). Please refer to Line 338 of the revised manuscript.

*Page 13, line 268-272: The stability of ammonium nitrate depends also on the humidity. In the book by Seinfeld and Pandis (2nd edition, Wiley and Sons, 2006, Chapter 10.4.3) it is shown that at 30% RH ammonium nitrate does not exist above 30 C. I would assume that the dry carrier gas in the evaporation section is below 30% RH. Thus, it may well be that NH4NO3 evaporates in your system.*

We agree with the comment. We have clarified the artificial effect on ammonium nitrate in the cloud residues. Please refer to Lines 344-346 of the revised manuscript.

*Page 13, lines 275-276: The sentences "The presence of abundant sulfate in aged EC cloud residues was considered to be a good CCN species before activation:" needs rephrasing. It is not clear to me what you want to say. Do you mean "aged EC particles mixed with sulfate are good CCN"?*

The sentence has been changed to "aged EC particles mixed with sulfate are good CCN".

Please refer to Lines 350 of the revised manuscript.

*Page 13, line 279: Ammonium will most likely play a key role in the form of ammonium*
*sulfate or ammonium nitrate. In organic particles, amines may play that role*
*(methylamines). Again: how do you identify ammonium and how do you distinguish*
*between amines and ammonium?*

We agree with the comment. Ammonium will most likely play a key role in the form of
ammonium sulfate or ammonium nitrate in the OC and aged EC cloud residues (Zhang et
al., 2017). Please refer to Lines 353-354 of the revised manuscript.

A $NH_4^+$ ion signal (m/z 18) was used for identification of ammonium. The existences of
m/z 59 $N(CH_3)_3^+$ (trimethylamine,TMA) and related amine ion signals m/z
$58C_2H_5NHCH_2^+$ (diethylamine, DEA) and m/z $86C_5H_{12}N^+$ (triethylamine, TEA)were used
for identification of amines (Angelino et al., 2001). Please refer to Lines 254-257 and 338
of the revised manuscript.

*Page 14, line 281 (and Figure 5): Why does oxalate nor correlate with OC?*

Classification of the OC particles mainly based on intense organic carbon ion signals (e.g.,
m/z $27C_2H_3^+$, $37C_3H^+$, $43C_2H_3O^+$ and $51C_4H_3^+$). However, majority of oxalate-containing
particles internally mixed with the K-rich type. Therefore, oxalate was classified to the K-
rich type, probably contributed from biomass burning. Noted that K-rich could contain a
large abundant of organics (Pratt et al. 2011), however, the signals of organics were covered
by the potassium due to its high sensitive to the laser. Please refer to Lines 371-378 of the
revised manuscript.

*Page 14, lines 284-285: What do you mean by "enrichment of TMA in amine cloud*
*residuals"? You observe that in 93% of those cloud residuals that are assigned to the*
*"amine" type contain TMA. That is not surprising, more surprising is that it's not 100%.*
*But that's inside the measurement uncertainties, to my opinion.*

We have changed "enrichment of TMA in amine cloud residuals" to "presence of TMA in
amine cloud residuals". Please refer to Lines 360-361 of the revised manuscript.

Amine family signals m/z $58C_2H_5NHCH^{2+}$ and m/z $86C_5H_{12}N^+$ were also selected to identify amines (Angelino et al., 2001), leading to only 93% of the "amine" residues containing TMA. Particles that exist a peak signal m/z $58C_2H_5NHCH_2^+$, were found to account for 99% of the Amine residues. Please refer to Lines 254-257 of the revised manuscript.

*Page 14, lines 294-295: "This may result in 33% by number to the Amine residues containing oxalate." Please rephrase, not clear what you want to say.*

We have rephrased this sentence to "it may facilitate the entrainment of oxalate (33% by number) in the Amine residues. ". Please refer to Lines 370-371 of the revised manuscript.

*Page 14, line 298: What does "unscaled" mean? These are absolute particle numbers.*

Considering that the SPAMS mainly detected in size range 0.2-2.0 μm and has size-dependent transmission efficiency. Detected particles were not corrected by a SMPS. Therefore, detected particles cannot represent real atmospheric particle level. "unscaled" has been changed to "detected particle counts". Please refer to Line 381 of the revised manuscript.

*Page 14, lines 302-303: You say that the air masses change from northerly on 18 Jan to southwesterly on 19 Jan, but the particles remain similar from 17 Jan (around noon) to 20 Jan (noon). On the other hand, the change in particle types is very abrupt from cloud residuals to ambient on Jan 17.*

Southwesterly wind flow on 19-20 Jan was too weak (~ 2.75 m s$^{-1}$) to dilute particles originated from northerly air masses (Figure 1). Additionally, high RH (90%) contour line at height 1,500 m (a.s.l.) gradually moved to north China from 19 to 20 Jan (Figure S5). These might lead to similar residual particle types observed from 19 Jan to 20 Jan, although the site encountered southwesterly cloudy air on 19-20 Jan (Figure 2). Please refer to Lines 400-404 of the revised manuscript.

Ambient RH showed an abrupt decrease from nearly 100% at 10:00 to 85% at 11:00 on 17 Jan (Figure 1). The entrained particles originated from northern air mass might have insufficient supersaturation to activate as cloud droplets. It leads to a very abrupt change in Nf of particle types from cloud residues to ambient particles on Jan 17. Please refer to Lines 382-390 of the revised manuscript.

*Page 15, lines 322-325: Do verify the possible transport of biomass burning particles to*

*the site, the vertical history of the trajectories is required.*

We have added the vertical evolution of the trajectories. The beginning of trajectories traversed at low heights (about 0-2 km above model ground) of Southeast Asia, where abundant fire dots occurred. Please refer to the Figure 2 (b).

(a)

[Figure]

(b)

[Figure]

Figure 2: (a) HYSPLIT back trajectories (72 h) for air masses at 1,800 m during the whole sampling period. The white borders and circle refers to the Pearl River Delta (city cluster 1) and Yangtze River Mid-Reaches city clusters (city cluster 2), respectively. The yellow rots represent fire dots during the study periods. The fire dots are available at https://earthdata.nasa.gov/; (b) Heights (above model ground) of the air masses as a function of time.

*Page 16, lines 337-229: "Note that after the activation of amine particles, the partitioning of the gas amine on cloud droplets may further contribute to the enhanced Amine cloud residues". That is true, but holds also for other species, as nitrate (HNO3) or water-soluble OC.*

We have strengthened the important contribution of uptake of gaseous $HNO_3$ or water-soluble OC to cloud droplets. Please refer to Lines 334-335, 364-366 and 490-494 of the revised manuscript.

*Section 3.5: Here I have a major concern: You report interstitial particles containing sulfate and nitrate in the size range between 200 and 1300 nm. It is very hard to believe (not to say impossible) that such large particles are not activated in a cloud.*

*Later (page 17, lines 366-368) you write " However, few studies have focused on this issue, in part because interstitial particles show a smaller size than that detected by single-particle mass spectrometry (Roth et al., 2016)." Since the SPAMS has a very similar lower detection size range as the ALABAMA used by my group in Roth et al., 2016), you can not expect that you detect non-activated interstitial particles which should be in the size range below 200 nm.*

*My suspicion is: The clouds became thinner, and entrainment of cloud-free air has mixed "normal" aerosol particles into the cloud. But such particles cannot be referred to as "interstitial". As long as you don't have cloud microphysics (number and size of cloud droplets) or at least liquid water content (Particle Volume Monitor) available, I would suggest to remove this chapter on interstitial particles.*

The period of collecting interstitial particles on 22-23 Jan encountered initial mixing of northerly cloud-free air (dry and cold airstreams) and southwesterly cloudy air (moist airflows). The dry northern air mass might lower supersaturation, only larger particles could be activated. This might result in above 200 nm non-activated particles observed here (Mertes et al., 2005; Kleinman et al., 2012; Hammer et al., 2014). To make it more
accurate, we prefer to name "non-activated particles", rather than "interstitial particles".
We have clarified them. Please refer to Lines 453-461 of the revised manuscript.

*Page 17, line 358 / Table 1: I would prefer a graph with bars or pie charts. I also strongly*
*recommend showing the SMPS size distributions from residues, ambient and interstitial.*
*That might help to identify the issues with the large interstitial particles.*

Table 1 was replaced by pie charts. Please refer to Figure 7 in the revised manuscript. The
SMPS size distributions from residues, ambient and interstitial particle were provided in
Figure S2.

[Figure]

[Figure]

Figure S2: Size distribution (electrical mobility diameter 20-900 nm) of cloud residues (a), ambient (b) and non-activated (c) particles were measured a scanning mobility particle sizer (SMPS). Black lines represent particles concentrations integrated by the SMPS. The data of cloud residual concentrations was corrected by enrichment factor of 5.25.

*Page 18, lines 374-375 / Figure 7 & 8: How are the difference mass spectra of Figure 7*

*and 8 calculated? Is it ambient - residues and interstitial – residues? Or vice versa? How*

*were the spectra normalized? Please explain.*

We have provide the information in the captions of Figures 8 and 9 of the revised manuscript. Please refer to Lines 484-485 of the revised manuscript.

Figures 8: Mass spectral subtraction plot of the average mass spectrum corresponding to cloud residues minus ambient particles. Positive area peaks correspond to higher abundance in cloud residues, whereas negative area peaks show higher intensity in ambient particles.

Figure 9: Mass spectral subtraction plot of the average mass spectrum corresponding to cloud residues particles minus non-activated particles. Positive area peaks correspond to higher abundance in cloud residues, whereas negative area peaks show higher intensity in non-activated particles.

*Page 18, lines 376-382: Why not? I drew the same conclusion as Hayden et al. (2008) in*

*my 2017 paper (Schneider et al., 2017, please note the update from ACPD 2016 to ACP*

*2017). HNO3 uptake may not be the source of the particles but explains the high amount*

*of nitrate found on many particles, also on the Na-rich and dust particles discussed above.*

We agree with the comment. We have strengthened the contribution of uptake of gaseous

$HNO_3$ to the enhanced nitrate in the cloud residues. We have update the citation of

Schneider et al. (2017) from ACPD 2016 to ACP 2017. Please refer to Lines 334-335 and

490-494 of the revised manuscript.

*Page 18, lines 384-386: I agree with that, but wouldn't that support the idea of uptake of*

*HNO3 from the gas phase? If the nitrate content does not play the major role in the*

*activation, but more nitrate is found in the residues, that's an argument for HNO3 uptake.*

We agree with the comment. We have strengthened the contribution of uptake of gaseous HNO$_3$ to the enhanced nitrate in the cloud residues. Please refer to Lines 334-335 and 490-494 of the revised manuscript.

*Page 18, lines 387-388: Can intensity simply be compared like this? What about size effects and matrix effects? But again, an explanation how Figures 7 and 8 were calculated might help here.*

We agree with the comment. Size and matrix might affect the mass spectra of single particles. Such comparison has been perform in previous single particle studies (Moffet et al., 2008; Pratt et al., 2011). In addition to comparison of certain compound's intensity, its size distribution and number fractions of cloud residues was compared with ambient or non-activated particles, to discuss size effect. Please refer to Lines 487-492 and 504-505 of the revised manuscript.

Figure 8 and 9 show differences in average mass spectra for cloud residues versus ambient particles, as well as cloud residues versus non-activated particles, respectively. Intensity refers to peak area. Please refer to Lines 484-485 of the revised manuscript.

Figures 8: Mass spectral subtraction plot of the average mass spectrum corresponding to cloud residues minus ambient particles. Positive area peaks correspond to higher abundance in cloud residues, whereas negative area peaks show higher intensity in ambient particles.

Figure 9: Mass spectral subtraction plot of the average mass spectrum corresponding to cloud residues particles minus non-activated particles. Positive area peaks correspond to higher abundance in cloud residues, whereas negative area peaks show higher intensity in non-activated particles.

*Page 18, lines 391-392: "Compared with interstitial particles, sulfate enhanced in the Fe cloud residues." I think an "is" is missing here.*

We have changed "Compared with interstitial particles, sulfate enhanced in the Fe cloud residues." to "Compared with non-activated particles, sulfate was found to enhance in the Fe cloud residues.". Please refer to Lines 503-504 of the revised manuscript.

*Page 19, lines 398-399: Better: "The in-cloud process has been reported to be an important*
*pathway: : :"*

We have changed accordingly. Please refer to Lines 510-511 of the revised manuscript.

*Page 20, lines 421-422: The Jungfraujoch is a station located mostly in the free*
*troposphere and in a remote region, so the biomass burning contribution can be expected*
*to be lower than at other sites.*
We agree with the comment. We have discussed less number fraction of biomass burning
in the observed cloud residues at the Jungfraujoch station, where located mostly in the free
troposphere and in a remote region. Please refer to Lines 251-253 of the revised manuscript.

Figures
*Figure 3: Please improve resolution. Labels can't be read upon zooming in.*
We have improved Figure 3 resolution. Please refer to the modified Figure 3. Average mass
spectrum of Pb, OC and Other types have been moved to the supplemental information
(Figure S4)

[Figure]

Figure 3: Averaged positive and negative mass spectra for the main 6 particle types (Aged EC, K-rich, Amine, Dust, Fe, Na-rich) of the sampled particles during the whole sampling period. RPA in the vertical axis refers to relative peak area. m/z in the horizontal axis represents mass-to-charge ratio.

[Figure]

Figure S4: Averaged positive and negative mass spectra for Pb, OC and Other types of the sampled particles during the whole sampling period. RPA in the vertical axis refers to relative peak area. m/z in the horizontal axis represents mass-to-charge ratio.

*Figure 6: The ambient particle time series (b) are broader than the corresponding gaps in (a). Please make the Figure broader. You can move the legend with the particle types to above or below the graphs, plus the legend is only needed once.*

Ambient and cloud residues were collected at the same hour, which lead to ambient particle time series (b) broader than the corresponding gaps in (a). Figure 6 has been changed accordingly. Please refer to the modified Figure 6.

[revised manuscript text omitted]

Guangdong Environmental Monitoring Center, Guangzhou 510308, PR China

* Correspondence to: Xinhui Bi (bixh@gig.ac.cn)

Tel.: +86-20-85290195

**Highlights**

1. EC-containing particles comprised the largest fraction of the total cloud residues (49.3% by number), with dominating size 0.2-1.0 μm.

2. Amine particles represented 0.2% to 15.1% by number of the total cloud residues dependent on the air mass history, with dominating size 0.7-1.9 μm.

3. Higher fraction, intensity (average ion peak area) and larger size of nitrate-containing particles were found in the cloud residues relative to the ambient particles.

**Abstract**

[revised manuscript text omitted]

However, few studies employ direct observation of the chemical composition and mixing state of cloud/fog droplets. Li et al. (2011b) used transmission electron microscopy to obtain mixing state of single ambient particle during cloud events at Mt. Tai in northern

China. Their result showed that sulfate-related salts dominated in large particle. Bi et al.

(2016) used a ground-counterflow virtual impactor (GCVI) coupled with a real-time single particle aerosol mass spectrometer (SPAMS) to explore the chemical composition and mixing state of single fog residue particles in an urban area of South China at ground level.

They found abundant anthropogenic emitted particles including soot or element carbon (EC) in fog residues.

Here, we present a study on the chemical composition and mixing state of individual cloud residue particles at a mountain site of South China. The same experimental methods of Bi et al. (2016) were used in this study on the summit of South China's Nanling mountain region. The size distribution, chemical composition and mixing state of cloud residues during cloud events are discussed. Moreover, the chemical compositions of ambient and non-activated particles were also compared with the cloud residues. The aim of this study is to assess the potential effects of anthropogenic aerosols from regional transportation on cloud formation and to investigate the dominant particle types in cloud droplets at a mountain site in South China.

**2 Experimental**

**2.1 Measurement site**

Measurements were carried out 15-26 Jan, 2016. The sampling site was located in the

Nanling Background Station (112 °53' 56" E, 24° 41' 56" N, 1,690 m a.s.l.) at the National

Air Pollution Monitoring System in South China (Figure S1). This station is 200 km north of the metropolitan city Guangzhou and 350 km north of the South China Sea. This site is also surrounded by a national park forest (273 km$^2$) where there are hardly any emissions from anthropogenic activities. However, during the winter monsoon period, air pollution from northern China moves south to the southern coastal region and crosses the study region (Lee et al., 2005).

**2.2 Instrumentation**

In this study, a GCVI inlet system (GCVI Model 1205, Brechtel Mfg. Inc.) was used to sample cloud droplets with a diameter greater than 8 μm. Ambient temperature on average was 6.9 ℃ (ranging from -7.2 to 11.4 ℃) during cloud events in this study. Therefore, the clouds here consisted of liquid droplets only. Measurement of drop size spectrum in this region performed during winter of 1999-2001 shows that size of cloud droplets ranged from 4 to 25 μm, with average size of 10 μm and a corresponding liquid water content of

0.11-0.15 g m$^{-3}$ (Deng et al., 2007). Some studies in other locations also showed an average size at ~10 μm (Freud et al., 2008; Shingler et al., 2012). Therefore, it is reasonable to select a cut size at 8 μm for cloud droplets in the present study. The sampled cloud droplets were passed through an evaporation chamber (air flow temperature at 40 ℃), where the associated water was removed and the dry residue particles (with the air flow RH lower than 30%), considered as CCN, remained. The particle transmission efficiency of the cut size (8 μm) was 50% (Shingler et al., 2012). The enrichment factor of the particles collected by the GCVI inlet was estimated to be 5.25 based on theoretical calculation (Shingler et al., 2012). Ambient particles were collected through an ambient inlet with a cut-off aerodynamic diameter ($d_a$) of 2.5 μm when no cloud events were present. Additionally, non-activated particles were sampled through the ambient inlet during the cloud events.

The cloud droplet residues, ambient or non-activated particles were subsequently analyzed by a suite of aerosol measurement devices, including a SPAMS (Hexin Analytical

Instrument Co., Ltd., Guangzhou, China), a scanning mobility particle sizer (SMPS) (MSP

Cooperation) and an aethalometer (AE-33, Magee Scientific Inc.). Detailed information and parameter settings regarding the GCVI operation can be found in the work of Bi et al.

(2016).

A detailed operational principle of the SPAMS has been described elsewhere (Li et al.,

2011a). Briefly, aerosol particles are drawn into SPAMS through a critical orifice. The particles are focused and aerodynamically sized by two continuous diode Nd:YAG laser beams (532 nm). The particles are subsequently desorbed/ionized by a pulsed laser (266

nm) triggered exactly based on the velocity of the specific particle. The positive and negative ions generated are recorded with the corresponding size of each singe particle.

Polystyrene latex spheres (Nanosphere Size Standards, Duke Scientific Corp., Palo Alto)

of 0.2-2.0 μm in diameter were used to calibrate the sizes of the detected particles on the mountain top station. The ambient pressure was 830 hPa (826-842 hPa) during the measurements and during the calibration. Particles measured by SPAMS mostly fell within the size range of $d_{va}$ 0.2-2.0 μm (Li et al., 2011a).

**2.3 Definition of cloud events**

To reliably identify the presence of cloud events, an upper-limit visibility threshold of 5

km and a lower-limit relative humidity (RH) threshold of 95% were set in the GCVI

software (Bi et al., 2016). Three long-time cloud events occurred during the periods of

16:00 (local time) 15 Jan - 07:00 17 Jan (cloud I), 20:00 18 Jan - 12:00 19 Jan (cloud II)

and 17:00 19 Jan - 13:00 23 Jan (cloud III), as marked in Figure 1. In addition, a cloud event occurred at 14:40 - 15:00 on 17 Jan, but we did not complete an analysis due to the short duration of this cloud event. The measured cloud residual concentration was integrated by a SMPS and was divided by 5.25 (enrichment factor of CVI). The corrected cloud residual concentrations on average were 218 cm$^{-3}$, 284 cm$^{-3}$ and 272 cm$^{-3}$ for cloud

I, cloud II and cloud III, respectively (Figure S2). Note that during cloud events, ambient

RH was close to 100%, as illustrated in Figure 1. Low level of PM2.5 (~ 12.7 μg m$^{-3}$)

excludes the influence of hazy days. A rainfall detector of the GCVI system was also used to exclude rain droplet contamination. When cloud events occurred without precipitation, sampling was automatically triggered by the GCVI control software.

**2.4 Particle classification**

During the study period, a total of 73996 sampled particles including 49322 ambient, 23611

cloud residues and 1063 non-activated particles with bipolar mass spectra were chemically analyzed in the size range of $d_{va}$ 0.2-1.9 μm. The sampled particles were first classified into 101 clusters using an Adaptive Resonance Theory neural network (ART-2a) with a vigilance factor of 0.75, a learning rate of 0.05, and 20 iterations (Song et al., 1999). By manually combining similar clusters, aged EC, Potassium-rich (K-rich), Amine, Dust, Fe,

Pb, Organic carbon (OC), and Sodium-rich (Na-rich), eight major particle types with distinct chemical patterns were obtained, which represented ~99.9% of the population of the detected particles. The remaining particles were grouped together as "Other".

Assuming that number of individual particles follows Poisson distribution, standard errors for number fraction of particle type were estimated (Pratt et al., 2010a).

**3 Results and discussion**

**3.1 Back trajectories and meteorological conditions**

Back trajectories in this study were calculated using the Hybrid Single Particle Lagrangian Integrated Trajectory (HYSPLIT Model). Heights of the HYSPLIT model in the study region (a spatial resolution of $0.5\degree \times 0.5\degree$) is averaged 500 m a.s.l., lower than height of the observed site (1,690 m a.s.l.). Therefore, a height of 1,800 m a.s.l. (approximately 100 m above the observed site) was chosen as an endpoint in the model. During the study period, the station was mainly affected by southwesterly or northerly air masses (Figure 2). In addition, the beginning of southwesterly air masses traversed at lower heights relative to northerly air masses (Figure 2). The southwesterly air masses, accompanied by warm and moist airflows, occurred during 15-17 and 20-21 Jan, which promoted cloud formation (Figure 1). Conversely, the northerly air masses, associated with cool and dry airstreams, occurred during 18 and 23-24 Jan and led to a decrease in temperature and relative humidity. Note that on 18-19 and 22-23 Jan, the air mass encountered initial mixing of northerly cloud-free air and southwesterly cloudy air. Entrained of nuclei particles originated from northern air mass would be activated to become cloud droplets (Sect. 3.4). On the other hand, entrainment of non-activated particles originated from northern air mass has also mixed into the cloud (Sect. 3.5).

Meteorological conditions were unstable, with high southwesterly flow ($\sim 6.5$ m s$^{-1}$) during 15-17 and 20-22 Jan (Figure 1). The level of PM$_{2.5}$ remained low with a value of approximately 3 μg m$^{-3}$ for this time period. A high level of PM$_{2.5}$ ($\sim$20 μg m$^{-3}$) was observed during 18 Jan when the northerly flow dominated. Similarly, the average PM$_{2.5}$ value reached 24 μg m$^{-3}$ during 24 Jan when the local northerly and southwesterly flows occurred alternately. However, the particles still originated from northerly air masses for this period (Figure 2). During 23-24 Jan, a big freeze associated with a violent northerly flow and a wind speed that exceeded the upper-limit speed (~12 m/s) of a wind speed sensor resulted in a sharp decrease in temperature (Figure 1).

**215 3.2 The chemical characterization of cloud droplet residues**

Figure 3 shows the average positive and negative mass spectra of main six particle types.

The aged EC particles were identified by EC cluster ions (e.g., m/z $\pm 12C^{+/-}$, $\pm 36C_3^{+/-}$,

$\pm 48C_4^{+/-}$, $\pm 60C_5^{+/-}$, …) and a strong $K^+$ ion signal (m/z $39K^+$) as well as a sulfate ion signal (m/z $-97HSO_4^-$), and minor organic markers (m/z $27C_2H_3^+$, $43C_2H_3O^+$) (Moffet and

Prather, 2009). EC particles mainly originated from combustion processes (Bond et al.,

2013). Strong $K^+$ ion signal observed here, it is likely that the aged EC particles in part are from biomass burning (Bi et al. 2011). The aged EC particle type was the largest fraction (49.3% by number) of the the total cloud residues (Figure S3). In addition, Nf of the aged

EC residues significantly decreased from 54.1% in the size range of 0.2-1.0 μm to 19.2%

in the size range of 1.1-1.9 μm (Figure 4). Note that the chemical composition of cloud residues is dependent on the particle size (Roth et al., 2016), and the number reported for each particle type might suffer the bias from size-dependent transmission efficiency (Qin et al., 2006). The relative fraction of cloud residues in 100 nm size interval is presented to minimize the influence of size-dependent transmission efficiency of single particle mass spectrometry (Roth et al., 2016).

The K-rich particles exhibited the highest peak at m/z $39K^+$, mainly combined with sulfate and nitrate (m/z $-46NO_2^-$, $-62NO_3^-$). The K-rich particles presumably result from biomass/biofuel burning source (Moffet et al., 2008; Pratt et al. 2011; Zhang et al., 2013).

Aged time 81-88 min of biomass burning particles show increase in the mass fractions of ammonium, sulfate, and nitrate (Pratt et al. 2011). In this study, the K-rich particles would be expected to experience aged process due to strong sulfate and nitrate signals (Hudson et al. 2004; Pratt et al. 2011). Aged biomass burning particles can participate in cloud droplets formation and show an effective CCN activity (Pratt et al. 2010a). The K-rich particle type, the second largest contributor, accounted for 33.9% by number of the the total cloud residues (Figure S3).

The abundant aged soot/EC and biomass burning particles were often detected in cloud residues (Pratt et al., 2010a; Roth et al., 2016). The contribution of local anthropogenic origins to aged soot and/or biomass burning particles in cloud/fog residues has been reported in Schmücke (Roth et al., 2016) and Guangzhou city (Bi et al., 2016). At the North Slope of Alaska, the measurement of biomass burning particles in cloud residues mainly resulted from Asia sources (Zelenyuk et al., 2010). Similarly, majority of aged EC and K-rich cloud residues observed here are expected to originate from long-range transportation due to insignificant sources of local anthropogenic emissions or fire dots. At the Jungfraujoch station (3,580 m a.s.l.) in Europe, the K-rich (biomass burning) particles was only found to contribute 3% of the cloud droplets and the aged EC cloud residue was insignificant (<1% by number) (Kamphus et al., 2010). The Jungfraujoch is a station where located mostly in the free troposphere and in a remote region, so the biomass burning contribution can be expected to be lower than at other sites.

The Amine particles were characterized by related amine ion signals at m/z $58C_2H_5NHCH_2^+$ (diethylamine, DEA), $59N(CH_3)_3^+$ (trimethylamine, TMA) and $86C_5H_{12}N^+$ (triethylamine, TEA) (Angelino et al., 2001; Moffet et al., 2008; Pratt and

Prather, 2010). Note that amine peaks would be enhanced when using a 266 nm ionization laser and that the amines themselves may not comprise the majority of the particle mass (Pratt et al., 2009). This particle type also contained sulfuric acid ion signals at m/z -195H(HSO$_4$)$_2$⁻, indicative of acidic particles (Rehbein et al., 2011). The Amine particles represented 3.8% by number of the total cloud residues (Figure S3). Higher Nf of the Amine residues was detected in size range from 0.7 to 1.9 μm relative to size range from 0.2 to 0.6 μm (16.7% versus 0.4%), as shown in Figure 4. Aqueous reaction improving the participation of amine has been observed in Guangzhou (Zhang et al., 2012a) and Southern Ontario (Rehbein et al., 2011). A recent study also shows a clear enhancement of amine-containing particles in cloud residues compared to the ambient particles (9% versus 2% by number) (Roth et al., 2016). It indicates a preferential formation of amine in cloud. However, this possibility was not supported by the observations of Bi et al. (2016), who did not detect amine-containing particles in fog residues. In this study, the Nf of the Amine particles varied from 0.2% to 15.1% of the total cloud residues dependent on air mass history (see Sect. 3.4).

The Dust particles presented significant ions at m/z 40Ca⁺, 56CaO⁺/Fe⁺, 96Ca$_2$O⁺ and -76SiO$_3$⁻. Internal mixing with sulfate and nitrate in the Dust particle is expect to act as CCN (Twohy and Anderson 2008; Twohy et al., 2009; Matsuki et al., 2010), despite sulfate and nitrate partly contribution from in-cloud production. This type contributed 2.9% by number of the total cloud residues (Figure S3). A slightly increase in Nf of the Dust residues was observed in size range above 0.5 μm relative to below 0.5 μm (3.0% versus 1.0%). At Mt. Tai in northern China, a high concentration of Ca²⁺ in cloud/fog water was mainly attributed to a sandstorm event during spring season (Wang et al., 2011). At Mt.

Heng in southern China, abundant crust-related elements (e.g., Al) observed in cloud water is due to Asian dust storms occurred on March-May (Li et al., 2017). Based on the backward trajectory, the site was less affected by sandstorm source in northwestern China during cloud events. Local dust emission by anthropogenic-disturbing soils or removing vegetation cover can be excluded as a result of forest protection. Additionally, dust residues may occupied larger CCN (Tang et al., 2016), which cannot be detected by the SPAMS. Therefore, a low fraction (2.9% by number) of dust cloud residue is reasonable in the present study.

The Fe particles had its typical ions at m/z $56Fe^+$ and internally mixed with sulfate and nitrate. The Fe particle type made up 4.1% by number of the total cloud residues. Approximately 16% of the Fe cloud residues contained $Ca^+$ peak (m/z 40). Relatively weak $Na^+$ and $K^+$ peaks in the Fe particles possibly contributes to anthropogenic sources (Zhang et al., 2014), especially northern air mass across iron/steel industrial activities in Yangtze River Mid-Reaches city clusters (Figure 2). These might suggest that the Fe residues was likely to have come from mixed sources. The presence of Fe in the cloud droplets play an important role in aqueous-phase $SO_2$ catalytic oxidation in cloud processing (Harris et al., 2013), thus accelerating the sulfate content of Fe-containing particles in cloud processing.

The Na-rich particles were mainly composed of ion peaks at m/z $23Na^+$ and $39K^+$ in the positive mass spectra, and nitrate and sulfate species in the negative mass spectra. The Na-rich particle type made up 3.0% by number of the total cloud residues. Na-rich particles were resulted from varied sources including industrial emissions, sea salt or dry lake beds (Moffet et al. 2008). The Nf of the Na-rich cloud residues did not increase from continental (Northerly) air mass to maritime (southwesterly) air mass on 21 Jan (3.3% versus 2.4% by number). However, related sea salt ion peak area (m/z, 81/83 $Na_2{}^{35}Cl/Na_2{}^{37}Cl$) were enhanced for Na-rich particles origination from maritime air mass relative to continental air mass (3.8 ±2.4 times). Continental air masses crossed industrial areas where located in the Yangtze River Mid-Reaches city cluster (Figure 2). Industrial emissions was a possible contributor to Na-rich particles under the influence of continental air masses (Wang et al. 2016). This might suggests that the Na-rich particles were contributed from both the industrial emissions and sea salt. Therefore, under the influence of maritime air mass, the signals for sea salt contribution became obvious.

The OC, Pb and Other particle types contributed 0.1-2.3% by number to the total cloud residues (Figure S3). Their average mass spectra can be found in Figure S4. The OC particles presented dominant intense OC signals (e.g., m/z $27C_2H_3{}^+$, $37C_3H{}^+$, $43C_2H_3O{}^+$ and $51C_4H_3{}^+$) and abundant sulfate. Presence of $K^+$ signal was found in the OC particles suggesting possible biomass burning sources (Bi et al. 2011). The Pb particles showed its typical ions at m/z $208Pb{}^+$ and internally mixed with $K^+$ and $Cl^-$. Previous studies found that K and Cl internally mixed with Pb particles have a possible origination of waste incineration (Zhang et al., 2009) or iron and steel facility (Tsai et al., 2007). Internally mixed EC with metal signatures was observed in the Other particles.

Previous measurements found that dust, playa salts or sea salt particles are often enriched in larger cloud droplets (~20 μm) (Bator and Collett, 1997; Pratt et al., 2010b). Organic carbon tend to be enriched in small cloud/fog droplets, extending to 4 μm (Herckes et al., 2013). It is wealth to note that cloud droplets were above 8 μm in the present study. Thus, it partially leads to relatively larger fractions of the Dust and Na-rich cloud residues observed, and the less fractions of the OC cloud residues.

**3.3 Mixing state of secondary species in cloud residues**

The Nf of sulfate-containing particles were found to be highly related to the K-rich (91%), OC (100%), aged EC (98%), Pb (74%), Fe (93%) and Amine (99%) cloud residues, as shown in Figure 5. Lower Nf of sulfate-containing particles were observed in the Na-rich (41%) and Dust (42%) cloud residues. In contrast, nitrate-containing particles contributed 89% and 88% by number to the Na-rich and Dust cloud residues, respectively. The heterogeneous chemistry of $HNO_3$ in the Na-rich and Dust particles may lead to the preferential enrichment of nitrate (Li and Shao, 2009). Note that after activation, uptake of gas-phase $HNO_3$ would increase nitrate level in the cloud residues (Schneider et al., 2017). The detection of nitrate in the cloud residues was thought to be the form of ammonium-nitrate by estimating the ratio of m/z 30 to m/z 46 in AMS data (Drewnick et al., 2007; Hayden et al., 2008). Low portions of ammonium (m/z, $18NH_4^+$) in the Na-rich (23% by number) and Dust (15% by number) cloud residues suggest that in this region, ammonium nitrate was not a predominant form of nitrate in the two cloud residual type. The Na-rich and Dust types were mainly composed of alkaline ion peaks (m/z, $23Na^+$, $39K^+$ and $40Ca^+$) in the position mass spectra (Figure 3), accompanied with larger fraction (88-89%) of nitrate. It suggests that nitrate might exist in the form of $Ca(NO_3)_2$, $NaNO_3$ or $KNO_3$ in the Dust and Na-rich cloud residues. It should be noted that the evaporation chamber of the GCVI may lead to a reduction of ammonium nitrate in the cloud residues (Hayden et al., 2008). We found that nitrate-containing particles accounted for only 46% by number of the aged EC cloud residues, which is significantly less than the contribution of sulfate-containing particles. Previous studies found that aged EC (soot) fog/cloud residues are mainly internally mixed with sulfate (Pratt et al., 2010a; Harris et al., 2014; Bi et al., 2016).

Aged EC particles mixed with sulfate are good CCN, rather than formed by in-cloud processing (Bi et al., 2016; Roth et al., 2016). High portions (75-86% by number) of ammonium-containing particles were observed for the OC and aged EC cloud residues, suggesting that ammonium will mostly be in the form of ammonium sulfate or ammonium nitrate for the two cloud residue types (Zhang et al., 2017). This result also implies that ammonium-containing particles are preferentially activated or enhanced by uptake of gaseous $NH_3$ to neutralize acidic cloud droplets for the OC and EC types.

Organics (e.g., amine and oxalate) have previously been measured in cloud water/residues (Sellegri et al., 2003; Sorooshian et al., 2007a; Pratt et al., 2010a). Amine and oxalate particles with mixtures of inorganic salts could enhance water uptake behavior (Sorooshian et al., 2008; Wu et al., 2011). The presence of TMA in the Amine cloud residues is expected to promote water uptake in sub- and supersaturated regimes (Sorooshian et al., 2007b). A total of 3,410 oxalate-containing particles (m/z, $-89HC_2O_4^-$)

represented 14.4% by number of the total cloud residues, which was mainly associated with the K-rich cloud residues (~70% by number). Note that after activation, gas phase partitioning into condensed phase or in-cloud production pathways would increase oxalate level in cloud droplets (Sellegri et al., 2003; Pratt et al., 2010a). Relative high portions (~30%

by number) of oxalate-containing particles in the metal (Pb, Fe) cloud residues might be the form of metal oxalate complexes from reactions of in-cloud formation oxalate with metals (Furukawa and Takahashi, 2011). Oxalate can readily partition into the particle phase to form amine salts (Pratt et al., 2009), it may facilitate the entrainment of oxalate (33% by number) in the Amine residues. A low fraction (4%) of oxalate-containing particles in the OC type is a result of restrictive classification. Classification of the OC particles mainly based on intense organic carbon ion signals (e.g., m/z $27C_2H_3^+$, $37C_3H^+$, $43C_2H_3O^+$ and $51C_4H_3^+$). However, majority of oxalate-containing particles internally mixed with the K-rich type. Therefore, oxalate was classified to the K-rich type, probably contributed from biomass burning. Noted that K-rich could contain a large abundant of organics (Pratt et al. 2011), however, the signals of organics were covered by the potassium due to its high sensitive to the laser.

**3.4 Comparison of cloud residues in different air mass sources**

Figure 6 displays hourly detected particle counts and Nf values of nine types of cloud residues and ambient particles. A very abrupt increase (decrease) in Nf of aged EC (Amine) particle types from cloud residues to ambient particles was observed on Jan 17. Ambient RH showed an abrupt decrease from nearly 100% at 10:00 to 85% at 11:00 on 17 Jan (Figure 1). Ambient temperature also decreased from 10 ℃ at 11:00 to 4 ℃ at 18:00 on 17 Jan (Figure 1). These changes imply that the air mass changed from southwesterly cloudy air to northerly cloud-free air around noon on 17 Jan (Figure 2). The entrained particles originated from northern air mass might have insufficient supersaturation to activate as cloud droplets. It is the reason that Nf of particle types abruptly varied from cloud residues to ambient particles on Jan 17 (Figure 6).

Ambient RH increased from 60% at 19:00 to nearly 100% at 21:00 on 18 Jan (Figure 1). Ambient temperature also increased from 1.3 ℃ at 22:00 on 18 Jan to 3.2 ℃ at 06:00 on 19 Jan (Figure 1). These changes imply that the air mass changed from northerly cloud-free air to southwesterly cloudy air at night on 18 Jan (Figure 2). During 18-19 Jan, the cloud residues and ambient particles showed similar chemical characteristics and were dominated by aged EC particles (Figure 6). A lack of significant variation in the Nf of particle types for this period suggests that nuclei particles originated from northerly cloud- free air could be activated to become cloud droplets. Note that ambient particles when a cloud-free event occurred at 11:00-17:00 on 19 Jan with a remaining high level of $PM_{2.5}$

($\sim$ 22.7 $\mu$g m$^{-3}$). Southwesterly wind flow on 19-20 Jan was too weak ($\sim$ 2.75 m s$^{-1}$) to dilute particles originated from northerly air masses (Figure 1). Additionally, high RH

(90%) air mass at height 1,500 m (a.s.l.) gradually moved to north China from 19 to 20 Jan (Figure S5). These might lead to similar residual particle types observed from 19 Jan to 20

Jan, although the site encountered southwesterly cloudy air on 19-20 Jan (Figure 2).

During 16-17 and 21-22 Jan, the cloud residues consisted of a high fraction of the

Amine type, which significantly differed from the observation during 18-19 Jan. Clearly, the observations during 16-17 and 21-22 Jan were influenced by a strong southwesterly flow with a low value of $PM_{2.5}$ ($\sim$ 3 $\mu$g m$^{-3}$).

As mentioned above, the Nf of the cloud residue types significantly changed as the air mass origin varied from northerly to southwesterly. To further investigate the influence of air mass history, we selected cloud residues that had arrived from a northerly air mass on

18-19 Jan and compared these to cloud residues originating from a southwesterly air mass during the periods of 16-17 and 21-22 Jan. The detected number of cloud residues for the northerly and southwesterly air masses are given in Table S1. Note that southwesterly air mass accompanied by high relative humidity (90%) (Figure S5) may have triggered particles activated to CCN prior to their arrival to the sampling site.

The K-rich type was found to contribute 23.9% to the cloud residues in the northerly air mass, which was significantly lower than its contribution to the southwesterly air mass (51.5%), as shown in Figure 7. A similarity in averaged mass spectrum of the K-rich residues was found for the southwesterly and northerly air masses (Figure S6). The considerable increase of K-rich cloud residues suggests a major influence of regional biomass-burning activities. Biomass-burning emissions from Southeast Asia, including

Myanmar, Vietnam, Laos and Thailand, where abundant fire dots are observed (Figure 2), could have been transported to the sampling site under a southwesterly air mass (Duncan et al., 2003). In contrast, the aged EC type represented only 23.7% of the cloud residues under the influence of the southwesterly air mass, which was significantly lower than observations for the northerly air mass (59.9%), as shown in Figure 7. This result suggests that the northern air mass has a greater influence on the presence of aged EC cloud residues.

In addition, an obvious increase in Nf of the Amine type was observed in the southwesterly air mass (15.1%) compared to the northerly air mass (0.2%), as shown in

Figure 7. This implies that the sources or formation mechanisms of amine in cloud residues varied in different air masses. The southwesterly air mass arrived from as far as the Bay of

Bengal and then travelled through Southeast Asia before reaching South China (Figure 2).

The potential gas amine emissions from ocean (Facchini et al., 2008) and livestock areas (90 million animals, data was available at the website http://faostat3.fao.org) in Southeast

Asia might promote the enrichment of amine particles. Note that after activation, the partitioning of the gas amine on cloud droplets may further contribute to the enhanced

Amine cloud residues (Rehbein et al., 2011), especially for air masses delivered via routes with high relative humidity, as mentioned above (Figure S5). In contrast, northerly air mass accompanied with dry airstreams may inadequately induce the partitioning of gas amines into the particle phase (Rehbein et al., 2011).

**3.5 Comparison of cloud residues with ambient and non-activated particles**

A direct comparison between cloud residues and ambient particles was limited because of their differences in air mass origins. During the sampling period, the cloud events occurred once the southwesterly air masses were dominant. Therefore, a comparison between cloud residues and ambient particles cannot be addressed under the influence of southwesterly air masses. Here, we chose five hours before and after the beginning of the cloud II period in order to compare cloud residues and ambient particles with similar northerly air mass origins, as discussion in Sect. 3.4. The time and detected counts of cloud residues and ambient particles for this comparison are listed in Table S1.

From 10:00 21 Jan to 13:00 23 Jan, cloud residues and non-activated particles were alternately sampled with interval of one hour. Ambient temperature decreased from 6 ℃

at 11:00 to 0 ℃ at 23:00 on 22 Jan (Figure 1). Additionally, ambient particles level (residual and non-activated particles) showed a clearly increase from 121 cm$^{-3}$ to 1339

cm$^{-3}$ during this period (Figure S2). It suggests that initial mixing of northerly cloud-free air and southwesterly cloudy air around noon on 22 Jan. It is noted that non-activated particles were detected in size range above 200 nm, extending to 500 nm (Figure S7). The dry northern air mass might lower supersaturation, only larger particles could be activated.

This might result in above 200 nm non-activated particles observed here (Mertes et al.,

2005; Kleinman et al., 2012; Hammer et al., 2014). The detected particle counts in the cloud residues and non-activated particles are given in Table S1.

The contribution of K-rich particles in cloud residues slightly decreased relative to ambient particles (23.9% versus 30.7%), as shown in Figure 7. Previous studies also showed that no significant change in Nf of biomass-burning particles for cloud residues relative to ambient particles (Pratt et al., 2010a; Roth et al., 2016). The biomass-burning particles internally mixed with soluble species (e.g., sulfate, nitrate and oxalate) enhanced their ability to act as CCN, as discussion in Sect. 3.3. However, Kamphus et al. (2010)

reported that biomass-burning particles account for only 3% of cloud residues compared with 43% of ambient particles, and they suspected that biomass-burning particles might exist in the form of tar balls (hydrophobic materials). A slight increase in Nf of the aged

EC cloud residues was observed relative to ambient particles (59.9% versus 53.8%), as shown in Figure 7.  In general, freshly emitted EC particles are less hydrophilic and do not active as CCN (Bond et al., 2013). The aged EC particles show a high degree of internal mixing with secondary inorganic compounds in this study (Figure 5), improving their ability to act as CCN. The remaining particle types showed no clear differences in Nf between cloud residues and ambient particles.

In comparing the cloud residues with non-activated particles, a significant change in Nf was found for the aged EC and K-rich type. A higher Nf of K-rich particles and a lower Nf of EC particles were found for the cloud residues relative to the non-activated particles (Figure 7). Entrainment of northerly cloud-free air might lower supersaturation during this period. Aged EC particles may require very high supersaturation to grow into cloud droplets and thus, only form hydrated non-activated aerosol (Hallberg et al., 1994).

Figure 8 and 9 show differences in average mass spectra for cloud residues versus ambient particles, as well as cloud residues versus non-activated particles, respectively.

Nitrate intensity (average ion peak area) was found to enhance in the cloud residues compared to ambient particles. In addition, nitrate-containing particles has been observed to account for 70% of the cloud residues compared to 38% of the ambient particles. Drewnick et al. (2007) suggested that rather than sulfate, high nitrate content in pre-existing particles preferentially acted as cloud droplets. Compared with containing-nitrate ambient particles, larger size of containing-nitrate residues (Figure S8) is more likely to be uptake of gaseous $HNO_3$ during cloud process (Hayden et al. 2008; Roth et al., 2016). A recent study also confirmed that uptake of gaseous $HNO_3$ was an important contributor for increasing in nitrate level in the cloud residues (Schneider et al., 2017). Interestingly, we observed a decrease in nitrate intensity in cloud residues except dust type (Figure 9) and a large size distribution of nitrate-containing cloud residues (Figure S7) compared with non-activated particles. This result suggests that particle size, rather than nitrate content, plays a more important role in the activation of particles into cloud droplets.

Sulfate intensity was only observed to enhance for the OC cloud residues relative to both ambient and non-activated particles. Although the in-cloud addition of sulfate can be produced from aqueous Fe-catalyzed or oxidation by $H_2O_2/O_3$ reactions (Harris et al., 2014), sulfate intensity was found to diminish in the Fe cloud residues relative to ambient particles. Compared with non-activated particles, sulfate intensity was found to enhance in the Fe cloud residues. Additionally, sulfate-containing particles accounted for 94%, 93% and 94% of cloud residues, ambient and non-activated particles, respectively. Previous studies also showed that the mass or number fraction of sulfate-containing particles in the cloud residues changed between ambient and non-activated particles (Drewnick et al., 2007; Twohy and Anderson, 2008; Schneider et al., 2017). However, the reason for these changes remains unclear.

The in-cloud process has been reported to be an important pathway for the production of amine particles (Rehbein et al., 2011; Zhang et al., 2012a). In this study, no remarkable change in Nf of the Amine cloud residues was obtained relative to the ambient particles (0.2% versus 0.2%), as shown in Figure 7. Bi et al. (2016) considered that the absence of amine species in fog residues may be partially affected by droplet evaporation in the GCVI.

We did find a high fraction of the Amine cloud residues when the southwesterly air mass prevailed, as discussion in Sect. 3.4. A lack of gas-phase amines may be the cause of few amine particles detected in the ambient particles and cloud residues (Rehbein et al., 2011).

An increase in Nf of cloud residues was observed compared with non-activated particles (5.2% versus 0.1%), as shown in Figure 7. Increasing the particle water content facilitates partitioning of gas-phase amine species into the aqueous phase when gas-phase amines present (Rehbein et al., 2011).

**523 4 Conclusions**

This study presented an in situ observation of individual cloud residues, non-activated and ambient particles at a mountain site in South China. The finding shows that internal mixing with soluble species (e.g., sulfate) in EC particles was an important contributor to cloud residues in a remote area of China. Change in Nf of the cloud residue types influenced by various air masses highlights the important role of regional transportation in the observed cloud residual chemistry. Initial mixing of northerly cloud-free air and southwesterly cloudy air can induce the activation of the nuclei particles to become cloud droplets. Higher fractions of nitrate (88-89% by number) were found in the Dust and Na-rich cloud residues relative to sulfate (41-42%) and ammonium (15-23%). Higher fraction, intensity (average ion peak area) and larger size of nitrate-containing particles were found in the cloud residues relative to the ambient particles. This result is most likely the cause of the uptake from gas-phase $HNO_3$.

**Acknowledgments**

This work was supported by the National Nature Science Foundation of China (No.

91544101 and 41405131), the National Key Research and Development Program of China (SQ2017ZY01014804) and the Foundation for Leading Talents of the Guangdong

Province Government. The authors thank Ji Ou from Shaoguan city Environmental

[revised manuscript text omitted]

---

## Referee Report (RR1)

Review of "In situ chemical composition measurement of individual cloud residue particles at a mountain site, South China (revised version)" by Lin et al.

**Summary and general comments**

The authors present the results of online physico-chemical characterization of cloud residuals (>~8 µm in diameter), cloud particles (<2.5 µm $d_a$) and ambient particles (<2.5 µm $d_a$) from the field study conducted in January, 2016 in a remote mountainous area of South China. Based on the dataset of the eight major particle types classified by using the SPAMS data obtained during several cloud events, the authors report some unique properties of cloud residuals, such as chemical composition, size and mixing state (that may be inherently related), presumably occurred during an aerosol's atmospheric aging and cloud processes.

The topic itself is an important addition to ACP. Overall, the authors conducted a careful study as well as rigorous data analyses to generate new results regarding cloud residuals that would be potentially valuable in the cloud microphysics research community. However, such care was unfortunately not taken in the preparation of the manuscript, with the manuscript containing a number of errors and typos. Although I do not have any major scientific concerns, I have numerous technical comments (including but not limited to) as listed below. I would urge the authors of the manuscript to thoroughly proof read their manuscript as this list gets too long.

**Specific comments**

P4 L64-P5L78: This part is not well written and poorly structured. The authors need to logically address why it is particularly important to study the aerosol mixing state rather than just focus on other general properties, such as size (e.g., Dusek et al., 2006, Science; Sotiropoulou et al 2006, Aerosol Science & Technology) and bulk composition (e.g., Wiedensohler et al., 2009, JGR; Twohy and Anderson, 2008, Environmental Research Letters), to improve our understanding of CCN activation. Aerosol mixing state indeed influences the ability of aerosol to act as CCN (e.g., Wang et al., 2010, Aerosol Science & Technology). For instance, Medina et al. (2007, JGR) estimated that internal mixing assumption resulted in a 35% over prediction of CCN concentration in one study for semi-urban settings. However, the relative importance of the mixing state as compared to other properties appear to vary depending on the proximity to the pollution plume source and/or photochemical ageing activity as a function of oxidant concentrations (Ervens et al., 2010, ACP). Hence, additional detailed measurements to characterize the timescale and effect of the aeorosl mixing state on CCN properties of particles are by all means needed to improve our theoretical understanding of CCN activation. Please do not copy and paste the reviewer's comments in the manuscript. The authors may want to do a cereful and through literature review, digest the contents in a diplomatic manner and describe your thoughts to the reader along with your own story line. I also suggest the authors to address up-to-date information of lab studies regarding the effect of the aerosol mixing state on CCN ability or droplet activation (e.g., Wang et al., 2017, ACPD-2017-454 and references therein; Broekhuizen et al., 2004, GRL; Abbatt et al., 2005, Atmospheric Environment; Shilling et al., 2007, Journal of Physical Chemistry A). Explaining how your field study would potentially shed light on lab works, vice versa, may strengthen the paper.

P7L125-126: The authors state that the observed clouds contained only liquid droplets. This statement is speculative since the ambient temperature seemed to be below -7 °C at some point of the field campaign (P7L125), which could trigger heterogeneous ice nucleation of some materials contained in dust aerosols (e.g., Atkinson et al., 2013, Nature). Further scientific backing seems necessary for the reader to understand the cloud properties.

P7 Ll26-128: Was the ambient inlet coupled with a dryer downstream? Please clarify.

P7L130-131: "Therefore, it is reasonable to select..." - I desagree with this statement. Correctly, the authors presumed the droplet size to be larger than 8 µm. This assumption should be stated in the manuscript.

P7L130-131: There have been observations of droplet size-depndant chemical composition in clouds (Moore et al., 2004, Atmospheric Environment). Please discuss it in the manuscript.

P7131-134: What gas (i.e., dry synthertic air, nitrogen, etc.) was used to create the counterflow? What is the background aerosol concentration through GCVI (i.e., the measurement with the counterflow only) in this study? In addition, can the authors at least provide the estimate of the number fraction of residuals to total particles relevant to your study (i.e., CCN active fraction)? Did the author measure the total cloud particle concentration through GCVI without any counterflows at some point?

P7L134-135: The particle transmission efficiency of residual sampling instruments is size-dependant. Did the authors take it into account for your analyses or apply 50% loss throughout the analyses? Please clarify. It is not clear if the size-dependence is incorporated in your analyses (P11L225-228). Just citing a paer seems not enogh to justify it.

P7L139: Why do the authors define what comes through an ambient inlet (2.5 micron 50% cut-off) as "non-activated" particles? Numerous lab and fild studies show that submicron particles can be activated to <2.5 micron.

P9L165-166: What about the transmission efficiency? Was it accounted, too? Please clarify.

P10L197: cloud formation → high RH condition. Unless the authors provide the data of cloud properties, it is not fair to say clound formation.

P12L244-246: The local biomass burning also contributes to the aerosol-cloud interaction in the North Slope of Alaska (Hiranuma et al., 2013, JGR), which seems more relevant to your study.

P13L257-259: So how did this 266 nm enrichment influece your own results? Please crarify.

P13L263-267: So what determines the ambient abundance of amines in this particular study? Please be conclusive.

P13L268-269: What is the implication of Bi et al. with respect to to your study? Please be conclusive.

P14L286-287: The authors may state that a low fraction of dust is the limitation/artifact.

P14L288-296: Moteki et al. (2017, Nature Comm.) reports the aircraft onservation of magnetite (up to ~1 cm$^{-3}$) over the East Asia. The authors may read, digest and incorporate it in your manuscript.

P14L299-P15L310: Was a depletion of chloride (e.g., Laskin et al., 2012, JGR) dominant in this study?

P15L308-309: Can the authors quantitatively differenciate the source (industrial vs. maritime) by looking at other types of particles came along with the Na-rich particles?

P15L311-319: Just for curiosity, did the authors find any biological particles during the campaign? If so, how many of those are classified as Other?

P17L353-354: In P16L338-340, the authors state that the ammonium niotrate is not a dominant form of nitrate in this study, which seems contradicting to the statement given here...

P17L360-361: What is the measuremnt uncertainty regarding TMA counts?

P21L457-458: So size or mixing state – which factor was determinant to determine the cloud formation ability in this particular period?

**Technical comments**
P3L45 & P4L57: Be consistent with 'in situ' or 'in-situ'.
P4L56: → …and, in turn, affect…
P4L59: Although → Despite or Even with
P4L64: The formation of → The ability of aerosols to act as
P4L71 Too many 'however's are bothering. There are a total of 10 however-sentences appearing in this manuscript.
P5L79: → an Aerosol Mass Spectrometer (AMS) or other online/offline signle particle instruments is
P5L87: Oceans sound awkward.
P5L88: Srat a new paragraph.
P5L90-93: → Although scientists have worked to...in China (Zhang et al., 2012b), only few studies have employed...
P5L94: → obtain the mixing state of individual ambient particles during
P5L95: → Their results showed...large particles
P6L98: → fog residual particles at ground level in an urban area of South China.
P6L99: → They found an abundance of anthropogenic particles, including...
P6L102: → a mountain site in South China
P6L113: → Our measurements were carried out during...
P6L115: → This station is located at 200 km...
P6L117: → ...(273 km$^2$), where...
P7L124: → The ambient temperature
P7L126: → The measurements of the droplet size spectra in this resion prerformed during the winter of...
P7L129: Some → Previous
P8L151: each singe → single (or individual particles)
P9 170-171: Providing references for these two sentences would be nice.
P9L169-170: → Low levels of...exclude
P9L175: 73,996 – be consiste with the use of "," to describe numbers throughout the manuscript.
P9L180: → similar clusters, such as aged EC, ...
P9L184: → Assuming that the number of...
P10L198-199: I suggest delting two 'and's
P10L200: → Note that, on...
P11L211: The word "big freeze" is a nomencreature that may refer to something else. I suggest rewording it.
P11L216: → the main six particle types
P11L221: → The strong $K^+$ ion...
P11L221-222: Awkward/incomplete sentence - I suggest rephrazing the sentence.
P11L227: → suffer from the bias related to...
P11L232: → ...(m/z -46NO2-, -62NO3-) and presumably derived from...
P12L234: → An aged time of 81-88 min...showed an increase...
P12L246: → the majority of aged EC
P12L246: → Asian
P12L249-250: → ...particles were only...droplets, and the aged EC residuals were...
P12L251-252: → The Jungraujoch station is predoiminantly within the free trophospheric condition, such that the biomass...
P13L262: → the size range
P13L263: → Aqueous reactions improving ... have been observed...
P13L267: → ...amine wiithin the cloud.

P13L273: → Previous studies showed that dust particles that are internally mixed with sulfate and nitrate promote CCN activities...

P13L275: partly → partial

P13L276: A slightly increase → A slight increase

P13L273: Internal mixing ... is expect to act as CCN... → Dust particles that are internally mixed with sulfate and nitrate are expected to act as CCN...

P13L279: → during the spring season

P14L281: → Asian dust storms that occured in March-May

P14L283: → Local dust emissions

P14L289 → and nitrate, making up 4.1%

P14L291: contributes → contribute

P14L293-294: → ...Fe-containing residuals have presumably come from...

P14L285: may occupied → may have occupied

P14L293: → Fe-containing residuals were

P14L299-300: → ...Na-rich particles are formed from varied sources...

P15L305: → The continental air masses

P15L306: → Industrial emissions were

P15L308: → This might suggest that the Na-rich particles were contributed from both industrial emissions and sea salts.

P15L318: → iron and steel products manufacturing facilities

P15L322: → Organic carbon tends to be...

P15L323: wealth → worth (I am not a big fun of too many "note" phrases. There are 13 notes in this manuscript, which seems a lot. Consider making smoother transitions and better flows between sentences/paragraphs without using too many notes).

P16L327: secondary inorganic species?

P16L336: → to be in the form of

P16L338: Low as compared to what?

P16L342: → in mass spectra (Figure 3)...

P16L343: → Thus, our data suggest that...

P16L340: → in two cloud residual types

P16L346: → In this study, we found that...

P16L346: → The data indicate that nitrate-containing particles account for...

P17L366: → Relatively high portions...

P17L368: → in the form of....

P17L370: → ...(Pratt et al., 2009). It may...

P18L373: → ...is mainly based on...

P18L374: → particles were internally...

P18L375: → ...to the K-rich type and probably...

P18L378: sensitive → sensitivity

P18L381: → Figure 6 displays the hourly... and Nf values of the nine types of...

P18L382-383: Awkward sentence – please rephrase.

P18L386: changed → shifted

P19L398-400: Incomplete sentence.

P19L400: → The southwesterly...

P19L401: → from the northerly

P19L402: → North China or northern China (the authors use the South China word consistetly in the manuscript... why not for North???)

P19L403: → These changes might have led...

P19L411-412: → we selected to analyze cloud residuals that...on 18-19 Jan as compared to cloud residuals that...

P20L420: → for both the...

P20L431: → This data implies...
P21L453: → ...with one hour intervals. The ambient...
P21L456: → Thus, the data suggest that the initial...
P21L457: → cloudy air occured around
P21L458: → in the size range of 200 nm up to 500 nm
P21L458-461: Awkward/imcomplete sentences.
P22L465: → showed that there were no significant changes
P22L468: → as discussed in Sect. 3.3.
P22L473-474: do not active as CCN → are not active as CCN
P22L478: In comparing → When comparing
P22L478-479: has been observed to account for → accounted for
P22L484: → the differences
P23L487: → when compared to
P23L490: → nitrate-containing
P23L491: is most likely to be → possibly reflect the
P23L493: → confirmed that the update of gaseous $HNO_3$ is an...
P23L494: → the increased nitrate level
P23L495: → (Figure 9), and...
P23L495: → in the cloud residuals
P23L496: → when compared with
P24L515: We did find → There was a high fraction... amine cloud residuals found when the...
P24L527: → The change observed in Nf
P24L530: induce the activation of the nuclei particles to become cloud droplets → act as CCN
Figure 2: yellow rots and fire dots sound awkward...
Figure 3: 'of the sampled particles during the whole sampling period' – not necessary.
Figure 7: → with an interval of one hour
Figure 9: → ; wherears,

---

## Author Response (AR2)

**Response to comments**

Manuscript Number: acp-2017-23

Title: In situ chemical measurement of individual cloud residue particles at a mountain site, South China. Qinhao Lin et al.

**Summary and general comments**

*The authors present the results of online physico-chemical characterization of cloud residuals (>~8 μm in diameter), cloud particles (<2.5 μm da) and ambient particles (<2.5 μm da) from the field study conducted in January, 2016 in a remote mountainous area of South China. Based on the dataset of the eight major particle types classified by using the SPAMS data obtained during several cloud events, the authors report some unique properties of cloud residuals, such as chemical composition, size and mixing state (that may be inherently related), presumably occurred during an aerosol's atmospheric aging and cloud processes.*

*The topic itself is an important addition to ACP. Overall, the authors conducted a careful study as well as rigorous data analyses to generate new results regarding cloud residuals that would be potentially valuable in the cloud microphysics research community. However, such care was unfortunately not taken in the preparation of the manuscript, with the manuscript containing a number of errors and typos. Although I do not have any major scientific concerns, I have numerous technical comments (including but not limited to) as listed below. I would urge the authors of the manuscript to thoroughly proof read their manuscript as this list gets too long.*

We would like to thank the reviewer for his/her useful comments and recommendations to improve the manuscript. We agree with the comments, and careful revision has been made according to the suggestions.

**Specific comments**

*P4 L64-P5L78: This part is not well written and poorly structured. The authors need to logically address why it is particularly important to study the aerosol mixing state rather than just focus on other general properties, such as size (e.g., Dusek et al., 2006, Science; Sotiropoulou et al 2006, Aerosol Science & Technology) and bulk composition (e.g., Wiedensohler et al., 2009, JGR; Twohy and Anderson, 2008, Environmental Research Letters), to improve our understanding of CCN activation. Aerosol mixing state indeed influences the ability of aerosol to act as CCN (e.g., Wang et al., 2010, Aerosol Science & Technology). For instance, Medina et al. (2007, JGR) estimated that internal mixing assumption resulted in a 35% over prediction of CCN concentration in one study for semi-urban settings. However, the relative importance of the mixing state as compared to other properties appear to vary depending on the proximity to the pollution plume source and/or photochemical ageing activity as a function of oxidant concentrations (Ervens et al., 2010, ACP). Hence, additional detailed measurements to characterize the timescale and effect of the aerosol mixing state on CCN properties of particles are by all means needed to improve our theoretical understanding of CCN*

*activation. Please do not copy and paste the reviewer's comments in the manuscript. The authors may want to do a careful and through literature review, digest the contents in a diplomatic manner and describe your thoughts to the reader along with your own story line. I also suggest the authors to address up-to-date information of lab studies regarding the effect of the aerosol mixing state on CCN ability or droplet activation (e.g., Wang et al., 2017, ACPD-2017-454 and references therein; Broekhuizen et al., 2004, GRL; Abbatt et al., 2005, Atmospheric Environment; Shilling et al., 2007, Journal of Physical Chemistry A). Explaining how your field study would potentially shed light on lab works, vice versa, may strengthen the paper.*

The part has been restructured. The ability of aerosol particles to act as cloud condensation nuclei (CCN) is dependent on the size and chemical composition of atmospheric aerosol particles at a given supersaturation (McFiggans et al., 2006). Wiedensohler et al. (2009) found that the enhancement of particles CCN ability was related to an increase in the average sulfate mass concentration. Dusek et al. (2006) demonstrated that CCN behavior was more effected by aerosol size than chemical composition. Meanwhile, aerosol mixing state also play an important role in the ability of aerosol to act as CCN. It has been reported that freshly emitted elemental carbon (EC) particles generally exhibit low CCN activity, whereas aged EC particles show high CCN activity after experienced atmospheric processes (Zhang et al., 2008). Pratt et al. (2011) found that number fractions of ammonium or oxalate internally mixed with biomass burning particles increased with an aged time of 81-88 min, which promote CCN behavior. Laboratory studies have shown that low-solubility organic particles internally mixed with ammonium sulfate would suppress water uptake of mixed particle and thus might affect CCN activity (Wise et al., 2003; Svenningsson et al., 2006; Sjogren et al., 2007). An over prediction of CCN concentration by up to 35% was estimated based on particle internal mixing state assumption (Medina et al., 2007; Collins et al., 2013). The influence of mixing state on aerosol CCN activity varies depending on the proximity to the pollution plume source and/or photochemical ageing activity (Ervens and Volkamer, 2010). More detailed measurements to characterize the mixing state of CCN particles would improve our understanding of aerosol-cloud interactions. Our result was compared with previous lab measurements (McMeeking et al., 2011; Tang et al., 2016). Please refer to Lines 64-83, 356-358 and 375-378 of the revised manuscript.

*P7L125-126: The authors state that the observed clouds contained only liquid droplets. This statement is speculative since the ambient temperature seemed to be below -7 ℃ at some point of the field campaign (P7L125), which could trigger heterogeneous ice nucleation of some materials contained in dust aerosols (e.g., Atkinson et al., 2013, Nature). Further scientific backing seems necessary for the reader to understand the cloud properties.*

Only 20 cloud residues that accounted for 0.08% of the total cloud residues were detected when the ambient temperature was below -7 ℃ observed from 06:00 to 08:00

on 23 Jan. Thus, cloud droplets were dominated by liquid water droplets. Please refer to Lines 130-133 of the revised manuscript.

*P7 Ll26-128: Was the ambient inlet coupled with a dryer downstream? Please clarify.*

The ambient inlet was dried using a diffusion dryer. Please refer to Lines 148-149 of the revised manuscript.

*P7L130-131: "Therefore, it is reasonable to select..." - I disagree with this statement. Correctly, the authors presumed the droplet size to be larger than 8 μm. This assumption should be stated in the manuscript.*

"Therefore, it is reasonable to select..." assuming that size distribution of cloud droplet mostly was above 8 μm in this region". Please refer to Lines 137-138 of the revised manuscript.

*P7L130-131: There have been observations of droplet size-dependent chemical composition in clouds (Moore et al., 2004, Atmospheric Environment). Please discuss it in the manuscript.*

We have added the reference as "Previous measurements have found that dust, playa salts, sea salt or metal particles were often enriched in larger cloud droplets (~20 μm) (Bator and Collett, 1997; Moore et al., 2004; Pratt et al., 2010b).". Please refer to Lines 341-343 of the revised manuscript.

*P7131-134: What gas (i.e., dry synthertic air, nitrogen, etc.) was used to create the counterflow? What is the background aerosol concentration through GCVI (i.e., the measurement with the counterflow only) in this study? In addition, can the authors at least provide the estimate of the number fraction of residuals to total particles relevant to your study (i.e., CCN active fraction)? Did the author measure the total cloud particle concentration through GCVI without any counterflows at some point?*

A stream of filtered and heated ambient air (counterflow) was provided by a compressor. During cloud-free periods, a ratio of concentration (below 1 $cm^{-3}$) behind the CVI to background aerosol concentration (2,000 $cm^{-3}$) was 0.0005, indicating that instances of particle breakthrough and small particle contamination were absent. A ratio of number residuals to total number particles (sum of cloud residues and non-activated particle) on average was $0.43 \pm 0.20$, when cloud residues and non-activated particles were alternately sampled with an interval of one hour during the cloud III event. We have not measured the total cloud particle concentration through GCVI without any counterflows. Please refer to Lines 141-142, 149-152 and 179-182 of the revised manuscript.

*The particle transmission efficiency of residual sampling instruments is size-dependent.*

*Did the authors take it into account for your analyses or apply 50% loss throughout the analyses? Please clarify. It is not clear if the size-dependence is incorporated in your analyses (P11L225-228). Just citing a paper seems not enough to justify it.*

e have discussed effect of particle transmission efficiency of the SPAMS and GCVI on change in number fractions of cloud residual types. Dust residues may have occupied larger CCN (Tang et al., 2016) and OC particles existed in smaller cloud residues (Sellegri et al., 2003a), which cannot be detected by the SPAMS. This might lead to underestimate fractions of these particle types due to the limitation of the SPAMS. Moreover, the particle transmission efficiency of the GCVI increased with increasing cloud droplet size (Shingler et al., 2012), might leading to relatively larger fractions of the large cloud droplets. W Please refer to Lines 303-304, 332-334 and 345-348 of the revised manuscript.

*P7L139: Why do the authors define what comes through an ambient inlet (2.5 micron 50% cut-off) as "non-activated" particles? Numerous lab and field studies show that submicron particles can be activated to <2.5 micron.*

It is assumed that the mean diameter of cloud droplets was around 8 μm. It is possible that cloud droplet could have size < 2.5 μm. It is hard to define a critical fog/cloud droplet diameter that is highly variable (ranging from 1 to 5 μm) (Hammer et al., 2014). The "non-activated" (interstitial) particles inlet has a cutoff of 2.5 μm in previous work (Verheggen et al., 2007). Therefore, a similar cutoff was used to sample non-activated particles during the cloud events. Please refer to Lines 137-138 and 147-148 of the revised manuscript.

*P9L165-166: What about the transmission efficiency? Was it accounted, too? Please clarify.*

The measured cloud residual concentration was integrated by a SMPS and was then corrected by the enrichment factor and transmission efficiency of the GCVI. Please refer to Line 177 of the revised manuscript.

*P10L197: cloud formation → high RH condition. Unless the authors provide the data of cloud properties, it is not fair to say cloud formation.*

We agree with comment and have corrected accordingly. Please refer to Line 214 of the revised manuscript.

*P12L244-246: The local biomass burning also contributes to the aerosol-cloud interaction in the North Slope of Alaska (Hiranuma et al., 2013, JGR), which seems more relevant to your study.*

We have added the reference and have discussed contribution of local biomass burning to the aerosol-cloud interaction in the North Slope of Alaska. Please refer to Lines 263-264 of the revised manuscript.

*P13L257-259: So how did this 266 nm enrichment influence your own results? Please clarify.*

Amine peak areas would be enhanced when using a 266 nm ionization laser and amines themselves may not comprise the majority of the particle mass (Pratt et al., 2009). The SPAMS only provide a qualitative information and is not directly related to particle mass. Therefore, the sentence has been detected in the revised manuscript.

*P13L263-267: So what determines the ambient abundance of amines in this particular study? Please be conclusive.*

Presence of gas-phase amine sources determines the ambient abundance of amines. The potential gas amine emissions from ocean and livestock areas might promote the enrichment of amine particles in this study. Please refer to Lines 284-287 and 456-459 of the revised manuscript.

*P13L268-269: What is the implication of Bi et al. with respect to your study? Please be conclusive.*

It might suggests that enhancement of particle amine is not only depend on high RH or fog/cloud process, but also sensitive to other parameters, such as presence of gas phase amine source (Rehbein et al., 2011). Please refer to Lines 284-287 of the revised manuscript.

P14L286-287: The authors may state that a low fraction of dust is the limitation/artifact.

We agree with comment and have corrected accordingly as "Hence, a low fraction (2.9% by number) of dust cloud residue might be due to the limitation of the SPAMS." Please refer to Lines 303-304 of the revised manuscript.

*P14L288-296: Moteki et al. (2017, Nature Comm.) reports the aircraft observation of magnetite (up to ~1 cm$^{-3}$) over the East Asia. The authors may read, digest and incorporate it in your manuscript.*

We have cited the reference. The contribution of anthropogenic and natural Fe-containing particles sources (Moteki et al., 2017) to observed Fe-containing residues is discussed. Please refer to Lines 310-311 of the revised manuscript.

*P14L299-P15L310: Was a depletion of chloride (e.g., Laskin et al., 2012, JGR) dominant in this study?*

The acid displacement reactions of sea salt chloride with $HNO_3$ during atmospheric aged process might lead to a depletion of chloride (37%) and a high fraction (89%) of nitrate in the Na-rich residues. Please refer to Lines 354-356 of the revised manuscript.

*P15L308-309: Can the authors quantitatively differentiate the source (industrial vs. maritime) by looking at other types of particles came along with the Na-rich particles?*

The number fraction of the Na-rich cloud residues did not increase from continental (Northerly) air mass on 19 Jan to maritime (southwesterly) air mass on 21 Jan (3.3% versus 2.4% by number). It limits the analysis to quantitatively differentiate other types of particles came along with the Na-rich particles. Instead, a comparison of related sea salt ion peak area was performed for the Na-rich residues at varied air masses. Please refer to Lines 319-323 of the revised manuscript.

*P15L311-319: Just for curiosity, did the authors find any biological particles during the campaign? If so, how many of those are classified as Other?*

Only three particles in the Other type were found to contain calcium, organic carbon, organic nitrogen and phosphate ion signals, suggesting existence of biological particles (Pratt et al., 2009a). Please refer to Lines 337-340 of the revised manuscript.

[Figure]

*P17L353-354: In P16L338-340, the authors state that the ammonium nitrate is not a dominant form of nitrate in this study, which seems contradicting to the statement given here...*

Relative to nitrate, low portions of ammonium (m/z, $18NH_4^+$) in the Na-rich (23% by number) and Dust (15% by number) cloud residues suggest that in this region, ammonium nitrate was not a predominant form of nitrate in the Na-rich and Dust cloud residues. High portions (75-86% by number) of ammonium-containing particles were observed for the OC and aged EC cloud residues. This result implies that ammonium-containing particles are preferentially activated or enhanced by uptake of gaseous $NH_3$ to neutralize acidic cloud droplets for the OC and EC types. Please refer to Lines 362-364 and 378-381 of the revised manuscript.

*P17L360-361: What is the measurement uncertainty regarding TMA counts?*

The TMA accounted for up to 93% by number of the Amine cloud residues. Please refer to Line 386 and Figure 5 of the revised manuscript.

*P21L457-458: So size or mixing state-which factor was determinant to determine the cloud formation ability in this particular period?*

The enhancement on number fraction or intensity of nitrate-containing cloud residues was not observed when compared with non-activated particles during this particular period. On the contrary, a large size distribution of nitrate-containing cloud residues (Figure S7) was found when compared with non-activated particles. This result reflects that particle size, rather than mixing state/nitrate content, plays a more important role in the activation of particles into cloud droplets during this particular period. Please refer to Lines 515-519 of the revised manuscript.

**Technical comments**
*P3L45 & P4L57: Be consistent with 'in situ' or 'in-situ'.*

We have corrected accordingly. Please refer to Line 47 and 57 of the revised manuscript.

*P4L56: →…and, in turn, affect…*

We have corrected accordingly. Please refer to Line 56 of the revised manuscript.

*P4L59: Although → Despite or Even with*

We have corrected accordingly. Please refer to Line 60 of the revised manuscript.

*P4L64: The formation of → The ability of aerosols to act as*

We have corrected accordingly. Please refer to Line 64 of the revised manuscript.

*P4L71: Too many 'however's are bothering. There are a total of 10 however-sentences appearing in this manuscript.*

We have detected some 'however's in the revised manuscript.

*P5L79: → an Aerosol Mass Spectrometer (AMS) or other online/offline single particle instruments is*

We have corrected accordingly. Please refer to Lines 84-85 of the revised manuscript.

*P5L87: Oceans sound awkward.*

We have changed "Oceans" to "marine areas". Please refer to Line 93 of the revised manuscript.

*P5L88: Start a new paragraph.*

We have corrected accordingly. Please refer to Line 95 of the revised manuscript.

*P5L90-93: → Although scientists have worked to...in China (Zhang et al., 2012b), only few studies have employed...*

We have corrected accordingly. Please refer to Lines 96-99 of the revised manuscript.

*P5L94: → obtain the mixing state of individual ambient particles during*

We have corrected accordingly. Please refer to Lines 100-101 of the revised manuscript.

*P5L95: → Their results showed...large particles*

We have corrected accordingly. Please refer to Lines 101-102 of the revised manuscript.

*P6L98: → fog residual particles at ground level in an urban area of South China.*

We have corrected accordingly. Please refer to Line 105 of the revised manuscript.

*P6L99: → They found an abundance of anthropogenic particles, including...*

We have corrected accordingly. Please refer to Line 106 of the revised manuscript.

*P6L102: → a mountain site in South China*

We have corrected accordingly. Please refer to Line 108 of the revised manuscript.

*P6L113: → Our measurements were carried out during...*

We have corrected accordingly. Please refer to Line 118 of the revised manuscript.

*P6L115: → This station is located at 200 km...*

We have corrected accordingly. Please refer to Lines 120-121 of the revised manuscript.

*P6L117: → ...(273 km2), where...*

We have corrected accordingly. Please refer to Lines 122-123 of the revised manuscript.

*P7L124: →The ambient temperature*

We have corrected accordingly. Please refer to Line 129 of the revised manuscript.

*P7L126: → The measurements of the droplet size spectra in this region performed during the winter of...*

We have corrected accordingly. Please refer to Lines 133-134 of the revised manuscript.

*P7L129: Some → Previous*

We have corrected accordingly. Please refer to Line 136 of the revised manuscript.

*P8L151: each singe → single (or individual particles)*

We have corrected accordingly. Please refer to Line 162 of the revised manuscript.

*P9 170-171: Providing references for these two sentences would be nice.*

We have added a reference (Bi et al., 2016). Please refer to Line 186 of the revised manuscript.

*P9L169-170: → Low levels of...exclude*

We have corrected accordingly. Please refer to Line 183 of the revised manuscript.

*P9L175: 73,996 →be consistent with the use of "," to describe numbers throughout the manuscript.*

We have corrected accordingly. Please refer to Lines 190-192 of the revised manuscript.

*P9L180: → similar clusters, such as aged EC, ...*

We have corrected accordingly. Please refer to Lines 195-196 of the revised manuscript.

*P9L184: → Assuming that the number of...*

We have corrected accordingly. Please refer to Line 199 of the revised manuscript.

*P10L198-199: I suggest deleting two 'and's*

We have corrected accordingly. Please refer to Lines 213-215 of the revised manuscript.

*P10L200: → Note that, on...*

We have corrected accordingly. Please refer to Line 216 of the revised manuscript.

*P11L211: The word "big freeze" is a nomencreature that may refer to something else. I suggest rewording it.*

"big freeze" has been modified to "cold wave". Please refer to Line 227 of the revised manuscript.

*P11L216: → the main six particle types*

We have corrected accordingly. Please refer to Lines 232-233 of the revised manuscript.

*P11L221: → The strong K+ ion...*

We have corrected accordingly. Please refer to Line 237 of the revised manuscript.

*P11L221-222: Awkward/incomplete sentence - I suggest rephrazing the sentence.*

We have changed to "The strong K+ ion signal in the aged EC particles implies partially originated from biomass burning sources." Please refer to Lines 237-238 of the revised manuscript.

*P11L227: → suffer from the bias related to...*

We have corrected accordingly. Please refer to Lines 243-244 of the revised manuscript.

*P11L232: → ...(m/z -46NO2-, -62NO3-) and presumably derived from...*

We have corrected accordingly. Please refer to Line 249 of the revised manuscript.

*P12L234: → An aged time of 81-88 min...showed an increase...*

We have corrected accordingly. Please refer to Lines 250-251 of the revised manuscript.

*P12L246:  → the majority of aged EC*

We have corrected accordingly. Please refer to Line 264 of the revised manuscript.

*P12L246:  → Asian*

We have corrected accordingly. Please refer to Line 263 of the revised manuscript.

*P12L249-250:  → ...particles were only...droplets, and the aged EC residuals were...*

We have corrected accordingly. Please refer to Lines 268-269 of the revised manuscript.

*P12L251-252:  → The Jungraujoch station is predominantly within the free tropospheric condition, such that the biomass...*

We have corrected accordingly. Please refer to Lines 269-271 of the revised manuscript.

*P13L262:  → the size range*

We have corrected accordingly. Please refer to Line 278 of the revised manuscript.

*P13L263:  → Aqueous reactions improving ... have been observed...*

We have corrected accordingly. Please refer to Lines 279-280 of the revised manuscript.

*P13L267:  → ...amine within the cloud.*

We have corrected accordingly. Please refer to Lines 283-284 of the revised manuscript.

*P13L273:  → Previous studies showed that dust particles that are internally mixed with sulfate and nitrate promote CCN activities...*

We have corrected accordingly. Please refer to Lines 289-290 of the revised manuscript.

*P13L275: partly  → partial*

We have corrected accordingly. Please refer to Line 291 of the revised manuscript.

*P13L276: A slightly increase  → A slight increase*

We have corrected accordingly. Please refer to Line 293 of the revised manuscript.

*P13L273: Internal mixing ... is expect to act as CCN...  → Dust particles that are*

*internally mixed with sulfate and nitrate are expected to act as CCN...*

We have corrected accordingly. Please refer to Lines 289-290 of the revised manuscript.

*P13L279: → during the spring season*

We have corrected accordingly. Please refer to Line 296 of the revised manuscript.

*P14L281: → Asian dust storms that occurred in March-May*

We have corrected accordingly. Please refer to Line 298 of the revised manuscript.

*P14L283: → Local dust emissions*

We have corrected accordingly. Please refer to Line 300 of the revised manuscript.

*P14L289 → and nitrate, making up 4.1%*

We have corrected accordingly. Please refer to Line 306 of the revised manuscript.

*P14L291: contributes → contribute*

We have rephrased the sentence to "Predominant Fe ion peaks possibly indicates the contribution from anthropogenic sources". Please refer to Lines 307-308 of the revised manuscript.

*P14L293-294: → ...Fe-containing residuals have presumably come from...*

We have corrected accordingly. Please refer to Lines 310-311 of the revised manuscript.

*P14L285: may occupied → may have occupied*

We have corrected accordingly. Please refer to Line 302 of the revised manuscript.

*P14L293: → Fe-containing residuals were*

We have corrected accordingly. Please refer to Line 311 of the revised manuscript.

*P14L299-300: → ...Na-rich particles are formed from varied sources...*

We have corrected accordingly. Please refer to Lines 317-318 of the revised manuscript.

*P15L305: → The continental air masses*

We have corrected accordingly. Please refer to Line 323 of the revised manuscript.

*P15L306:  →  Industrial emissions were*

We have corrected accordingly. Please refer to Line 324 of the revised manuscript.

*P15L308:  →  This might suggest that the Na-rich particles were contributed from both industrial emissions and sea salts.*

We have corrected accordingly. Please refer to Lines 326-327 of the revised manuscript.

*P15L318:  →  iron and steel products manufacturing facilities*

We have corrected accordingly. Please refer to Lines 336-337 of the revised manuscript.

*P15L322:  →  Organic carbon tends to be...*
We have corrected accordingly. Please refer to Line 343 of the revised manuscript.

*P15L323: wealth  →  worth (I am not a big fun of too many "note" phrases. There are 13 notes in this manuscript, which seems a lot. Consider making smoother transitions and better flows between sentences/paragraphs without using too many notes).*

We have corrected accordingly and have detected some "note" in the revised manuscript.

*P16L327: secondary inorganic species?*

Secondary species contained inorganic substances (e.g., sulfate, nitrate and ammonium) and organic substances (e.g., oxalate and trimethylamine). Please refer to Figure 5 of the revised manuscript.

*P16L336:  →  to be in the form of*

We have corrected accordingly. Please refer to Line 360 of the revised manuscript.

*P16L338: Low as compared to what?*

Relative to nitrate (88-89%), low portions of ammonium (m/z, $18NH_4^+$) in the Na-rich (23% by number) and Dust (15% by number) cloud residues were found. Please refer to Lines 362-364 of the revised manuscript.

*P16L342:  →  in mass spectra (Figure 3)...*

We have corrected accordingly. Please refer to Line 366 of the revised manuscript.

*P16L343:  → Thus, our data suggest that...*

We have corrected accordingly. Please refer to Line 367 of the revised manuscript.

*P16L340:  → in two cloud residual types*

We have corrected accordingly. Please refer to Line 364 of the revised manuscript.

*P16L346:  → In this study, we found that...*

We have rephrased the sentence to "The nitrate-containing particles accounted for only 46% by number of the Aged EC cloud residues, which is significantly less than the sulfate-containing particles". Please refer to Lines 370-371 of the revised manuscript.

*P16L346:  → The data indicate that nitrate-containing particles account for...*

We have rephrased the sentence to "The nitrate-containing particles accounted for only 46% by number of the Aged EC cloud residues, which is significantly less than the sulfate-containing particles". Please refer to Lines 370-371 of the revised manuscript.

*P17L366:  → Relatively high portions...*

We have corrected accordingly. Please refer to Line 390 of the revised manuscript.

*P17L368:  → in the form of....*

We have corrected accordingly. Please refer to Line 391 of the revised manuscript.

*P17L370:  → ...(Pratt et al., 2009). It may...*

We have corrected accordingly. Please refer to Line 393 of the revised manuscript.

*P18L373:  → ...is mainly based on...*

We have corrected accordingly. Please refer to Line 396 of the revised manuscript.

*P18L374:  → particles were internally...*

We have corrected accordingly. Please refer to Line 398 of the revised manuscript.

*P18L375:  → ...to the K-rich type and probably...*

We have corrected accordingly. Please refer to Line 399 of the revised manuscript.

*P18L378: sensitive → sensitivity*

We have corrected accordingly. Please refer to Line 401 of the revised manuscript.

*P18L381: → Figure 6 displays the hourly... and Nf values of the nine types of...*

We have corrected accordingly. Please refer to Line 404 of the revised manuscript.

*P18L382-383: Awkward sentence – please rephrase.*

We have modified to "The Nf of aged EC particle type showed a very abrupt increase from cloud residues to ambient particles on Jan 17." Please refer to Lines 405-406 of the revised manuscript.

*P18L386: changed → shifted*

We have corrected accordingly. Please refer to Line 409 of the revised manuscript.

*P19L398-400: Incomplete sentence.*

We have modified to "When a cloud-free event occurred at 11:00-17:00 on 19 Jan, ambient particles remained a high level of $PM_{2.5}$ (~ 22.7 μg m$^{-3}$) during this period." Please refer to Lines 421-423 of the revised manuscript.

*P19L400: → The southwesterly...*

We have corrected accordingly. Please refer to Line 423 of the revised manuscript.

*P19L401: → from the northerly*

We have corrected accordingly. Please refer to Line 424 of the revised manuscript.

*P19L402: → North China or northern China (the authors use the South China word consistently in the manuscript... why not for North???)*

We have corrected accordingly. Please refer to Line 425 of the revised manuscript.

*P19L403: → These changes might have led...*

We have corrected accordingly. Please refer to Line 426 of the revised manuscript.

*P19L411-412: → we selected to analyze cloud residuals that...on 18-19 Jan as compared to cloud residuals that...*

We have corrected accordingly. Please refer to Lines 431-432 of the revised manuscript.

*P20L420: → for both the...*

We have corrected accordingly. Please refer to Line 434 of the revised manuscript.

*P20L431: → This data implies...*

We have corrected accordingly. Please refer to Lines 452-453 of the revised manuscript.

*P21L453: → ...with one hour intervals. The ambient...*

We have corrected accordingly. Please refer to Lines 474-475 of the revised manuscript.

*P21L456: → Thus, the data suggest that the initial...*

We have corrected accordingly. Please refer to Line 478 of the revised manuscript.

*P21L457: → cloudy air occurred around*

We have corrected accordingly. Please refer to Line 479 of the revised manuscript.

*P21L458: → in the size range of 200 nm up to 500 nm*

We have rephrased the sentence to "A reduction of supersaturation due to entrainment of the dry northern air mass might have insufficient moisture to activate small particles, leading to unactivated particles above 0.2 μm (Figure S7)". Please refer to Lines 479-482 of the revised manuscript.

*P21L458-461: Awkward/imcomplete sentences.*

We have modified to "A reduction of supersaturation due to entrainment of the dry northern air mass might have insufficient moisture to activate small particles, leading to unactivated particles above 0.2 μm (Figure S7)". Please refer to Lines 479-482 of the revised manuscript.

*P22L465: → showed that there were no significant changes*

We have corrected accordingly. Please refer to Lines 485-486 of the revised manuscript.

*P22L468: → as discussed in Sect. 3.3.*

We have corrected accordingly. Please refer to Line 489 of the revised manuscript.

*P22L473-474: do not active as CCN → are not active as CCN*

We have corrected accordingly. Please refer to Lines 494-495 of the revised manuscript.

*P22L478: In comparing → When comparing*

We have corrected accordingly. Please refer to Line 499 of the revised manuscript.

*P22L478-479: has been observed to account for → accounted for*

We have corrected accordingly. Please refer to Lines 508-509 of the revised manuscript.

*P22L484: → the differences*

We have corrected accordingly. Please refer to Line 505 of the revised manuscript.

*P23L487: → when compared to*

We have corrected accordingly. Please refer to Line 508 of the revised manuscript.

*P23L490: → nitrate-containing*

We have corrected accordingly. Please refer to Line 511 of the revised manuscript.

*P23L491: is most likely to be → possibly reflect the*

We have corrected accordingly. Please refer to Line 512 of the revised manuscript.

*P23L493: → confirmed that the update of gaseous HNO3 is an...*

We have corrected accordingly. Please refer to Line 514 of the revised manuscript.

*P23L494: → the increased nitrate level*

We have corrected accordingly. Please refer to Lines 514-515 of the revised manuscript.

*P23L495: → (Figure 9), and...*

We have corrected accordingly. Please refer to Line 516 of the revised manuscript.

*P23L495: → in the cloud residuals*

We have corrected accordingly. Please refer to Line 515 of the revised manuscript.

*P23L496: → when compared with*

We have corrected accordingly. Please refer to Lines 517-518 of the revised manuscript.

*P24L515: We did find →There was a high fraction... amine cloud residuals found when the...*

We have corrected accordingly. Please refer to Lines 534-535 of the revised manuscript.

*P24L527: → The change observed in Nf*

We have corrected accordingly. Please refer to Line 548 of the revised manuscript.

*P24L530: induce the activation of the nuclei particles to become cloud droplets → act as CCN*

We have detected the sentence due to incomplete sentence.

*Figure 2: yellow rots and fire dots sound awkward...*

We have modified to "The fire date (yellow dots) are available at https://earthdata.nasa.gov/". Please refer to Figure 2 caption of the revised manuscript.

*Figure 3: 'of the sampled particles during the whole sampling period' – not necessary.*

We have detected accordingly. Please refer to Figure 3 caption of the revised manuscript.

*Figure 7: → with an interval of one hour*

We have corrected accordingly. Please refer to Figure 7 caption of the revised manuscript.

*Figure 9: → ; whereas,*
We have corrected accordingly. Please refer to Figure 9 caption of the revised manuscript.

References:

[revised manuscript text omitted]